# Unveiling the Mechanism of Continuous Representation Full-Waveform Inversion: A Wave Based Neural Tangent Kernel Framework

**Ruihua Chen**[1], **Yisi Luo**[1,†], **Bangyu Wu**[1,†], **Deyu Meng**[1,2]

[1]School of Mathematics and Statistics, Xi'an Jiaotong University, Xi'an, Shaanxi, China
[2]Macao Institute of Systems Engineering, Macau University of Science and Technology, Taipa, Macao
`Ruihua.Chen@stu.xjtu.edu.cn`, `yisiluo1221@foxmail.com`,
`{bangyuwu, dymeng}@mail.xjtu.edu.cn`

## Abstract

Full-waveform inversion (FWI) estimates physical parameters in the wave equation from limited measurements and has been widely applied in geophysical exploration, medical imaging, and non-destructive testing. Conventional FWI methods are limited by their notorious sensitivity to the accuracy of the initial models. Recent progress in continuous representation FWI (CR-FWI) demonstrates that representing parameter models with a coordinate-based neural network, such as implicit neural representation (INR), can mitigate the dependence on initial models. However, its underlying mechanism remains unclear, and INR-based FWI shows slower high-frequency convergence. In this work, we investigate the general CR-FWI framework and develop a unified theoretical understanding by extending the neural tangent kernel (NTK) for FWI to establish a wave-based NTK framework. Unlike standard NTK, our analysis reveals that wave-based NTK is not constant, both at initialization and during training, due to the inherent nonlinearity of FWI. We further show that the eigenvalue decay behavior of the wave-based NTK can explain why CR-FWI alleviates the dependency on initial models and shows slower high-frequency convergence. Building on these insights, we propose several CR-FWI methods with tailored eigenvalue decay properties for FWI, including a novel hybrid representation combining INR and multi-resolution grid (termed IG-FWI) that achieves a more balanced trade-off between robustness and high-frequency convergence rate. Applications in geophysical exploration on Marmousi, 2D SEG/EAGE Salt and Overthrust, 2004 BP model, and the more realistic 2014 Chevron models show the superior performance of our proposed methods compared to conventional FWI and existing INR-based FWI methods.

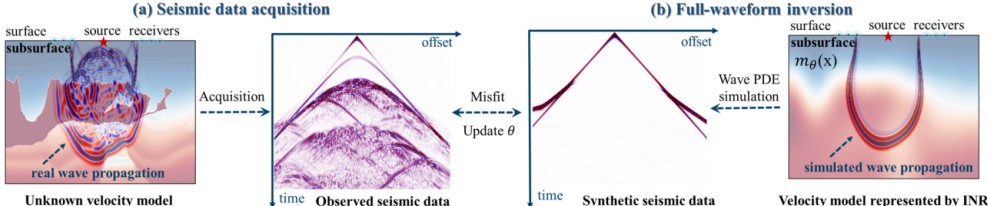

Figure 1: (a) The acquisition process of observed seismic data obtained from the receivers placed on the surface; (b) FWI is a nonlinear inverse problem that inverts the subsurface models from observed seismic data. The CR-FWI framework synthesizes seismic data by solving the wave equation 1, where the model $m(\mathbf{x})$ is represented by a coordinate-based neural network $m_\theta(\mathbf{x})$, e.g., INR.

## 1 Introduction

Full-waveform inversion (FWI) is an ill-posed and partial differential equation (PDE)-constrained (i.e., wave equation) nonlinear inversion problem in seismic imaging, which estimates subsurface properties (e.g., velocity, impedance, or density models) by iteratively minimizing the discrepancy between observed seismic data (i.e., seismograms) and synthetic seismic data (i.e., the solution of the wave equation) (Virieux & Operto, 2009). Since having a theoretically highest resolution,

---

† Corresponding authors.

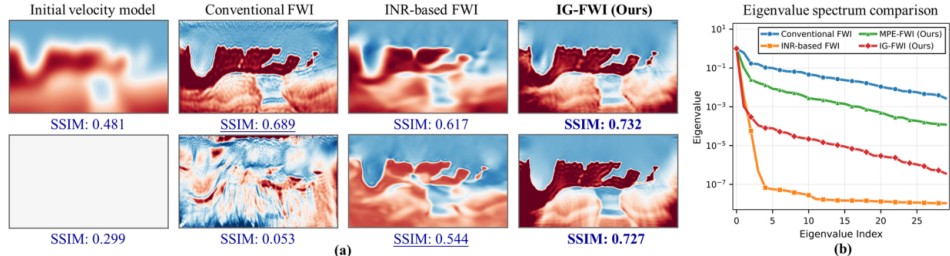

Figure 2: **(a)** Inversion results using conventional FWI (i.e., Multiscale ADFWI (Liu et al., 2025a)), existing INR-based FWI (i.e., IFWI (Sun et al., 2023a)), and our proposed IG-FWI methods with smooth (after 500 epochs) and constant initial models (after 1500 epochs) on the 2004 BP model; **(b)** Eigenvalue spectrum comparison of wave-based NTK using different FWI methods.

FWI has been widely applied across multiple domains, including geophysical subsurface exploration (Tromp, 2020), medical imaging (Guasch et al., 2020), planetary imaging (Hanasoge, 2014), and non-destructive testing (Nguyen & Modrak, 2018). However, the nonlinearity of the inversion problem, complex geological structures (e.g., salt bodies and faults), and practical limitations in seismic data acquisition (e.g., sparse sampling, missing low-frequency components, and noise interference) may pose significant challenges to achieving stable convergence towards high precision inversion results in practical applications (Virieux et al., 2017). An accurate and smooth initial velocity model, including abundant low-frequency information, can be used to mitigate the ill-posedness and cycle-skipping (i.e., half-cycle waveform misfit causing inversion failure) (Chen et al., 2022). Still, **obtaining such an accurate initial model remains a significant challenge** due to the complex near-surface conditions and heterogeneous subsurface layers (Yang et al., 2025).

To alleviate the aforementioned challenges, previous studies have explored various approaches, including initial model generation (Chen et al., 2016; Sun et al., 2017), misfit function modification (Yang & Ma, 2023), regularization design (Yan & Wang, 2018), advanced optimization algorithms (Sun & Alkhalifah, 2020), and multiscale inversion strategies (Fichtner et al., 2013). However, most of these conventional FWI methods are still sensitive to initial model accuracy and seismic data quality. Recently, Sun et al. (2023a) and Yang & Ma (2025) utilized a specific coordinate-based neural network, i.e., implicit neural representation (INR) (Mildenhall et al., 2021; Sitzmann et al., 2020) to reparameterize the velocity model as a continuous function (see Fig. 1). This novel continuous representation FWI (**CR-FWI**) framework can enhance the robustness w.r.t. the initial model accuracy and seismic data quality. Previous studies (Sun et al., 2023a; Yang & Ma, 2025) have identified two main distinctions in inversion performance between conventional FWI and CR-FWI methods:

- **Robustness**: CR-FWI can recover satisfactory inversion results even with a constant initial model and poor seismic data quality, whereas conventional FWI often fails under such conditions.
- **Convergence**: CR-FWI exhibits slower convergence rates, particularly in the high-frequency components, and requires a larger number of iterations to achieve high-precision inversion results. In contrast, conventional FWI converges more rapidly when smooth initial models are available.

We demonstrate this discrepancy in the 2004 BP model, as shown in Fig. 2 (a). In light of the above distinctions, we propose two natural and intriguing questions: *(i) Can a unified theoretical framework be established to explain the differences in robustness and convergence between conventional FWI and CR-FWI? (ii) Is there a continuous representation that achieves a balanced trade-off between robustness and convergence?* Positive answers to these questions are provided in this study.

## 1.1 OUR WORK AND CONTRIBUTIONS

In this work, we dive into the general continuous representation FWI framework and establish a unified theoretical foundation for both conventional FWI and CR-FWI. This is accomplished by introducing and analyzing a novel neural tangent kernel (NTK) tailored for FWI, i.e., the **wave kernel** for conventional FWI in Proposition 2.1 and the **wave-based NTK** for CR-FWI in Proposition 3.1. The eigenvalue of these NTKs provides a theoretical basis for understanding the convergence behavior and robustness of conventional FWI and CR-FWI. Our analysis leads to two key insights:

- **Insight I**: Distinct from standard NTK, both the wave kernel and wave-based NTK exhibit dynamic behavior during training (see Theorem 4.1) due to the nonlinearity of the inversion problem.
- **Insight II**: The eigenvalue decay of wave-based NTK is not slower than the wave kernel (see Theorem 4.2) due to the smooth kernel induced by the continuous representation (see Fig. 2 (b)).

The first insight establishes a dynamic analytical foundation for the wave-based NTK, providing a way for future theoretical research into NTK for PDE-constrained nonlinear inverse problems using stochastic analysis (Oksendal, 2013). Especially, under the quasi-static assumption, the wave-based NTK can be locally linearized, thereby enabling analysis of local convergence and optimization behavior based on its eigenvalue decay rate. The second insight reveals that the training dynamics of FWI can be decomposed along the directions defined by the eigenvectors of the wave-based NTK, each associated with a distinct convergence rate determined by the corresponding eigenvalues (Jacot et al., 2018). The rapid eigenvalue decay of wave-based NTK reveals that CR-FWI can inherently achieve effective multiscale inversion from low frequencies to high frequencies adaptively, i.e., frequency principle (Xu et al., 2025) or spectral bias (Cao et al., 2019). This property explains why CR-FWI reduces reliance on initial models and exhibits slower high-frequency convergence.

Motivated by the connection between the eigenvalue decay and its optimization behavior, we introduce several novel CR-FWI methods by incorporating low-rank tensor function representation (Luo et al., 2023) (termed LR-FWI) and multi-grid parametric encoding (Müller et al., 2022) (termed MPE-FWI). By encoding the inherent smoothness and low-rank properties of parameter models, LR-FWI improves inversion robustness and accelerates the convergence of high-frequency components due to an appropriate eigenvalue decay, as empirically verified in our numerical experiments (see Fig. 5 (c)). Moreover, the multigrid parametric encoding has been proven to yield a better eigenvalue spectrum distribution in its NTK (Audia et al., 2025). Hence, compared to INR-based FWI (Sun et al., 2023a; Yang & Ma, 2025), the proposed MPE-FWI method exhibits a slower eigenvalue decay (see Fig. 2 (b) and Theorem 5.1), which facilitates faster convergence of high-frequency components and leads to higher inversion accuracy (see Tab. 5) using smooth initial models. However, this gain in convergence speed and precision comes at the cost of reduced robustness. To inherit the robustness of INR-based FWI and the convergence benefits of MPE-FWI, we further propose a novel hybrid representation integrating INR with multi-resolution grids for FWI (termed **IG-FWI**, see Fig. 4). This method induces a suitable eigenvalue decay tailored for FWI (see Theorem 5.2 and Fig. 2 (b)), thereby achieving a trade-off between robustness and convergence (see Fig. 2 (a)).

To our knowledge, this is the first work that analyzes the FWI problem from the NTK perspective and utilizes multigrid parametric encoding for FWI. In summary, our main contributions lie in threefold:

- **Theory**: We develop a unified wave-based NTK framework for conventional FWI and CR-FWI. The eigenvalue analysis explains why CR-FWI reduces reliance on initial models and exhibits slower high-frequency convergence, with numerical tests confirming these insights (see Fig. 5).
- **Method**: Inspired by the eigenvalue decay analysis, we propose a novel integrated representation combining INR and multi-grid for FWI. This method achieves a tailored eigenvalue decay that is more suitable for FWI, leading to a better robustness-convergence trade-off (see Figs. 3 and 4).
- **Application**: We extensively evaluate our methods under challenging scenarios, including inaccurate initial models, sparse sampling, seismic data with missing low frequencies and noise interference, various benchmark datasets (i.e., Marmousi, SEG/EAGE Salt and Overthrust, and 2004 BP models), and more realistic 2014 Chevron blind data compared to conventional and existing CR-FWI methods. Inversion results show consistently superior performance (see Fig. 6).

A comprehensive review and discussion of related work (e.g., including conventional FWI, neural representation-based FWI, and NTK theory) is provided in Appendix A.

## 2 Full Waveform Inversion and Wave Kernel

Let $U$ and $T$ represent the spatial and temporal domains, respectively. For $\mathbf{x} \in U$ and $t \in T$, FWI aims to estimate the velocity model $m(\mathbf{x})$ from the observed seismic data $\{u_j^D(\mathbf{x}, t)\}_{j=1}^{N_s}$, where $u_j^D \triangleq \mathcal{O} \circ u_j$. Here, $\mathcal{O}$ denotes a linear observation operator, and $u_j(\mathbf{x}, t)$ corresponds to the full wavefield data with source wavelet function $s_j(\mathbf{x}, t)$. Mathematically, the velocity model and the full wavefield data are governed by the wave equation (Engquist & Yang, 2022) expressed as follows:

$$\begin{cases} m(\mathbf{x})\frac{\partial^2 u_j(\mathbf{x},t)}{\partial t^2} - \nabla^2 u_j(\mathbf{x}, t) = s_j(\mathbf{x}, t), & \mathbf{x} \in U, \ t \in T, \\ u_j(\mathbf{x}, 0) = 0, \quad \frac{\partial u_j}{\partial t}(\mathbf{x}, 0) = 0, & \mathbf{x} \in U, \\ \nabla u_j(\mathbf{x}, t) \cdot \mathbf{n} = 0, & \mathbf{x} \in U, \ t \in T, \end{cases} \tag{1}$$

where $\mathbf{n}$ is the unit outward normal to the boundary. This acoustic wave equation can be computed numerically via finite difference (Sun et al., 2020) (see Appendix B.3). Define the forward operator

$\mathcal{G}[\cdot] : m \mapsto u^D \triangleq [u_1^D, u_2^D, ..., u_{N_s}^D]$ and denote $D$ as the sampling space domain. FWI aims to recover the true subsurface model $m^\dagger$ by minimizing the misfit function between measurements $u_{obs}^D \triangleq \mathcal{G}[m^\dagger]$ and PDE-simulated seismic data $u_{syn}^D \triangleq \mathcal{G}[m]$ (Virieux & Operto, 2009), i.e.,

$$\min_{m \in \mathcal{A}} \left\{ \mathcal{J}(m) \triangleq \frac{1}{2} \left\| u_{syn}^D(m) - u_{obs}^D \right\|_{L^2(D \times T)}^2 \right\}, \text{ s.t. } u_{syn}^D(m) = \mathcal{G}[m], \tag{2}$$

where $\mathcal{A}$ is admissible set of velocity models and $\mathcal{J}$ is a misfit functional. In the continuous-time limit (i.e., as the learning rate tends to zero), this gradient flow of velocity model $m$ is described by

$$\frac{\partial m}{\partial \tau} = -\frac{\delta \mathcal{J}}{\delta m}[m] = -\frac{\delta \mathcal{G}}{\delta m}[m] \cdot \left( u_{syn}^D - u_{obs}^D \right), \tag{3}$$

where $\tau$ is a pseudo-time, $\delta \mathcal{J}/\delta m$ denotes the Fréchet derivative of the misfit functional $\mathcal{J}$ w.r.t. the parameter model $m$, and $\delta \mathcal{G}/\delta m$ is the sensitivity kernel operator. Applying the chain rule, we prove that the evolution of the wavefield during training is characterized by the following proposition.

**Proposition 2.1** (Evolution of wavefield in FWI). *Consider the loss function in equation 2 and the gradient flow in equation 3, the synthetic data $u_{syn}^D$ evolves according to the following equation:*

$$\frac{\partial u_{syn}^D(\mathbf{x}, t)}{\partial \tau} = -\int_D \int_T \Theta_{wave}\big((\mathbf{x}, t), (\mathbf{x}', t'); m\big) \cdot \big( u_{syn}^D - u_{obs}^D \big) dt' d\mathbf{x}', \tag{4}$$

*where $\Theta_{wave}\big((\mathbf{x}, t), (\mathbf{x}', t'); m\big) = \int_U \frac{\delta \mathcal{G}(\mathbf{x}, t)}{\delta m(\mathbf{y})}[m] \cdot \frac{\delta \mathcal{G}(\mathbf{x}', t')}{\delta m(\mathbf{y})}[m] dy$ is defined as the wave kernel.*

**Remark 2.1.** *The proof is provided in Appendix E.1. The fitting of the synthetic data is driven by the data misfit $(u_{syn}^D - u_{obs}^D)$ across the entire domain (**data-driven**), weighted by the wave kernel $\Theta_{wave}$ (**physical guide**). The wave kernel quantifies the coupling and correlation relationship between points $(\mathbf{x}, t)$ and $(\mathbf{x}', t')$ through point-wise perturbations in the model space. This often leads to uncoordinated updates and severe crosstalk, i.e., fitting one data point adversely affects others.*

## 3 WAVE-BASED NEURAL TANGENT KERNEL

Let the space of neural network parameters be denoted by $\Theta \subseteq \mathbb{R}^p$. CR-FWI utilizes a coordinate-based neural network $F_{\boldsymbol{\theta}}(\cdot) : U \to \mathbb{R}$ with learnable parameters $\boldsymbol{\theta} \in \Theta$, which maps a spatial coordinate $\mathbf{x} \in U$ to the corresponding velocity perturbation $F_{\boldsymbol{\theta}}(\mathbf{x})$. Hence, the velocity model can be reparameterized as $m_{\boldsymbol{\theta}}(\mathbf{x}) = F_{\boldsymbol{\theta}}(\mathbf{x}) + m_0(\mathbf{x})$ (where $m_0$ denotes the initial model) (Chen et al., 2025). Consider the following shallow fully-connected neural network with one hidden layer:

$$F_{\boldsymbol{\theta}}(\mathbf{x}) = \frac{1}{\sqrt{n}} W^1 \cdot \sigma(W^0 \mathbf{x} + b^0) + b^1, \quad \boldsymbol{\theta} = \{W^1, W^0, \beta^0, \beta^1\}, \tag{5}$$

where $\mathbf{x} \in U$ represents the coordinates input, $\sigma : \mathbb{R} \to \mathbb{R}$ denotes element-wise nonlinear activation function, $\boldsymbol{\theta} = \{W^1, W^0, \beta^0, \beta^1\}$ is the complete set of trainable parameters. In the usual initialization of NTK (Jacot et al., 2018; Bonfanti et al., 2024a), all the weights and biases are initialized to be independent and identically distributed as the standard normal distribution $\mathcal{N}(0, 1)$. Here, we employ the NTK rescaling factor of $1/\sqrt{n}$ as introduced in the original work (Jacot et al., 2018). This specific scaling is crucial for ensuring the convergence of the initialized neural tangent kernel (Zhou & Yan, 2024). Hence, the optimization variables of CR-FWI change from the discrete velocity model $m$ in equation 2 to neural parameters $\boldsymbol{\theta}$. The objective function can be expressed as:

$$\min_{\boldsymbol{\theta} \in \Theta} \left\{ \mathcal{J}(\boldsymbol{\theta}) \triangleq \frac{1}{2} \left\| u_{syn}^D(\boldsymbol{\theta}) - u_{obs}^D \right\|_{L^2(D \times T)}^2 \right\}, \text{ s.t. } u_{syn}^D(\boldsymbol{\theta}) = \mathcal{G}[m_{\boldsymbol{\theta}}]. \tag{6}$$

When minimizing this PDE-constrained residual via gradient descent using the CR-FWI method, the evolution of the wavefield during this flow is characterized by the following proposition:

**Proposition 3.1** (Evolution of wavefield in CR-FWI). *Consider the loss function in equation 6 and the gradient flow of the neural network in equation 5 with learning rate tends to zero, the synthetic data $u_{syn}^D$ evolves according to the following equation:*

$$\frac{\partial u_{syn}^D(\mathbf{x}, t)}{\partial \tau} = -\int_D \int_T \Theta_{wave}^{ntk}\big((\mathbf{x}, t), (\mathbf{x}', t'); \boldsymbol{\theta}\big) \cdot \big( u_{syn}^D - u_{obs}^D \big) dt' d\mathbf{x}', \tag{7}$$

*where $\Theta_{wave}^{ntk}$ is the wave-based neural tangent kernel, defined as:*

$$\Theta_{wave}^{ntk}\big((\mathbf{x}, t), (\mathbf{x}', t'); \boldsymbol{\theta}\big) = \int_U \int_U \frac{\delta \mathcal{G}(\mathbf{x}, t)}{\delta m(\mathbf{y})}[m] \cdot \frac{\delta \mathcal{G}(\mathbf{x}', t')}{\delta m(\mathbf{z})}[m] \cdot K_\tau(\mathbf{y}, \mathbf{z}; \boldsymbol{\theta}) dy d\mathbf{z}, \tag{8}$$

*where $K_\tau(\mathbf{y}, \mathbf{z}; \boldsymbol{\theta}) = \sum_{i=1}^p \frac{dm_{\boldsymbol{\theta}}(\mathbf{y})}{d\boldsymbol{\theta}_i} \cdot \frac{dm_{\boldsymbol{\theta}}(\mathbf{z})}{d\boldsymbol{\theta}_i}$ is the standard NTK of networks with $p$ parameters.*

**Remark 3.1.** *The proof can be found in Appendix E.2. When we set $m_{\boldsymbol{\theta}}(\mathbf{x}) = m(\mathbf{x})$, the proposition 3.1 degrades into 2.1 (see Proposition E.3). The wave kernel can be expressed as follows:*

$$\Theta_{wave}\big((\mathbf{x},t),(\mathbf{x}',t');m\big) = \int_U \int_U \frac{\delta\mathcal{G}(\mathbf{x},t)}{\delta m(\mathbf{y})}[m]\frac{\delta\mathcal{G}(\mathbf{x}',t')}{\delta m(\mathbf{z})}[m] \cdot K_\delta(\mathbf{y},\mathbf{z})d\mathbf{y}d\mathbf{z}, \tag{9}$$

*where $K_\delta(\mathbf{y},\mathbf{z}) \triangleq \delta(\mathbf{y}-\mathbf{z})$ is a Dirac kernel. Consequently, the wave-based NTK framework provides a unified framework for the simultaneous analysis of conventional FWI and CR-FWI.*

**Remark 3.2.** *Unlike the Dirac kernel, the wave-based NTK encodes a smooth and architecture-dependent kernel that incorporates a joint guidance mechanism derived from the product of sensitivity kernels **across different velocity model points**, rather than relying on point-wise computations in the wave kernel, which enables coordinated global updates and may help mitigate cycle-skipping.*

## 4 MAIN THEORETICAL RESULTS

Distinct from standard NTK, the wave-based NTK defined in Proposition 3.1 cannot converge to a deterministic kernel at initialization and during the training process (see the following Theorem 4.1) when the width of networks tends to infinity, due to the nonlinearity of FWI.

**Theorem 4.1.** *Consider the network defined in equation 5, which satisfies Assumption D.1. Let $m^*$ be the convergent velocity model, and suppose that for every $k \in \mathbb{N}$, there exists a time $\tau_k$ such that $\|m_{\boldsymbol{\theta}(\tau_k)} - m_*\|_U \leq \varepsilon_k$ with $\varepsilon_k \to 0$ as $k \to \infty$. Let $\boldsymbol{\theta}$ be the neural parameters under Gaussian random initialization, i.e., $\boldsymbol{\theta}(0) \sim \mathcal{N}(0,1)$, and denote by $\Theta_{\text{wave}}^{\text{ntk}}\big|_\tau^n$ the corresponding wave-based NTK with width $n$ at time $\tau$. Then, as the network width $n \to \infty$, the following hold:*

*(I) The wave-based NTK at initialization converges in distribution to a non-deterministic kernel:*

$$\Theta_{\text{wave}}^{\text{ntk}}\big|_0^n \xrightarrow{\mathcal{D}} \widehat{\Theta_{\text{wave}}^{\text{ntk}}}, \quad as\ n \to \infty.$$

*(II) The wave-based NTK during training satisfies:*

$$\lim_{n\to\infty} \sup_{\tau\in[0,\infty)} \big\|\Theta_{\text{wave}}^{\text{ntk}}\big|_\tau^n - \Theta_{\text{wave}}^{\text{ntk}}\big|_0^\infty\big\| > 0 \quad almost\ surely.$$

**Remark 4.1.** *The proof is provided in Appendix E.3. The proof core lies in the fact that the wave kernel changes in response to variations in the velocity model. Here, the global minimum convergence assumption can be guaranteed for the output of nonlinear functions (Liu et al., 2020; 2022), while having the potential to be extended to nonlinear operators in future work. By Theorem 4.1, we conclude that the wave kernel is not a deterministic kernel during the training process. The conclusion that NTK is not a deterministic kernel has also been presented in other literature and research areas, like nonlinear PINN (Bonfanti et al., 2024b) and classification problems (Yu et al., 2025).*

**Remark 4.2.** *Although the training dynamics of the wave-based NTK cannot be uniformly characterized by a fixed NTK matrix due to the stochasticity throughout the training trajectory, the time-varying NTK can still capture the local convergence behavior within sufficiently small intervals around time $\tau$ using tools such as stochastic differential equations and probability bounds (Evans, 2012), which is a promising direction for future research. Specifically, within a sufficiently small training window, the velocity model changes small, keeping the wave-based NTK nearly constant. This local stability allows the kernel's eigenvalue spectrum to quantitatively estimate local convergence rates. The wave-based NTK is a self-adjoint, compact, and non-negative definite operator (see Proposition E.2), which has a spectral decomposition $\mathcal{K}(\Theta_{\text{wave}}^{\text{ntk}}) = \sum_{k=1}^{\infty} \lambda_k \phi_k \otimes \phi_k$ (where $\mathcal{K}$ is wave-based NTK operator, $\{\lambda_k\}_{k=1}^{\infty}$ denotes eigenvalue, $\{\phi_k\}_{k=1}^{\infty}$ is the corresponding eigenfunction). The subsequent substitution of this decomposition into equation 8 leads to the following data misfit evolution equation along different spectral directions (see Proposition E.4):*

$$|\mathbf{Q}(u_{syn}^D(\tau) - u_{obs}^D)| = e^{-\boldsymbol{\Lambda}\tau}|\mathbf{Q} \cdot (u_{syn}^D(0) - u_{obs}^D)|, \tag{10}$$

*where $\Lambda = diag(\lambda_1, \lambda_2, \ldots)$ is eigenvalue sequence and $\mathbf{Q}f \triangleq \langle f, \phi_k\rangle_{k=1}^{\infty}$ denotes the spectral projection operator. Hence, larger NTK eigenvalues yield faster error reduction along their corresponding directions, i.e., the eigenvalue decay rate determines the convergence behavior.*

To explain the performance difference between conventional and INR-based FWI, the following theorem demonstrates that the latter exhibits a faster eigenvalue decay rate, i.e., INR-based FWI can capture large-scale low-frequency components to enhance the robustness and reduce the dependency on initial model accuracy, while its resolution of high-frequency details is consequently limited.

**Theorem 4.2.** *Consider the wave kernel $\Theta_{\text{wave}}$ in equation 9 and the wave-based NTK $\Theta_{\text{wave}}^{\text{ntk}}$ in equation 8 with eigenvalues $\{\lambda_j\}_{j=1}^{\infty}$ and $\{\mu_j\}_{j=1}^{\infty}$ (arranged in non-increasing order), respectively. Assume that the operator norm of $K_\tau$ satisfies $\|K_\tau\| \leq 1$. Then, for all $j \in \mathbb{N}$, we have $\mu_j \leq \lambda_j$.*

**Remark 4.3.** *The proof is provided in Appendix E.4.1. It is reasonable to assume $\|K_\tau\| \leq 1$ because the NTK is often normalized or scaled such that its largest eigenvalue is bounded by 1 (Jacot et al., 2018), which is a common practice in theoretical analysis.*

Theorem 4.2 indicates that the smooth kernel incorporated in the continuous representation truncates the rate along its high-frequency convergence direction. As a result, the rapid decay of eigenvalues in the wave-based NTK implies that low-frequency components (corresponding to larger eigenvalues) are optimized rapidly, thereby alleviating the risk of cycle-skipping and decreasing reliance on an accurate initial model. In contrast, high-frequency components, associated with the sharply decaying tail of the eigenvalue spectrum, converge more slowly due to their small eigenvalues, which lead to a slower reduction in error. By connecting the eigenvalue decay behavior of the wave-based NTK to optimization performance, such as robustness and convergence, it becomes essential to develop novel CR-FWI methods with deliberately tailored eigenvalue decay properties.

## 5 NOVEL CONTINUOUS REPRESENTATION FWI METHODS

We propose novel low-rank tensor function (LR-FWI) and multi-grid parametric encoding (MPE-FWI)-based FWI methods using continuous representations, shown in Fig. 3. LR-FWI and MPE-FWI improve the high-frequency convergence rate by reducing the attenuation rate of eigenvalues (Theorem 5.1). To further achieve a trade-off between robustness and convergence, we propose an integrated INR and multigrid representation (IG-FWI), whose eigenvalue decay rate is between INR-based and MPE-based FWI methods (Theorem 5.2).

**INR-based FWI.** Traditional INR methods (Sitzmann et al., 2020; Sun et al., 2023a; Yang & Ma, 2025) use a coordinate-based neural network (e.g., MLP) with special activation functions to parameterize the velocity model (Fig. 3 (a)). The INR can be expressed as

$$F_{\boldsymbol{\theta}}(\mathbf{x}) = \mathbf{H}_D(\sigma(\mathbf{H}_{D-1}(\cdots\sigma(\mathbf{H}_1(\mathbf{x}))))), \quad (11)$$

where $\boldsymbol{\theta} = \{\mathbf{H}_d\}_{d=1}^{D}$ are weights of the MLP and $\sigma(\cdot)$ is a scalar activation function. The choice of activation function significantly impacts the high-resolution representation capability.

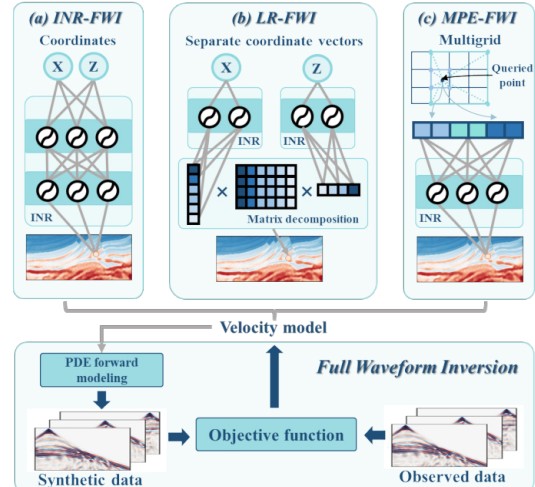

Figure 3: Pipeline of CR-FWI. CR-FWI employs (a) implicit neural representation, (b) low rank tensor function, or (c) multi-grid parametric encoding to represent the velocity parameter model and integrate the wave equation in a loop.

In seismic FWI, Sun et al. (2023a) developed IFWI using SINRE's activation function (Sitzmann et al., 2020), while Yang & Ma (2025) proposed WinFWI incorporating Gabor wavelet activation functions (Saragadam et al., 2023).

**LR-FWI.** The reparameterized subsurface geophysical parameters typically exhibit inherent structural constraints, such as low-rank and non-local similarity (Li et al., 2024a). To embed these properties, LR methods decompose the velocity model using tensor factorization methods (e.g., Tucker and CP decomposition) and represent the low-dimensional tensor separately using INRs (Luo et al., 2023), as shown in Fig. 3 (b). The general formulation of 2D LR-FWI can be expressed as:

$$F_{\boldsymbol{\theta}}(\mathbf{x}) = [\mathbf{C}; F_{\boldsymbol{\theta}_1}, F_{\boldsymbol{\theta}_2}](\mathbf{x}) = F_{\boldsymbol{\theta}_1}(x_1) \times \mathbf{C} \times F_{\boldsymbol{\theta}_2}(x_2)^{\top}, \quad (12)$$

where $\mathbf{x} = (x_1, x_2)$ denotes spatial coordinates, $F_{\boldsymbol{\theta}_1} : \mathbb{R} \to \mathbb{R}^{1 \times r_1}$, $F_{\boldsymbol{\theta}_2} : \mathbb{R} \to \mathbb{R}^{1 \times r_2}$ are one dimensional coordinate networks (e.g., INRs), and $\mathbf{C} \in \mathbb{R}^{r_1 \times r_2}$ is the core matrix. Hence, the neural parameters are $\boldsymbol{\theta} = \{\boldsymbol{\theta}_1, \boldsymbol{\theta}_2, \mathbf{C}\}$. Moreover, the LR-FWI method enhances the convergence rate for high-frequency components due to an appropriate eigenvalue decay rate, which we have empirically

verified through numerical experiments (see Fig. 5 (c)). However, providing a rigorous mathematical proof of the eigenvalue decay rate remains challenging due to the complex structure of the tensor product, which will be a key objective of our future research based on tensor decomposition theory (Kolda & Bader, 2009) and matrix analysis (Horn & Johnson, 2012).

**MPE-based FWI.** Multi-grid parametric encoding employs trainable auxiliary data structures (e.g., grid-based representations) to construct higher-dimensional embedding spaces. As illustrated in Fig. 3 (c), MPE-based FWI leverages a multigrid hash encoding (Müller et al., 2022) to represent the velocity model. The hash function $h(\cdot) : U \to \mathbb{R}^{n_g \times n_f}$ maps a coordinate point $\mathbf{x} \in U$ to a feature vector via $h(\mathbf{x}) \in \mathbb{R}^{n_g \times n_f}$, where $n_g$ denotes the number of multigrid levels and $n_f$ is the number of features per grid. Then, these interpolated features are passed through a lightweight INR to produce the physical velocity value. The overall representation can be expressed as:

$$F_{\boldsymbol{\theta}}(\mathbf{x}) = \mathrm{MLP}\left(h(\mathbf{x}); \boldsymbol{\theta}\right), \quad \text{where } h(\mathbf{x}) = \Big(\bigoplus_{i=1}^{d} x_i \pi_i\Big) \bmod T, \tag{13}$$

where $\pi_i$ are large prime numbers, $T$ is the hash table size, and $\oplus$ denotes the bit-wise exclusive-or operation. For each query point $\mathbf{x}$, the feature is obtained by interpolating the embeddings from adjacent grid vertices across multiple resolution levels (Audia et al., 2025; Luo et al., 2025).

The MPE approach integrates the representational flexibility of INR with the efficient spatial lookup offered by multigrid hash encoding, leveraging a grid structure to accelerate convergence and alleviate spectral bias (Audia et al., 2025). The effectiveness of MPE-FWI is explained in Theorem 5.1, which provides a lower bound on the eigenvalues of its wave-based NTK.

**Theorem 5.1.** *Let $\Theta_{INR}^{ntk}$ and $\Theta_{MPE}^{ntk}$ be the wave-based NTK for INR-based FWI and MPE-based FWI with eigenvalues $\{\lambda_i(\Theta_{\mathrm{INR}}^{ntk})\}_{i=1}^{\infty}$ and $\{\lambda_i(\Theta_{\mathrm{MPE}}^{ntk})\}_{i=1}^{\infty}$ in non-increasing order, respectively. Then, for a dataset of size $n$, the $i$-th eigenvalues satisfy $\lambda_i(\Theta_{MPE}^{ntk}) \geq \lambda_i(\Theta_{INR}^{ntk})$.*

**Remark 5.1.** *The proof can be found in Appendix E.4.2. The proof of this theorem is based on the fact that the eigenvalues of NTK of MPE are not smaller than those in INR, i.e., the inequality (Audia et al., 2025) $\lambda_i(\mathbf{K}_{MPE}) \geq \lambda_i(\mathbf{K}_{INR}) + \lambda_n(\mathbf{K}^+) \geq \lambda_i(\mathbf{K}_{INR})$, where $\mathbf{K}_{INR}$ and $\mathbf{K}_{MPE}$ are the NTKs for INR and the same INR composed with MPE, respectively, and $\mathbf{K}^+$ is a positive semi-definite matrix arising from the learnable grid parameters in MPE. This theorem guarantees a strict elevation of the entire eigenvalue spectrum, accelerating the learning of high-frequency components, especially starting from a smooth initial velocity model. However, the eigenvalue decay rate is inadequate to address the ill-posedness and nonlinearity inherent in FWI (see Tab. 1). Therefore, **a novel continuous representation tailored for FWI** needs to be developed, one that incorporates a suitable eigenvalue decay rate to achieve a more balanced trade-off between convergence and robustness.*

### 5.1 INTEGRATED INR-MULTIGRID HYBRID REPRESENTATION

Based on eigenvalue analysis for wave kernel and wave-based NTK, the high-frequency convergence of INR-based FWI methods is inherently limited due to faster eigenvalue decay (Theorem 4.2), whereas the robustness of classical FWI and MPE-based FWI methods remains constrained due to slower eigenvalue decay (Theorem 5.1). To achieve a balanced robustness-convergence trade-off, we propose a novel continuous representation by integrating the INR and MPE for FWI, termed IG-FWI, as shown in Fig. 4. IG-FWI employs a tiny INR to implicitly encode smooth features into the latent space, which are then combined with multi-

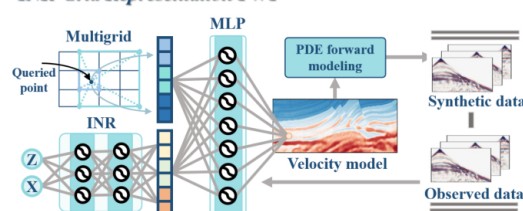

Figure 4: Pipeline of the proposed INR-Grid hybrid representation for FWI. This method combines encoding features from INR and MPE, which are then fused using an MLP network.

grid encoding features. Next, these features are combined using a tiny MLP. The reparameterized velocity model can be expressed as follows:

$$F_{\boldsymbol{\theta}}(\mathbf{x}) = \mathrm{MLP}\big(\mathbf{v}(\mathbf{x})\big), \quad \text{where } \mathbf{v}(\mathbf{x}) = \sqrt{\alpha} \cdot h(\mathbf{x}) \oplus \sqrt{(1-\alpha)} \cdot I(\mathbf{x}), \tag{14}$$

where $h(\cdot)$ is a hash encoding defined in equation 13, $I(\cdot)$ is a tiny INR defined in equation 11, and hybrid features $\mathbf{v} \in \mathbb{R}^{(n_g n_f + n_r)}$ is formed by splicing $h(\cdot)$ and $I(\cdot)$ with scaling $\sqrt{\alpha}$ and $\sqrt{1-\alpha}$,

respectively, where $n_r$ denotes the output dimension of $I(\cdot)$. The following theorem indicates that the eigenvalue decay of IG-FWI is, as desired, between that of INR-based FWI and MPE-FWI.

**Theorem 5.2.** *Let $\Theta_{\text{IG}}^{ntk}$ be the NTK of IG-FWI with eigenvalues $\{\lambda_i(\Theta_{\text{IG}}^{ntk})\}_{i=1}^{\infty}$. When INR and MPE features are normalized such that their gradient norms are comparable, then for all $i \in \mathbb{N}$,*

$$\lambda_i(\Theta_{INR}^{ntk}) \le \lambda_i(\Theta_{\text{IG}}^{ntk}) \le \lambda_i(\Theta_{MPE}^{ntk}).$$

*Hence, the eigenvalue decay rate of $\lambda_i(\Theta_{\text{IG}}^{ntk})$ lies between those of $\lambda_i(\Theta_{INR}^{ntk})$ and $\lambda_i(\Theta_{MPE}^{ntk})$.*

**Remark 5.2.** *The proof is provided in Appendix E.4.3. By balancing the eigenvalue decay in this manner, IG-FWI successfully inherits the robustness of INR-based methods and the superior convergence of MPE-based methods, thereby achieving more accurate and reliable inversion results.*

# 6 EXPERIMENTAL ANALYSIS AND APPLICATIONS

**Baselines and Implementation Details.** We carefully select widely used conventional FWI methods, including automatic differentiation-based FWI (i.e., **ADFWI**) (Liu et al., 2025a), ADFWI with TV regularization (i.e., **ADFWI-TV**) (Esser et al., 2018), ADFWI with weighted envelope correlation-based loss function (i.e., **WECI-FWI**) (Song et al., 2023), and frequency-based multi-scale inversion FWI (i.e., **MS-FWI**) (Fichtner et al., 2013), and existing state-of-the-art CR-FWI methods, including INR-based FWI with SINRE's activation function (i.e., **IFWI**) (Sun et al., 2023a) and Gabor wavelet activation function (i.e., **WinFWI**) (Yang & Ma, 2025) to compare with our proposed novel CR-FWI methods, including multigrid parametric encoding FWI (i.e., **MPE-FWI**), low-rank tensor decomposition-based FWI (i.e., **LR-FWI**), and INR-Grid based FWI (i.e., **IG-FWI**). We provide more implementation details for each baseline in the Appendix B.1. Moreover, we also provide comparisons with InversionNet (Wu & Lin, 2019b) on OpenFWI datasets.

**Datasets and Scenarios.** We evaluate the baselines and our proposed methods on Marmousi (Brougois et al., 1990), 2D SEG/EAGE Salt and Overthrust (Aminzadeh, 1996), and 2004 BP (Billette & Brandsberg-Dahl, 2005) models using different initial velocity models (i.e., smooth, linear, and constant velocity models). Moreover, we test the robustness w.r.t. the seismic data quality (i.e., noise interference, missing low frequencies, and sparse sampling) for the Marmousi model. The detailed forward modeling and acquisition system settings are provided in Appendix B.2.

## 6.1 EMPIRICAL VALIDATION OF OUR WAVE-BASED NTK THEORETICAL RESULTS

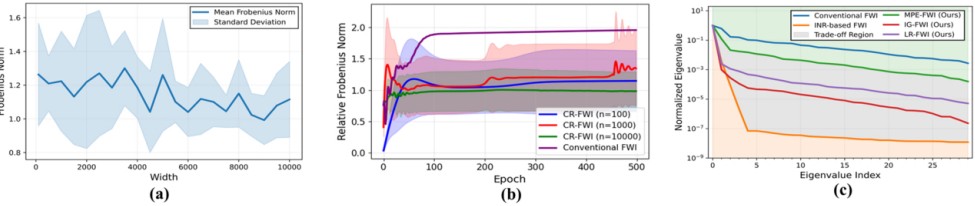

Figure 5: (a) Mean and standard deviation of the Frobenius norm of the wave-based NTK at initialization. (b) Relative Frobenius norm during training. (c) Eigenvalues decay using different methods.

To verify that the wave-based NTK does not converge to a fixed kernel either at initialization or during training, we conduct NTK experiments on a one-dimensional FWI problem. The detailed experimental setup is described in Appendix B.4, with the results illustrated in Fig. 5 (a) and (b). Furthermore, we compare the eigenvalue decay of the wave-based NTK using ADFWI (Liu et al., 2025a), INR-based FWI (Sun et al., 2023a), and our proposed MPE-FWI, LR-FWI, and IG-FWI methods, as shown in Fig. 5 (c). From these experiments, we draw two key observations:

- **Obs 1**: The wave-based NTK is non-stationary, not only during the inversion process, but also at initialization, even when the width $n$ tends to infinity, which supports Theorem 4.1.
- **Obs 2**: The eigenvalue decay is slowest in conventional FWI, faster in MPE-based FWI, and fastest in INR-based FWI. The proposed IG-FWI exhibits a decay rate between those of MPE-based and INR-based FWI, consistent with Theorems 4.2, 5.1, and 5.2. The spectrum of LR-FWI lies in a moderate region as well, and its theoretical analysis warrants discussion in the future.

## 6.2 ROBUSTNESS COMPARISONS WITH INITIAL MODELS AND DATA QUALITIES

To evaluate the performance of the FWI methods, we conduct a series of tests on both the baseline and our proposed CR-FWI methods. The experiments are carried out using four challenging datasets

Table 1: Performance comparison (MSE) of conventional FWI (ADFWI, ADFWI-TV, WECI-FWI, MS-FWI) and CR-FWI (IFWI, WinFWI, LR-FWI, MPE-FWI, IG-FWI) methods. Best results are highlighted in **bold**, and second-best are underlined. Complete results see Appendix B.6.1.

| Method | Marmousi Model | | | | | Overthrust Model | | Salt Model | | 2004 BP Model | | Computing Resource | | |
|---|---|---|---|---|---|---|---|---|---|---|---|---|---|---|
| | Smooth | Constant | G-Noise | F-Cutoff | S-Shots | Smooth | Constant | Smooth | Constant | Smooth | Constant | Time | Params | Memory |
| ADFWI | 0.2132 | 1.1522 | 0.5422 | 0.4975 | 0.4730 | 0.0592 | 1.3364 | 0.6462 | 0.6337 | 0.1003 | 0.4281 | 1.89 | 94.00 | **17.09** |
| ADFWI-TV | 0.2449 | 0.9386 | 0.3875 | 0.3764 | 0.3547 | 0.1154 | 1.3142 | 0.5041 | 0.5318 | 0.0995 | 0.4259 | 1.92 | 94.00 | **17.09** |
| WECI-FWI | 0.2977 | 1.2537 | 0.6366 | 0.3220 | 0.3337 | 0.1182 | 1.5526 | 0.3032 | 1.0293 | 0.0951 | 0.3847 | 3.12 | 94.00 | 17.56 |
| MS-FWI | 0.1683 | 1.1313 | 0.3547 | 0.4198 | 0.3325 | 0.0612 | 1.0802 | 0.5358 | 0.9234 | 0.0672 | 0.4667 | 2.25 | 94.00 | **17.09** |
| IFWI | 0.1907 | 0.9474 | 0.3374 | 0.3358 | 0.3483 | 0.0683 | 0.6432 | 0.1201 | 0.7412 | 0.0719 | 0.1412 | **1.88** | 50.05 | 17.48 |
| WinFWI | 0.2013 | 0.4689 | 0.3514 | 0.3460 | 0.3239 | 0.0682 | 0.5738 | 0.1223 | 0.3549 | 0.0812 | 0.1083 | 1.90 | 60.63 | 18.20 |
| **LR-FWI** | 0.1638 | **0.2893** | 0.2553 | 0.2276 | 0.2641 | **0.0548** | **0.1592** | 0.2785 | 0.2368 | 0.0481 | 0.1248 | 1.92 | 77.91 | 17.10 |
| **MPE-FWI** | 0.1427 | 2.2266 | 0.2990 | 0.3322 | 0.3338 | 0.0613 | 1.2887 | 0.8534 | 1.1008 | 0.0586 | 1.602 | 1.91 | **14.26** | 17.25 |
| **IG-FWI** | **0.1423** | 0.2961 | **0.2151** | **0.1846** | **0.1654** | 0.0587 | 0.5724 | **0.1181** | **0.1883** | **0.0521** | **0.0843** | 1.91 | 39.34 | 17.41 |

**G-Noise**: $8\sigma_0$ Gaussian noise added to data; **F-Cutoff**: Data with missing frequencies below 6 Hz; **S-Shots**: Only 5 shot gathers used; **Time**: average running time for one epoch (s); **Params**: Parameter count (K); **Memory**: GPU memory overhead (GB).

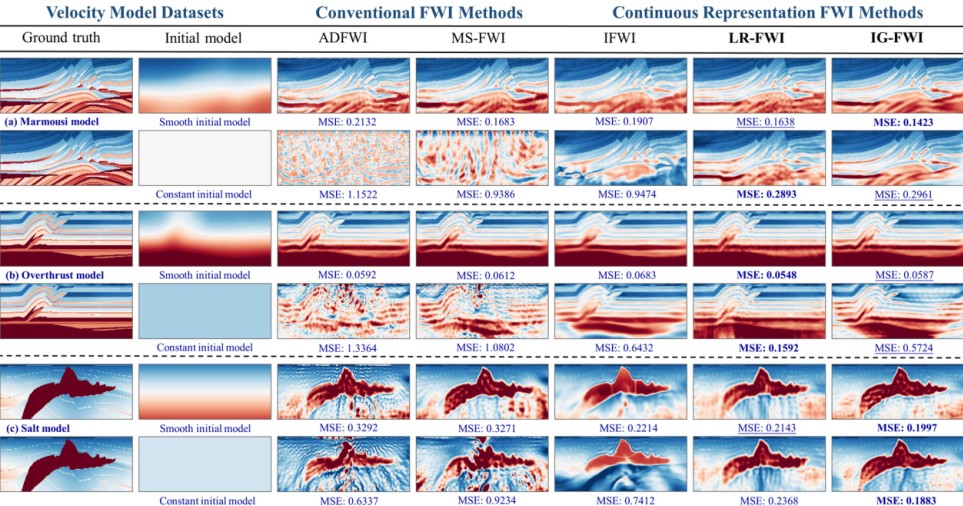

Figure 6: Conventional FWI and CR-FWI performance comparison using different initial velocity models on different velocity model datasets. More results are in the Appendices from B.6.2 to B.6.6.

with different initial models, as well as three data-degraded scenarios based on the Marmousi model. Partial results are shown in Fig. 6 and Tab. 1 (computing resource metrics are evaluated using a single shot of the BP model per training run). From these results, we draw three main observations:

- **Obs 3**: Conventional FWI (e.g., ADFWI and MS-FWI) and MPE-based FWI methods can recover high-precision velocity models with rapid high-frequency convergence when an accurate and smooth initial velocity model is available. However, the performance of these methods is highly sensitive to the quality of the initial model and the seismic data. This limitation can be attributed to the slow decay rate of eigenvalues, as established in Theorems 4.2 and 5.1.
- **Obs 4**: INR-based FWI methods (e.g., IFWI and WinFWI) can recover satisfactory inversion results even when starting from a constant initial model. However, this enhanced initial model robustness comes at the expense of a slower convergence rate and reduced resolution, which can be attributed to the relatively fast eigenvalue decay rate of the wave-based NTK (Theorem 4.2).
- **Obs 5**: Our proposed FWI methods (i.e., LR-FWI and IG-FWI) achieve a trade-off between high-precision results and robustness w.r.t. the initial model and data quality, which can be attributed to the suitably controlled eigenvalue decay rate of the wave-based NTK for FWI (Theorem 5.2).

Considering the inversion crime and computational challenges, we conduct a comparison on a more realistic 2014 Chevron model (see Appendix C.1) and the 3D Overthrust model (see Appendix C.2), which show that our proposed CRFWI can be used in more realistic large-scale inversion problems.

### 6.3 CONVERGENCE BEHAVIOR ANALYSIS FOR CRFWI

To validate the convergence behavior using different FWI methods, we provide convergence curves of velocity model error (see Fig. 7) and generated velocity visualizations corresponding to different

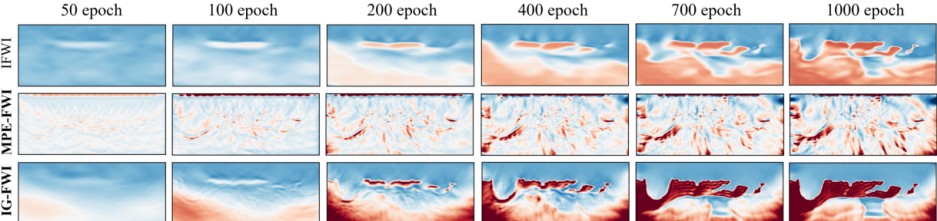

(a) Marmousi model      (b) Overthrust model      (c) Salt body model      (d) 2004 BP model

Figure 7: Convergence curves of velocity model MSE using different FWI methods and different datasets with constant initial models. (a) Marmousi. (b) Overthrust. (c) Salt body. (d) 2004 BP

Figure 8: Generated velocity visualizations corresponding to different iterations using IFWI, MPE-FWI, and IG-FWI methods on 2004 BP model. More detailed results are in Appendix B.6.7.

iterations (see Fig. 8), as detailed in Appendix B.6.7. These results indicate that conventional FWI and MPE-FWI methods achieve rapid convergence of high-frequency details at the cost of increased sensitivity, while IFWI and WinFWI enhance robustness at the expense of high-frequency details of inversion results. Our proposed IG-FWI and LR-FWI methods achieve a more balanced trade-off, maintaining robustness without sacrificing the convergence rate for high-frequency information.

### 6.4 ABLATION EXPERIMENTS FOR IG-FWI

We conduct a comprehensive ablation study to investigate the impact of key parameters (e.g., different INR frequency bases, different grid scales, and weighting factor), as shown in Fig. 9. The results show that both excessively high and low grid resolutions in MPE, as well as frequency settings in INR, may reduce the quality of inversion results, which is consistent with our theoretical results. Moreover, our proposed IG-FWI is robust to the weighting factor, which is typically set to 0.5.

### 6.5 COMPARISONS WITH DATA-DRIVEN METHOD

Evaluations on the OpenFWI dataset show that our proposed methods outperform InversionNet (Wu & Lin, 2019b) in reconstructing velocity structures from noisy data (see Tab. 2), as detailed in Appendix C.3.

## 7 CONCLUSION

In this study, we establish a unified theoretical analysis framework based on the wave kernel and wave-based NTK to investigate the convergence and robustness of both conventional and continuous representation FWI. Our analysis reveals that the wave kernel and wave-based NTK are non-stationary and exhibit distinct eigenvalue decay patterns, i.e., the wave kernel shows slow decay, whereas the wave-based NTK decays rapidly. From the theory of spectral analysis, this difference explains why

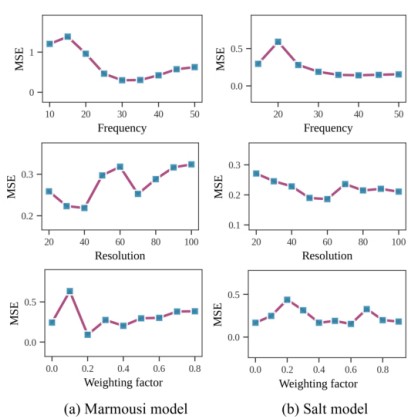

(a) Marmousi model     (b) Salt model

Figure 9: Ablation study for frequency, grid resolution, and weighting factor.

Table 2: MSE of data-driven FWI and CRFWI. Best results are highlighted in **bold**, and second-best are underlined.

| Methods | InvNet | ADFWI | **IFWI** | LR-FWI | IG-FWI |
|---|---|---|---|---|---|
| FaltVel | 36.78 | 21.30 | 11.72 | **6.65** | 6.94 |
| CurveVel | 94.16 | 57.07 | 16.87 | 9.89 | **7.93** |
| FaltFault | 113.18 | 48.23 | 46.59 | **19.59** | 28.84 |
| CurveFault | 100.09 | 99.14 | 65.89 | 60.23 | **24.61** |
| Style | 53.76 | 27.66 | 48.25 | **14.56** | 17.08 |

conventional FWI achieves fast convergence but lacks robustness, while INR-based CR-FWI offers improved robustness at the cost of slower convergence. Motivated by these findings, we generalize the INR-based CR-FWI to a broader continuous representation FWI framework, which includes the proposed LR-FWI and MPE-FWI methods. Furthermore, we introduce a hybrid representation that integrates INR with a multigrid strategy for FWI, aiming to balance robustness and convergence by better controlling the eigenvalue decay. Tests on challenging datasets and diverse seismic data scenarios show the superior performance of the proposed CR-FWI methods.

ACKNOWLEDGEMENTS

This work was supported by the Key Program of the National Natural Science Foundation of China (grant 42530802), the National Natural Science Foundation of China (No. 124B2029), the Key Program of the National Natural Science Foundation of China (grant 62476214), Fundamental and Interdisciplinary Disciplines Breakthrough Plan of the Ministry of Education of China (JYB2025XDXM101), and Tianyuan Fund for Mathematics of the National Natural Science Foundation of China (Grant No. 12426105).

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

SUPPLEMENTAL MATERIAL

This supplemental material is divided into the following five appendices.

- **Appendix A**: Related works about existing FWI, NTK, and discussion.
- **Appendix B**: Experimental details about baselines, datasets, forward modeling settings, the solution of the wave equation, wave-based NTK experiments, and main inversion results.
- **Appendix C**: additional experiments including 2014 Chevron blind test, 3D SEG/EAGE Overthrust model, and comparisons with supervised data-driven FWI method.
- **Appendix D**: Preliminary about standard NTK and hyperbolic PDE theory.
- **Appendix E**: Proof of Propositions 2.1, 3.1, and Theorems 4.1, 4.2, 5.1, and 5.2.

Comprehensive details on hyperparameters and model architectures are provided to ensure all experiments can be exactly replicated. For the theoretical components, all proofs and mathematical derivations are included in the appendix with detailed step-by-step explanations.

## A RELATED WORKS

In this section, we review relevant work covering conventional FWI methods, neural representation-based FWI methods, and the neural tangent kernel theory.

### A.1 CONVENTIONAL FWI METHODS

To alleviate the ill-posedness of FWI, many techniques like misfit functions, prior regularization, multiscale inversion strategies, and optimization algorithms have been developed. For instance, designing more convex misfit functions using correlation information (e.g., envelope (Wu et al., 2014), global correlation (Song et al., 2023)) and probability distribution (e.g., Student-T distribution (Aravkin et al., 2011), optimal transport (Yang & Ma, 2023)) to reduce the reliance on initial models. Additionally, incorporating prior knowledge using explicit regularization to enhance robustness, including local smoothness (e.g., total variation (Esser et al., 2018), Tikhonov (Yan & Wang, 2018)) and geological constraints (e.g., structural dips (Wo et al., 2025), structure tensor (Liu et al., 2025b)). Furthermore, progressively inverting velocity models from low-frequency to high-frequency (Bunks et al., 1995), or from coarse-grid to fine-grid (Fichtner et al., 2013), can mitigate the local minima problem; Using high-order optimization algorithms like the truncated Newton method (Métivier et al., 2017), the quasi-Newton method (Fabien-Ouellet et al., 2017), and the conjugate gradient method (Liu et al., 2017) can also stabilize the inversion process.

**Discussion.** Although existing conventional FWI methods can invert velocity models from inaccurate or linear initial models, they still struggle to produce satisfactory results using constant initial models. Moreover, their performance remains highly sensitive to the seismic data quality. A promising future research direction involves understanding more conventional FWI methods using our proposed wave-based NTK framework. Such analysis would help build a deeper understanding of the roles played by the misfit function, regularization, and optimization algorithms in the inversion process, ultimately facilitating the development of more advanced FWI methods.

### A.2 NEURAL REPRESENTATION-BASED FWI METHODS

Deep neural reparameterized FWI generates the velocity model using neural networks, which can be divided into data-driven supervised and physics-informed unsupervised FWI methods.

#### A.2.1 DATA-DRIVEN SUPERVISED REPRESENTATION

Data-driven supervised FWI methods directly learn the inverse operator of the wave equation by training a network from pairs of seismic data and corresponding label velocity models, like InversionNet (CNN) (Wu & Lin, 2019a), VelocityGAN (GAN) (Zhang et al., 2019), Fourier-DeepOnets (Neural Operator) (Zhu et al., 2023), PnPDP-based FWI (Diffusion model) (Zheng et al., 2025). Due to the limited availability of geophysical labeled data, numerous synthetic datasets have been developed and widely adopted in existing studies (Abdullin & Bin Waheed, 2023; Gupta et al., 2024; Jin et al., 2022), such as acoustic OpenFWI (Deng et al., 2022), elastic $\mathbb{E}^{\text{FWI}}$ (Müller et al., 2022), and GlobalTomo (Li et al., 2024b) datasets.

**Discussion.** However, the generalization performance of such supervised frameworks is constrained by the distribution shift between synthetic training data (e.g., OpenFWI) and real-world velocity models (Schuster et al., 2024), particularly in regions characterized by complex fault systems and

salt bodies. Although data-driven methods that generate initial models represent a promising alternative, this study focuses on achieving stable and high-precision inversion for challenging datasets such as the Marmousi and 2004 BP models. Our experiments with data-driven and our proposed CR-FWI methods on the OpenFWI datasets (Deng et al., 2022) demonstrated that our proposed unsupervised methods can outperform supervised approaches (see Appendix C.3).

### A.2.2 Physics-informed unsupervised representation

Distinct from pure data-driven FWI methods, physics-informed unsupervised representation utilizes a neural network, e.g., a coordinate-based neural network, to represent the velocity model and integrate the wave equation in a loop. For instance, Sun et al. (2023a) proposed implicit FWI, which represents the velocity model using an implicit neural representation with SINRE's activation function (Sitzmann et al., 2020). Subsequent advances include using INR with GELU activation (Du et al., 2024) and Gabor wavelet activation functions (Yang & Ma, 2025) to represent the velocity model. This INR-based CR-FWI method can reduce the dependence on initial models at the cost of a slow convergence rate. To our knowledge, only INR is used to represent the velocity models in FWI continuously. Moreover, Song & Alkhalifah (2021); Herrmann et al. (2023); Rasht-Behesht et al. (2022) introduced a physics-informed neural network (PINN)-based FWI methods, which use two INRs to represent the velocity model and the seismic wavefield, respectively. Physical constraints are incorporated via a penalty term derived from the wave equation within the loss function. In a different vein, Jin et al. (2022) proposed an unsupervised learning framework for FWI (termed UPFWI), which employs a CNN to predict velocity models directly from observed seismic data.

**Discussion.** Our research focuses on elucidating the mechanisms underlying the performance differences between conventional FWI methods and existing CR-FWI methods, as discussed in (Sun et al., 2023a; Yang & Ma, 2025). Building on these insights, we propose several novel CR-FWI methods specifically tailored for enhanced inversion performance. Although PINN-based FWI and UPFWI fall outside the immediate scope of this study, our proposed wave-based NTK framework can be naturally extended to these approaches, which can be a promising direction for future research.

### A.3 Neural Tangent Kernel

Standard NTK provides a mathematical framework for analyzing the training dynamics of infinitely wide neural networks and argues that the gradient descent dynamics converge to a linear kernel regression problem governed by a deterministic kernel as network width tends to infinity (Jacot et al., 2018; Arora et al., 2019). This fundamental result holds for various neural network architectures, including multilayer perceptrons (Chen et al., 2020), CNNs (Bietti & Mairal, 2019), recurrent neural networks (Alemohammad et al., 2020), and linear PINNs (Wang et al., 2022; Gan et al., 2025). However, some studies indicate that the NTK is not a deterministic kernel under some conditions and scenarios, like the last layer of the network is non-linear (Liu et al., 2020), nonlinear physics-informed neural networks (Bonfanti et al., 2024b), and classification problems using fully connected neural networks and residual neural networks (Yu et al., 2025). NTK analysis has been applied to many fields, e.g., physics-informed neural networks (Jha & Mallik, 2024; McClenny & Braga-Neto, 2023) and neural operator (Qin et al., 2024; Xu et al., 2024; Nguyen & Mücke, 2024).

**Discussion.** The proposed wave-based NTK also does not remain constant at initialization or during training, due to the inherent nonlinearity of FWI. It offers a novel theoretical perspective on continuous representation FWI as a PDE-constrained nonlinear inverse problem. In contrast to the NTK theory developed for nonlinear PINNs in the context of forward PDE problems (Bonfanti et al., 2024b), our work focuses specifically on the inverse problem and incorporates the numerical solution of the nonlinear wave equation through neural network guidance.

## B Experimental Details and Main Results

### B.1 Baselines and hyperparameter settings

We carefully select widely used conventional FWI methods and existing state-of-the-art CR-FWI methods to compare with our proposed novel CR-FWI methods, shown in Tab. 3. Moreover, we give more explanations and implementation details for each baseline and proposed method.

- **(Hyperparameter settings of conventional FWI methods)** For ADFWI with TV regularization, the isotropic constraint weights are set to $\alpha_x = 2e^{-9}$ and $\alpha_z = 2e^{-9}$, respectively, with a maximum step size of 20 and a scaling factor gamma of 0.9. Moreover, for WECI-FWI methods, we

Table 3: Conventional FWI, existing CR-FWI, and proposed CR-FWI methods.

| Category | Abbreviation | Method Name |
|---|---|---|
| Conventional FWI Methods | ADFWI (Liu et al., 2025a) | Automatic differentiation-based FWI |
| | ADFWI-TV (Esser et al., 2018) | ADFWI with total variant regularization |
| | WECI-FWI (Song et al., 2023) | Weighted envelope correlation-based FWI |
| | MS-FWI (Fichtner et al., 2013) | Frequency-based multiscale inversion FWI |
| Existing CR-FWI Methods | IFWI (Sun et al., 2023a) | INR-based FWI with SINRE's activation |
| | WinFWI (Yang & Ma, 2025) | INR-based FWI with Gabor wavelet activation |
| Proposed CR-FWI Methods | **MPE-FWI (Ours)** | Multigrid parametric encoding-based FWI |
| | **LR-FWI (Ours)** | Low-rank tensor decomposition-based FWI |
| | **IG-FWI (Ours)** | Hybrid INR-Grid based FWI |

use weighted envelope misfit and global correlation misfit (Song et al., 2023), where the weights change according to the logistic function as the number of iterations increases following (Song et al., 2023). Furthermore, for the MS-FWI method, we divide the observed seismic data into 6 frequency bands, ranging from 4 Hz to 14 Hz, with an interval of 2 Hz.

- **(Hyperparameter settings of existing CR-FWI methods)** For IFWI, we use the sine activation function with frequency hyperparameter $\omega_0 = 30$ (where $\omega_0$ controls the frequency of the sine function) for MLP, and the depth of the MLP is set to 4. The number of neurons of the MLP is set to 128, following the recommended settings in the paper (Sun et al., 2023a). Moreover, for WinFWI, we use the Gabor wavelet activation function with hyperparameters $\omega_0 = 5$ and $s_0 = 5$ (where $\omega_0$ controls the frequency of the wavelet and $s_0$ controls the frequency of the wavelet). The depth of the MLP is set to 4, and the number of neurons of the MLP is set to 200, following the recommended settings in the paper (Yang & Ma, 2025).

- **(Hyperparameter settings of our proposed CR-FWI methods)** For the proposed LR-FWI method, we employ the same sine activation function as IFWI. The rank of the low-rank matrix factorization is set to half of the model dimension. The depth of the MLPs is set to 3, and the number of neurons in the MLPs is set to 128. Moreover, the MPE-FWI method utilizes a multi-resolution hash grid (Müller et al., 2022) encoding with 16 levels, a base resolution of 50, and a per-level scale factor of 1.05. This feature grid is combined with a compact MLP with two hidden layers of 64 neurons each to model the inverse problem. Furthermore, the IG-FWI model employs a hybrid architecture, combining a multi-resolution hash grid encoding (base resolution 50 with 16 levels) with a sinusoidal feature network (2 layers of 128 neurons with $\omega_0 = 30$). These extracted features are subsequently fused and processed by a compact MLP (2 layers of 64 neurons) to generate the final physical property output.

The Adam optimizer is used to optimize the learnable parameters in all methods. The learning rate is set to 5 for conventional FWI methods and 0.0001 for CR-FWI methods. All numerical experiments are conducted using an RTX 3090 GPU, leveraging the PyTorch framework.

## B.2 DATASETS AND FORWARD MODELING SETTINGS

We evaluate baselines and our proposed methods on Marmousi (Brougois et al., 1990; Versteeg, 1994), 2D SEG/EAGE Salt and Overthrust Aminzadeh (1996), and 2004 BP models (Billette & Brandsberg-Dahl, 2005) (see Fig. 10) using different initial velocity models (i.e., smooth, linear, and constant velocity models). The detailed forward modeling parameters are as follows:

- **Marmousi model:** A $94 \times 288$ modified Marmousi model is used (15 m grid spacing) with 13 shots deployed at 300 m intervals (8 Hz Ricker wavelet). Receivers are spaced 15 m apart along the surface. Wave propagation employs a 1.9 ms time step, recording 1,000 samples, following the recommended setting in the paper (Sun et al., 2023a).
- **2D SEG/EAGE Salt model:** A $75 \times 250$ slice of the SEG/EAGE Overthrust model (10 m grid spacing) is used, with 24 shots at 100 m intervals (10 Hz Ricker wavelet). Receivers are placed at 10 m intervals on the surface. Wave propagation uses 1.5 ms time steps and 2,500 samples.
- **2D SEG/EAGE Overthrust model:** A $94 \times 401$ slice of the SEG/EAGE Salt model (15 m grid spacing) is used, with 20 shots at 270 m intervals (10 Hz Ricker wavelet). Receivers are placed at 15 m intervals on the surface. Wave propagation uses 4.0 ms time steps and 1,200 samples.
- **2004 BP model:** A $188 \times 500$ slice of the 2004 BP model (10 m grid spacing) is used, with 20 shots at 200 m intervals (10 Hz Ricker wavelet). Receivers are placed at 10 m intervals on the surface. Wave propagation uses 4.0 ms time steps and 2,000 samples.

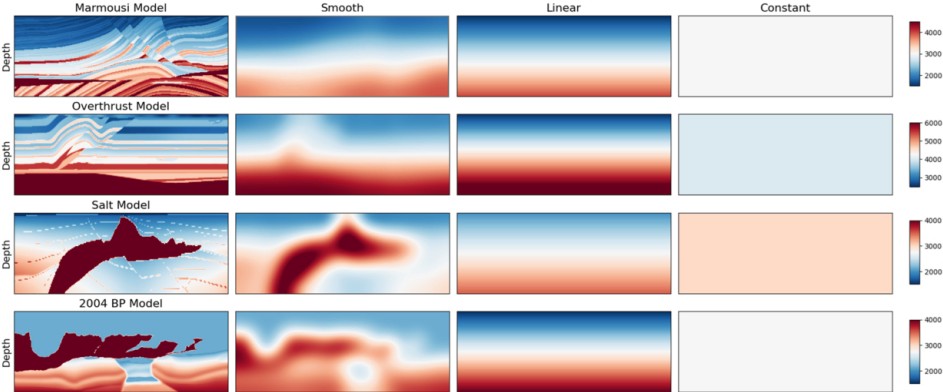

Figure 10: Datasets and Initial velocity model, including smooth, linear, and constant initial models.

Marmousi, SEG/EAGE Salt and Overthrust models configuration features a free-surface top boundary and perfectly matched layer (PML) absorbing boundaries on the remaining three sides. Moreover, we set PML absorbing boundaries on all sides for the 2004 BP model to reduce the impact of wavefield crosstalk. We employ the eighth-order finite difference method to solve the wave equation using deepwave (Simeoni et al., 2019). Representative observed seismic data are shown in Fig. 11.

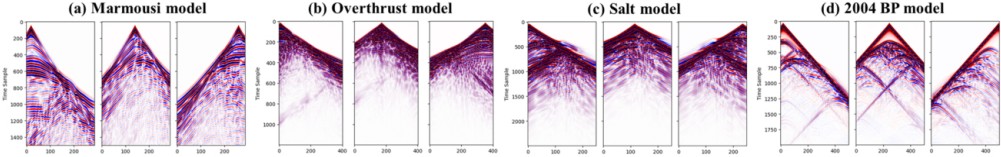

Figure 11: Observed seismic data using different velocity models, including (a) Marmousi model, (b) SEG/EAGE Overthrust model, (c) SEG/EAGE Salt model, and (d) 2004 BP model.

Furthermore, the robustness tests can be divided into three scenarios using the Marmousi model, including noisy seismic data, seismic data with missing low frequencies, and sparse acquisition.

- **(Seismic data with noise)** To evaluate the robustness of our proposed CR-FWI method against noise, we conduct noise perturbation experiments using constant initial models. Gaussian noise with standard deviations of $\sigma = 4\sigma_0$ and $\sigma = 8\sigma_0$ is added to the observed seismic data generated from the Marmousi velocity model, where $\sigma_0$ denotes the standard deviation of the original seismic data. Noise contamination is a common issue in field data acquisition due to environmental factors, which often obscure meaningful signals and degrade the inversion quality.
- **(Seismic data with missing low-frequencies)** We further assess the robustness of CR-FWI under the challenge of absent low-frequency components, a frequent limitation in field surveys caused by bandwidth constraints of sources or receivers, and attenuation effects. The observed seismic data are high-pass filtered at cutoff frequencies of 4 Hz and 6 Hz to simulate realistic scenarios where low-frequency information is unavailable. The lack of low frequencies often leads to cycle-skipping and convergence to local minima in conventional FWI.
- **(Seismic data with sparse acquisition)** To mimic realistic acquisition constraints such as limited survey access, high costs, or logistical barriers, which often result in spatially under-sampled data, we decimate the source-receiver configurations in the seismic dataset derived from the Marmousi model. Sparse acquisition leads to inadequate spatial sampling, posing significant challenges for high-resolution velocity model building. This experiment tests the ability of CR-FWI to recover accurate subsurface structures under such spatially limited conditions.

### B.3 NUMERICAL SOLUTION OF ACOUSTIC WAVE EQUATION

We define the discrete wavefield solution, velocity model, and source function as tensors $\mathcal{U} \in \mathbb{R}^{N_x \times N_z \times N_t}$, $\mathcal{M} \in \mathbb{R}^{N_x \times N_z}$, and $\mathcal{S} \in \mathbb{R}^{N_t}$, respectively, where $N_x$ and $N_z$ denote the number of discrete points in the **x** and **z** directions, and $N_t$ represents the number of time steps. The derivatives in equation 1 are approximated using the second-order central finite difference method:

$$\frac{\partial^2 \mathcal{U}^\ell}{\partial t^2} \approx \frac{\mathcal{U}^{\ell+1} - 2\mathcal{U}^\ell + \mathcal{U}^{\ell-1}}{\Delta t^2}, \quad \nabla^2 \mathcal{U}^\ell \approx \mathcal{U}^\ell * \mathbf{K}, \tag{15}$$

where $\mathcal{U}^\ell$ denotes the wavefield tensor at time step $\ell$, $\Delta t$ is the time step size, and $\mathbf{K}$ is the discrete convolution kernel for the Laplacian operator. Under the assumption that $\Delta \mathbf{x} = \Delta \mathbf{z} = \Delta h$, the kernel $\mathbf{K}$ is given by:

$$\mathbf{K} = \frac{1}{\Delta h^2} \begin{bmatrix} 0 & 1 & 0 \\ 1 & -4 & 1 \\ 0 & 1 & 0 \end{bmatrix}. \tag{16}$$

The iterative update scheme for the wavefield without considering boundary conditions is:

$$\mathcal{U}^{\ell+1} = 2\mathcal{U}^\ell - \mathcal{U}^{\ell-1} + \Delta t^2 \left[ \mathcal{M} \odot \mathcal{M} \odot \left( \mathcal{U}^\ell * \mathbf{K} - \mathcal{S}^\ell \right) \right], \tag{17}$$

where $\odot$ denotes the Hadamard (element-wise) product, and $\mathcal{S}^\ell$ is the source term at time step $\ell$. To prevent unwanted reflections from the boundaries, a Perfectly Matched Layer (PML) is incorporated. Following the approach in the provided content, we introduce auxiliary variables $p$ and $z$ to handle the PML (Pasalic & McGarry, 2010). The modified update scheme in the PML region becomes:

$$\begin{aligned} \mathcal{U}^{\ell+1} &= 2\mathcal{U}^\ell - \mathcal{U}^{\ell-1} + \Delta t^2 \left[ \mathcal{M} \odot \mathcal{M} \odot \left( \nabla^2 \mathcal{U}^\ell + \nabla p^\ell + z^\ell - \mathcal{S}^\ell \right) \right], \\ p^\ell &= a \odot p^{\ell-1} + b \odot \nabla_x \mathcal{U}^\ell, \\ z^\ell &= a \odot z^{\ell-1} + b \odot \left( \nabla^2 \mathcal{U}^\ell + \nabla_x p^\ell \right), \end{aligned} \tag{18}$$

where $a$ and $b$ are spatially varying coefficients within the PML, and $\nabla_x$ denotes the spatial derivative in the $x$-direction. The time-stepping scheme can be interpreted as a recurrent process, analogous to a recurrent neural network (RNN), where each time step corresponds to a layer in the network (Sun et al., 2020; Simeoni et al., 2019).

## B.4 WAVE-BASED NTK FOR 1D FWI EXPERIMENTAL SETTING

Our experiments are based on the following one-dimensional acoustic wave equation:

$$m(x)\partial_t^2 u(x,t) - \partial_x^2 u(x,t) = s(x,t), \tag{19}$$

where $x \in \mathbb{R}$ denotes the spatial coordinate, $t \in \mathbb{R}$ represents time, $m$ is the velocity model, $u$ is the wavefield, and $s$ is the source function. The target velocity model is constructed from trace data extracted from the Marmousi model with a grid spacing of 15 m (see Fig. 12 (a)). The seismic observations are generated using a finite-difference method (see Fig. 12 (b)). An 8 Hz Ricker wavelet is employed as the source function, with a time step of 1.9 ms and 1000 samples (see Fig. 12 (c)).

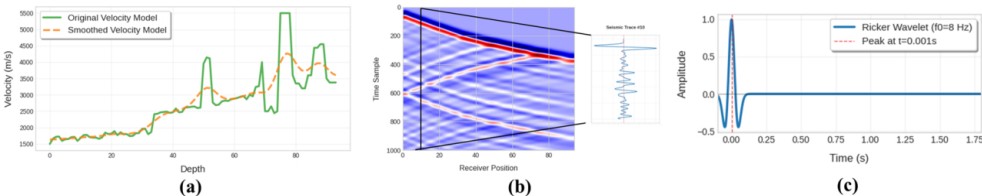

Figure 12: (a) 1D velocity model. (b) observed seismic data. (c) Ricker source function.

To compare the eigenvalue decay across different FWI methods, including ADFWI, INR-based FWI, MPE-FWI, LR-FWI, and IG-FWI. We compute the wave-based NTK for CR-FWI using a network width of 100,000 and a wave kernel for ADFWI. Furthermore, we test the wave-based NTK relative nuclear norm variation at initialization and during the training process, i.e.,

$$\Delta\Theta(\tau) \triangleq \frac{||\Theta_{\text{wave}}^{\text{NTK}}(\tau) - \Theta_{\text{wave}}^{\text{NTK}}(0)||}{||\Theta_{\text{wave}}^{\text{NTK}}(0)||}, \tag{20}$$

over the network's width, for 10 independent experiments. The results are presented in Fig. 5.

## B.5 USAGE OF LLMS

We employed large language models solely for the purpose of polishing and refining the writing of this paper. The models assisted with improving grammatical accuracy, enhancing sentence fluency, and ensuring clarity of expression, while strictly preserving the original technical content and scientific meaning. All ideas, methodologies, results, and conclusions remain entirely our own.

## B.6 MAIN EXPERIMENTAL RESULTS

### B.6.1 EVALUATION METRICS

Table 4: Comparisons between different conventional FWI (i.e., ADFWI, ADFWI-TV, WECI-FWI, and MS-FWI) and CR-FWI (i.e., IFWI, WinFWI, LR-FWI, MPE-FWI, and IG-FWI) methods across different evaluation metrics for the 2D Marmousi model. Here, **bold** represents the best evaluation result, while underline represents the second-best evaluation

| Metric | Method | Initial velocity model | | | Gaussian noise | | Frequency cutoff | | Shots number | |
|---|---|---|---|---|---|---|---|---|---|---|
| | | Smooth | Linear | Constant | $4\sigma_0$ | $8\sigma_0$ | 4 Hz | 6 Hz | 9 shots | 5 shots |
| SSIM ↑ | ADFWI | 0.6260 | 0.2689 | 0.0549 | 0.2167 | 0.1401 | 0.2594 | 0.2287 | 0.2499 | 0.1969 |
| | ADFWI-TV | 0.5466 | 0.2735 | 0.1503 | 0.2628 | 0.2456 | 0.2607 | 0.2462 | 0.2718 | 0.2289 |
| | WECI-FWI | 0.4239 | 0.3638 | 0.0220 | 0.1193 | 0.0948 | 0.3581 | 0.3475 | 0.3240 | 0.2926 |
| | MS-FWI | 0.7093 | 0.6559 | 0.0952 | 0.5274 | 0.4218 | 0.5298 | 0.4193 | 0.5362 | 0.4798 |
| | IFWI | 0.6510 | 0.6173 | 0.5252 | 0.5410 | 0.4607 | 0.5587 | 0.5124 | 0.5802 | 0.4234 |
| | WinFWI | 0.6498 | 0.5982 | 0.5617 | 0.5220 | 0.3960 | 0.5442 | 0.4514 | 0.5938 | 0.5002 |
| | **LR-FWI** | 0.6853 | 0.6662 | 0.5917 | 0.5714 | 0.5079 | 0.6414 | 0.6166 | 0.6250 | 0.5665 |
| | **MPE-FWI** | 0.7177 | 0.3681 | 0.1888 | 0.3620 | 0.3008 | 0.2828 | 0.2723 | 0.2837 | 0.2745 |
| | **IG-FWI** | 0.7183 | 0.7130 | 0.6773 | 0.6182 | 0.6048 | 0.6774 | 0.6650 | 0.7030 | 0.6808 |
| MSE ↓ | ADFWI | 0.2132 | 0.4921 | 1.1522 | 0.4965 | 0.5422 | 0.4965 | 0.4975 | 0.4831 | 0.4730 |
| | ADFWI-TV | 0.2449 | 0.3780 | 0.9386 | 0.3831 | 0.3875 | 0.3807 | 0.3764 | 0.3741 | 0.3547 |
| | WECI-FWI | 0.2977 | 0.3347 | 1.2537 | 0.5414 | 0.6366 | 0.3320 | 0.3220 | 0.3338 | 0.3337 |
| | MS-FWI | 0.1683 | 0.2651 | 1.1313 | 0.2824 | 0.3547 | 0.3556 | 0.4198 | 0.2867 | 0.3325 |
| | IFWI | 0.1907 | 0.2375 | 0.9474 | 0.3072 | 0.3374 | 0.3040 | 0.3358 | 0.2857 | 0.3483 |
| | WinFWI | 0.2013 | 0.2917 | 0.4689 | 0.3042 | 0.3514 | 0.3029 | 0.3460 | 0.2401 | 0.3239 |
| | **LR-FWI** | 0.1638 | 0.1733 | 0.2893 | 0.2218 | 0.2553 | 0.1998 | 0.2276 | 0.2064 | 0.2641 |
| | **MPE-FWI** | 0.1427 | 0.3755 | 2.2266 | 0.3271 | 0.2990 | 0.3322 | 0.3320 | 0.3338 | 0.3393 |
| | **IG-FWI** | 0.1423 | 0.1534 | 0.2961 | 0.2124 | 0.2151 | 0.1653 | 0.1846 | 0.1602 | 0.1654 |
| MAE ↓ | ADFWI | 0.2682 | 0.4571 | 0.8866 | 0.4678 | 0.5084 | 0.4626 | 0.4757 | 0.4708 | 0.4555 |
| | ADFWI-TV | 0.2800 | 0.3979 | 0.8074 | 0.4028 | 0.4084 | 0.4005 | 0.4034 | 0.4077 | 0.3874 |
| | WECI-FWI | 0.3389 | 0.3628 | 0.9170 | 0.5719 | 0.6251 | 0.3475 | 0.3605 | 0.3726 | 0.3817 |
| | MS-FWI | 0.2201 | 0.2719 | 0.8590 | 0.3314 | 0.3917 | 0.2941 | 0.3996 | 0.3146 | 0.3435 |
| | IFWI | 0.2468 | 0.2786 | 0.5568 | 0.3224 | 0.3504 | 0.3165 | 0.3415 | 0.3053 | 0.3737 |
| | WinFWI | 0.2480 | 0.3097 | 0.3979 | 0.3227 | 0.3633 | 0.3182 | 0.3536 | 0.2744 | 0.3331 |
| | **LR-FWI** | 0.2278 | 0.2472 | 0.3149 | 0.2841 | 0.3090 | 0.2495 | 0.2679 | 0.2561 | 0.2921 |
| | **MPE-FWI** | 0.2114 | 0.3977 | 1.1312 | 0.3649 | 0.3514 | 0.3791 | 0.3833 | 0.3811 | 0.3863 |
| | **IG-FWI** | 0.2109 | 0.2196 | 0.2989 | 0.2644 | 0.2706 | 0.2312 | 0.2368 | 0.2243 | 0.2307 |

Table 5: Comparisons between conventional FWI (i.e., ADFWI, ADFWI-TV, WECI-FWI, and MS-FWI) and CR-FWI (i.e., IFWI, WinFWI, LR-FWI, MPE-FWI, and IG-FWI) methods for the other challenging models. Here, **bold** represents the best evaluation result, while underline represents the second-best evaluation.

| Metric | Method | 2004 BP model | | | Salt model | | | Overthrust model | | |
|---|---|---|---|---|---|---|---|---|---|---|
| | | Smooth | Linear | Constant | Smooth | Linear | Constant | Smooth | Linear | Constant |
| SSIM ↓ | ADFWI | 0.5104 | 0.2391 | 0.0920 | 0.1717 | 0.3770 | 0.2334 | 0.7334 | 0.4699 | 0.0324 |
| | ADFWI-TV | 0.5042 | 0.2573 | 0.0963 | 0.2778 | 0.5272 | 0.3315 | 0.4497 | 0.3098 | 0.1035 |
| | WECI-FWI | 0.4392 | 0.2111 | 0.0973 | 0.3290 | 0.2525 | 0.0877 | 0.4958 | 0.1413 | 0.0931 |
| | MS-FWI | 0.6872 | 0.4157 | 0.0493 | 0.2590 | 0.4515 | 0.3020 | 0.7033 | 0.6802 | 0.0456 |
| | **IFWI** | 0.6168 | 0.5807 | 0.5533 | 0.5882 | 0.5177 | 0.4189 | 0.7124 | 0.6849 | 0.6398 |
| | **WinFWI** | 0.5926 | 0.3579 | 0.6362 | 0.5507 | 0.5205 | 0.5123 | 0.6946 | 0.6737 | 0.6148 |
| | **LR-FWI** | 0.7003 | 0.4992 | 0.4868 | 0.4222 | 0.4944 | 0.4636 | 0.7305 | 0.7005 | 0.6985 |
| | **MPE-FWI** | 0.6351 | 0.3221 | 0.0968 | 0.1554 | 0.5053 | 0.2100 | 0.6571 | 0.4506 | 0.0557 |
| | **IG-FWI** | 0.7316 | 0.6459 | 0.7274 | 0.5910 | 0.5265 | 0.5138 | 0.7272 | 0.6985 | 0.6501 |
| MSE ↓ | ADFWI | 0.1003 | 0.2568 | 0.4281 | 0.6462 | 0.3292 | 0.6337 | 0.0592 | 0.1902 | 1.3364 |
| | ADFWI-TV | 0.0995 | 0.2593 | 0.4259 | 0.5041 | 0.2143 | 0.5318 | 0.1154 | 0.1985 | 1.3142 |
| | WECI-FWI | 0.0951 | 0.2748 | 0.3847 | 0.3032 | 0.9294 | 1.0293 | 0.1182 | 0.2987 | 1.5526 |
| | MS-FWI | 0.0672 | 0.1631 | 0.4667 | 0.5358 | 0.3271 | 0.9234 | 0.0752 | 0.0797 | 1.0802 |
| | IFWI | 0.0719 | 0.1143 | 0.1412 | 0.1201 | 0.2214 | 0.7412 | 0.0683 | 0.0804 | 0.6432 |
| | WinFWI | 0.0812 | 0.1827 | 0.1083 | 0.1223 | 0.2241 | 0.3549 | 0.0682 | 0.0932 | 0.5738 |
| | **LR-FWI** | 0.0481 | 0.1193 | 0.1248 | 0.2785 | 0.2143 | 0.2368 | 0.0548 | 0.0724 | 0.1592 |
| | **MPE-FWI** | 0.0586 | 0.3387 | 1.6024 | 0.8534 | 0.2942 | 1.1008 | 0.0613 | 0.2113 | 1.2887 |
| | **IG-FWI** | 0.0521 | 0.1057 | 0.0843 | 0.1181 | 0.1997 | 0.1883 | 0.0587 | 0.0762 | 0.5724 |
| MAE ↓ | ADFWI | 0.1987 | 0.3697 | 0.5690 | 0.6168 | 0.4024 | 0.5412 | 0.1710 | 0.3182 | 0.9107 |
| | ADFWI-TV | 0.2006 | 0.3672 | 0.5685 | 0.5051 | 0.3071 | 0.4852 | 0.2653 | 0.3479 | 0.9184 |
| | WECI-FWI | 0.2024 | 0.4064 | 0.5045 | 0.3791 | 0.6999 | 0.7845 | 0.2536 | 0.4314 | 1.0064 |
| | MS-FWI | 0.1595 | 0.2717 | 0.5612 | 0.5007 | 0.3794 | 0.5705 | 0.1802 | 0.2118 | 0.8087 |
| | IFWI | 0.1637 | 0.2230 | 0.2030 | 0.2129 | 0.3200 | 0.5210 | 0.2239 | 0.2028 | 0.4672 |
| | WinFWI | 0.1883 | 0.3018 | 0.2061 | 0.2152 | 0.3284 | 0.3325 | 0.2392 | 0.1962 | 0.4564 |
| | **LR-FWI** | 0.1322 | 0.2128 | 0.2295 | 0.3272 | 0.3011 | 0.2962 | 0.1671 | 0.1907 | 0.2793 |
| | **MPE-FWI** | 0.2099 | 0.3918 | 0.6462 | 0.6667 | 0.3734 | 0.6513 | 0.1974 | 0.3392 | 0.8742 |
| | **IG-FWI** | 0.1386 | 0.2085 | 0.1839 | 0.2118 | 0.3096 | 0.2700 | 0.1702 | 0.1870 | 0.4088 |

B.6.2 INVERSION RESULTS OF MARMOUSI MODEL WITH DIFFERENT INITIAL MODELS

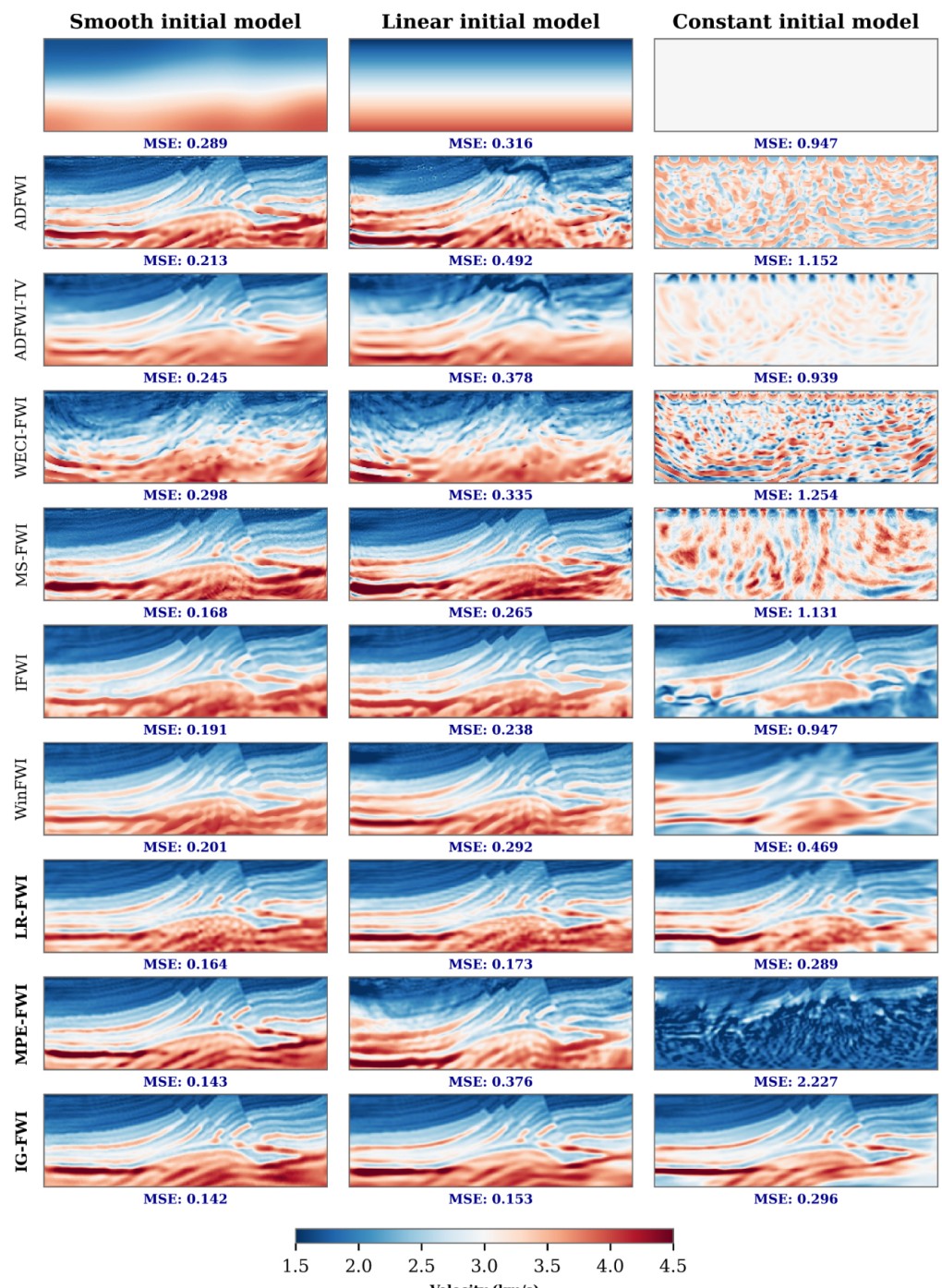

Figure 13: Conventional FWI, INR-based FWI, and our proposed FWI methods (**bold**) performance comparison using different initial velocity models (i.e., smooth, linear, and constant) on the 2D Marmousi model.

### B.6.3 INVERSION RESULTS OF MARMOUSI MODEL WITH DIFFERENT SEISMIC DATA QUALITY

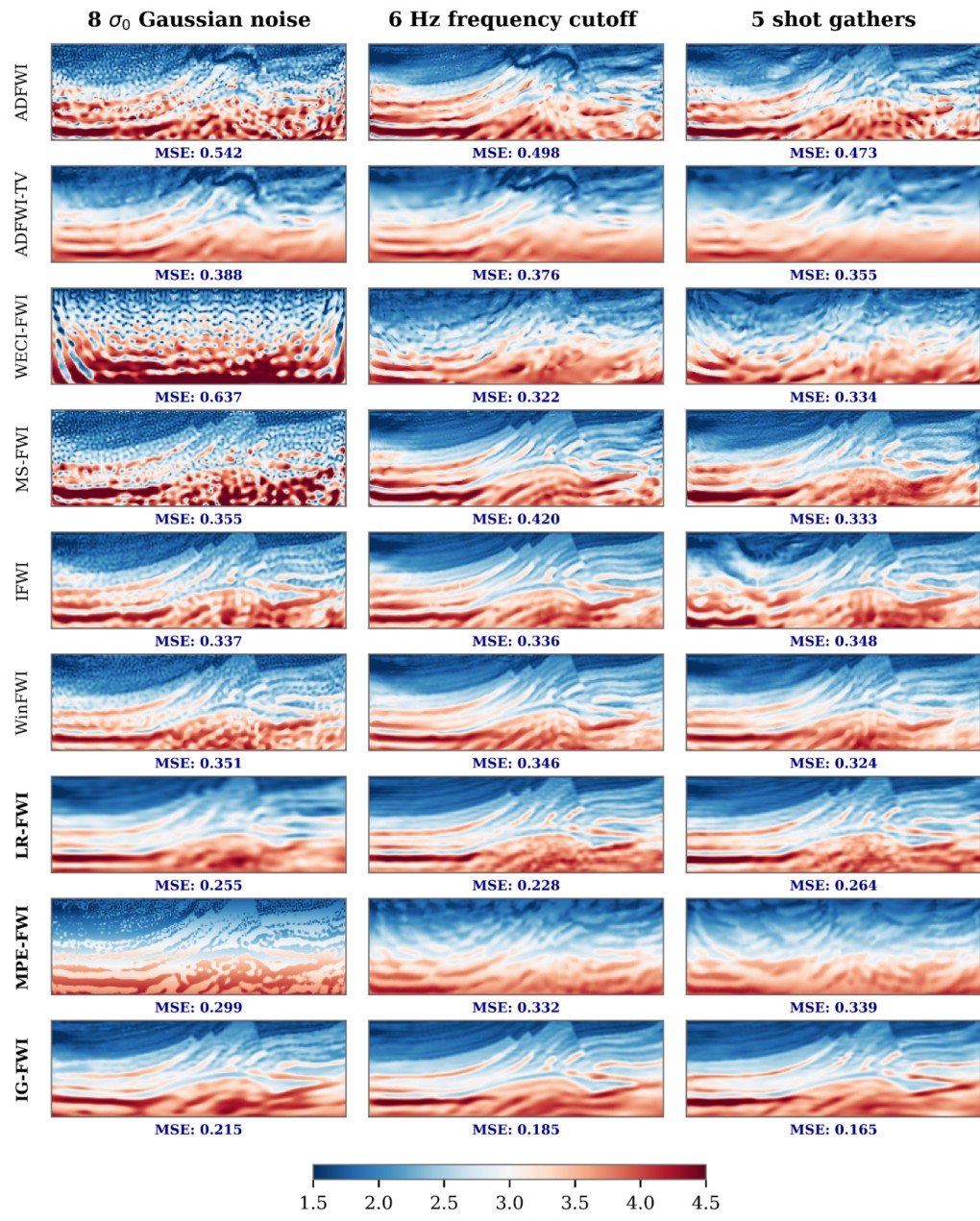

Figure 14: Conventional FWI, INR-based FWI, and our proposed FWI methods (**bold**) performance comparison using a linear initial model with different seismic data quality on the 2D Marmousi model.

B.6.4   INVERSION RESULTS OF OVERTHRUST MODEL WITH DIFFERENT INITIAL MODELS

Figure 15: Conventional FWI, INR-based FWI, and our proposed FWI methods (**bold**) performance comparison using different initial velocity models (i.e., smooth, linear, and constant) on the 2D SEG/EAGE Overthrust model.

### B.6.5 INVERSION RESULTS OF SALT MODEL WITH DIFFERENT INITIAL MODELS

Figure 16: Conventional FWI, INR-based FWI, and our proposed FWI methods (**bold**) performance comparison using different initial velocity models (i.e., smooth, linear, and constant) on the 2D SEG/EAGE Salt model.

B.6.6   INVERSION RESULTS OF 2004 BP MODEL WITH DIFFERENT INITIAL MODELS

Figure 17: Conventional FWI, INR-based FWI, and our proposed FWI methods (**bold**) performance comparison using different initial velocity models (i.e., smooth, linear, and constant) on the 2004 BP model.

### B.6.7 GENERATED VELOCITY CORRESPONDING TO DIFFERENT ITERATIONS

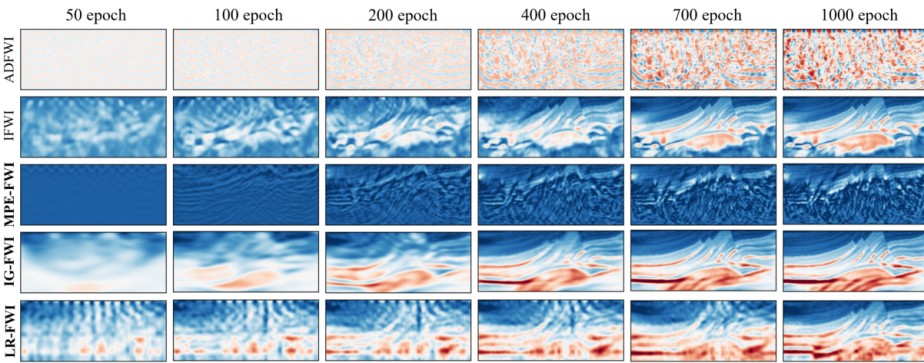

Figure 18: Generated velocity model corresponding to different iterations with conventional FWI, IFWI, MPE-FWI, IG-FWI, and LR-FWI using Marmousi model

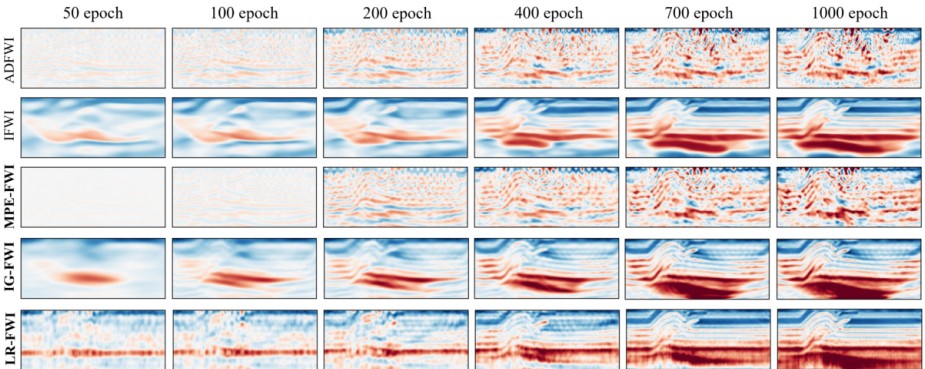

Figure 19: Generated velocity model corresponding to different iterations with conventional FWI, IFWI, MPE-FWI, IG-FWI, and LR-FWI using Overthrust model

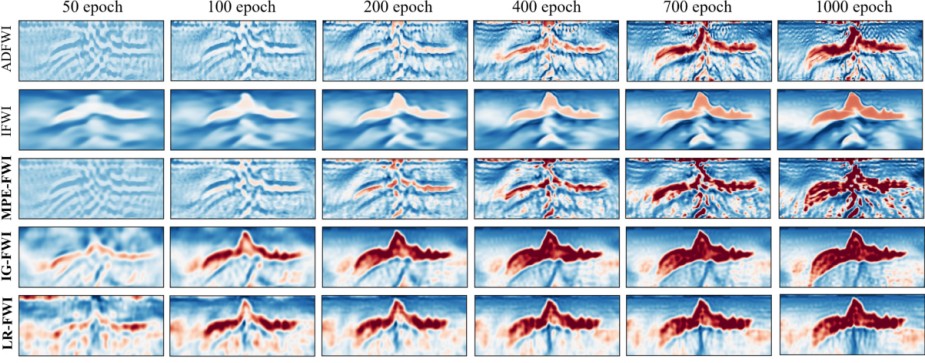

Figure 20: Generated velocity model corresponding to different iterations with conventional FWI, IFWI, MPE-FWI, IG-FWI, and LR-FWI using Salt model

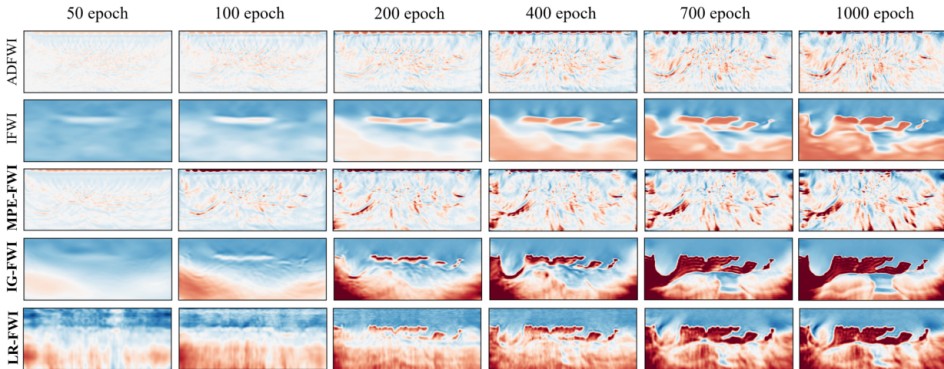

Figure 21: Generated velocity model corresponding to different iterations with conventional FWI, IFWI, MPE-FWI, IG-FWI, and LR-FWI using 2004 BP model

## C ADDITIONAL EXPERIMENTS

In this Appendix, we provide additional numerical results, including a more realistic 2014 Chevron blind test, a 3D SEG/EAGE Overthrust model test with computational challenges, and comparisons with data-driven method on the OpenFWI dataset. Finally, we present two failure cases using our proposed methods.

### C.1 2014 CHEVRON BLIND TEST AND RESULTS

To further validate our proposed IG-FWI method on more realistic field data and consider the inversion crime problem, we applied multiscale ADFWI (Liu et al., 2025a; Fichtner et al., 2013), INR-based FWI (Sun et al., 2023a), and IG-FWI to the realistic Chevron 2014 benchmark dataset.

This 2D marine towed-streamer dataset with a maximum offset of 8 km, provided by Chevron Oil Company for FWI, was modeled using a **2D isotropic, elastic wave equation** with a free surface at the top and absorbing boundaries on the sides and the bottom. It contains noise, with the signal predominating above 3 Hz. We used 146 shots with a spacing of 150 m, where each was recorded by 321 receivers spaced 25 m apart, all at a depth of 15 m. The velocity model was discretized at 25 m in both directions. Data were recorded at a 4-ms time step over 8 seconds. The source wavelet and its spectrum are shown in Fig. 22. A 1D initial velocity model and a well log were provided, but the true model was withheld.

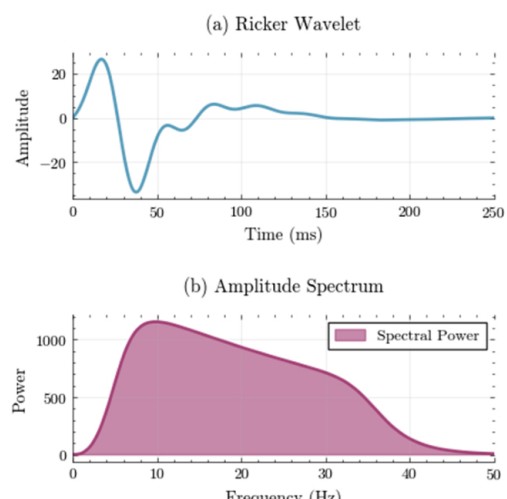

Figure 22: (a) Initial source wavelet and (b) its corresponding normalized spectrum, as provided by Chevron Oil Company for FWI.

For the multiscale FWI method, we adopt five frequency bands ranging from 3 Hz to 11 Hz with a 2 Hz interval and 50 iterations. The relevant hyperparameters for both the INR-based FWI and our proposed IG-FWI methods are set as described in Appendix B and run 500 epochs. Here, we produce the synthetic seismic data using the acoustic wave equation rather than the isotropic and elastic wave equation, which avoids the inversion crime problem. Inversion results on the Chevron blind test data (see Fig. 23) show that all methods accurately recover the shallow structures. However, challenges emerge at greater depths. For instance, velocity profiles (see Fig. 23) indicate that the obtained model aligns well with well-log data from 1 km to 2.5 km, though some discrepancies appear below 2.0 km. Despite these difficulties, IG-FWI delivers superior performance compared to both multiscale FWI and INR-based FWI. Because the true model was unknown, we

extensively assessed the inversion performance by comparing common-source shot gathers, shown in Fig. 23 Right (b). This improvement can be attributed to the suitable eigenvalue decay of IG-FWI, thereby enhancing the stability and accuracy of the inversion.

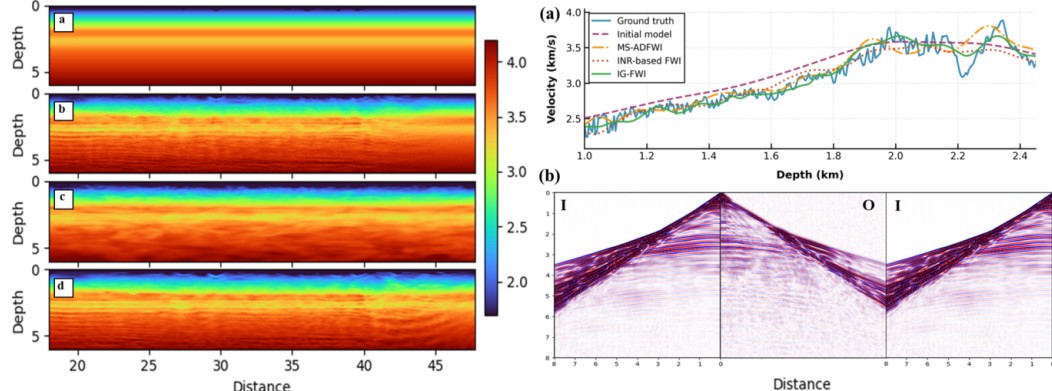

Figure 23: Inversion Results of 2014 Chevron blind test model. **Left:** (a) 2014 Chevron initial velocity model. (b) Inversion results using multiscale ADFWI (Fichtner et al., 2013; Liu et al., 2025a). (c) Inversion results using INR-based FWI (Sun et al., 2023b). (d) Inversion results using our proposed IG-FWI; **Right:** (a) Velocity profiles of the well log, initial model, and inverted models using different FWI methods. (b) Comparison of a common-source shot gather between the true and modeled records. Here $I$ denotes simulated data and $O$ denotes the observed data.

### C.2  3D SEG/EAGE Overthrust model test

Considering the computational time and memory challenges of 3D domains, we test conventional FWI, INR-based FWI, and our proposed CRFWI methods on the $60 \times 160 \times 160$ modified 3D SEG/EAGE Overthrust velocity model (50 m grid spacing) with 36 shots deployed at 1.2 km intervals (12 Hz Ricker wavelet). Receivers are placed at 50 m intervals on the surface, and wave propagation uses 2.5 ms time steps and 1000 samples. Here, we maintain the same hyperparameters and neural structure as 2D FWI settings. Moreover, we utilize the finite difference method to solve the 3D acoustic wave equation, and the representative observed seismic data are shown in Fig. 24.

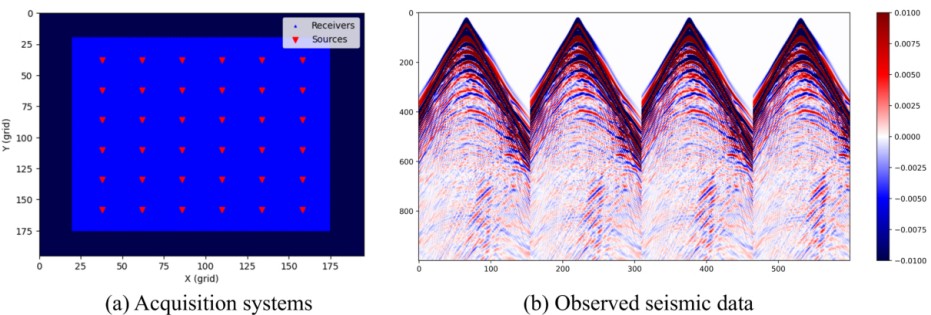

(a) Acquisition systems          (b) Observed seismic data

Figure 24:  (a) 3D acquisition systems. (b) Representative seismic data.

To alleviate the computational burden of the 3D domain, we introduce some advanced techniques like checkpointing (Anderson et al., 2012), gradient accumulation (Simeoni et al., 2019), and random mini-batch strategy (van Herwaarden et al., 2020). Hence, our proposed CRFWI methods can be implemented in one RTX 3090 GPU to conduct the 3D FWI experiments. The inversion results are shown in Fig. 25 using a smooth initial velocity model with 500 iterations. We can see that our proposed CR-FWI methods achieve more robust and high-precision inversion results. Moreover, we provide a detailed comparison of computational resources, as shown in Tab. 6. We can see that our proposed LR-FWI and IG-FWI can reduce the memory usage of the continuous representation

compared to the existing INR-based CRFWI method, which is attributed to the decoupling low-rank representation in LR-FWI and the compact hash representation in IG-FWI. These results show that our proposed CR-FWI methods do not impose additional computational burden compared with conventional FWI methods.

Table 6: Performance comparison of different FWI methods on 3D Overthrust model. Here, **bold** represents the best evaluation result, while underline represents the second-best evaluation.

| Method | Normalized MSE ↓ | SSIM ↑ | Memory (GB) | Time (s/it) |
|--------|------------------|--------|-------------|-------------|
| ADFWI | 0.1469 | 0.5576 | **15.89** | **13.82** |
| IFWI | 0.1302 | 0.4634 | 30.61 | 15.08 |
| IG-FWI | 0.1151 | 0.5729 | 22.84 | 14.61 |
| LR-FWI | **0.0968** | **0.5914** | 16.01 | 14.09 |

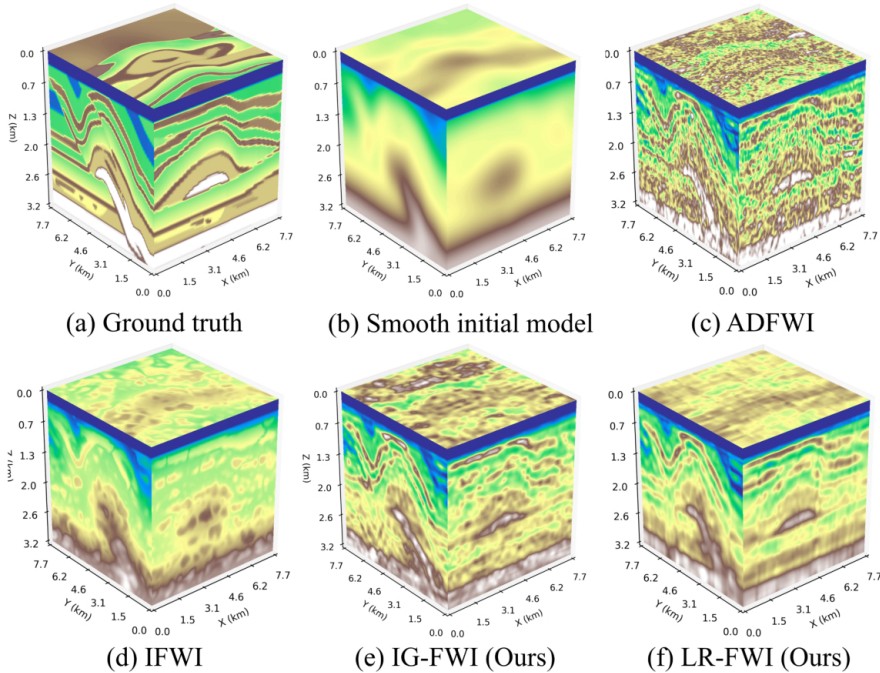

(a) Ground truth     (b) Smooth initial model     (c) ADFWI

(d) IFWI     (e) IG-FWI (Ours)     (f) LR-FWI (Ours)

Figure 25: 3D SEG/EAGE Overthrust model test using a smooth initial model. (a) Ground truth. (b) Smooth initial model. (c) ADFWI. (d) INR-based FWI. (e) IG-FWI. (f) LR-FWI.

## C.3 COMPARISONS WITH DATA-DRIVEN FWI

We evaluate our proposed CRFWI approaches with the data-driven FWI method InversionNet (Wu & Lin, 2019b) on the synthetic OpenFWI dataset (Deng et al., 2022), a large-scale benchmark designed for seismic imaging tasks. Specifically, we predict velocity models, including FlatVel, FlatFault, CurveVel, CurveFault, and Style, from noisy seismic data (see Fig. 26) using pre-trained models provided by the original InversionNet. We compare the performance of conventional FWI, INR-based FWI, and our proposed LR-FWI and IG-FWI methods in inverting velocity models from the same noisy seismic data. The inversion results indicate that both LR-FWI and IG-FWI can achieve superior performance compared to some supervised CNN-based approaches (see Figs. 27 to 31).

While supervised frameworks suffer from generalization issues caused by the distribution shift between synthetic training data (e.g., from OpenFWI) and complex field models (e.g., Marmousi and Overthrust models), their computational speed remains highly appealing. A promising pathway is to embed such methods within our proposed CR-FWI framework to rapidly generate initial models, which we plan to explore in future work.

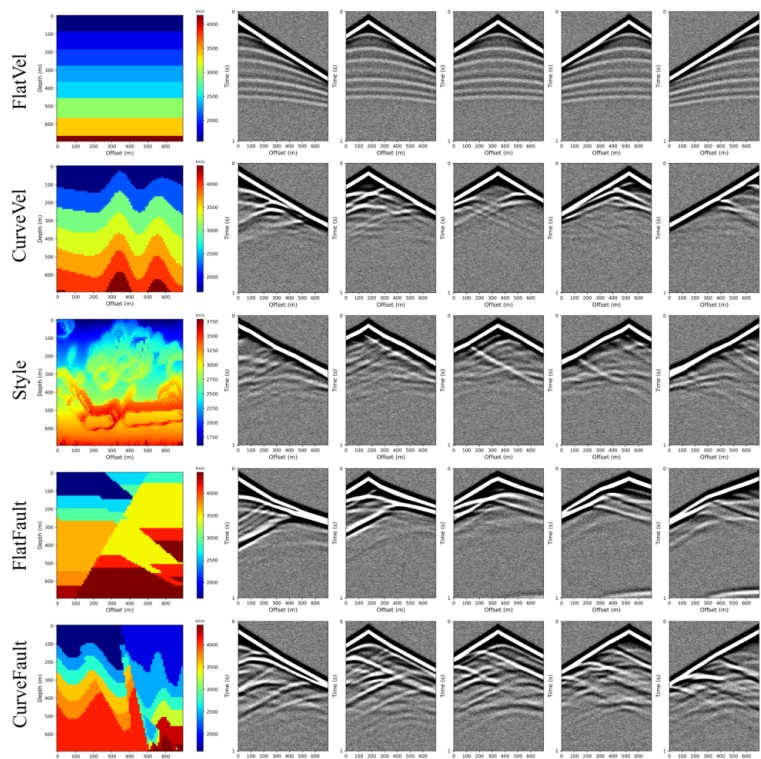

Figure 26: OpenFWI datasets and noisy seismic data input.

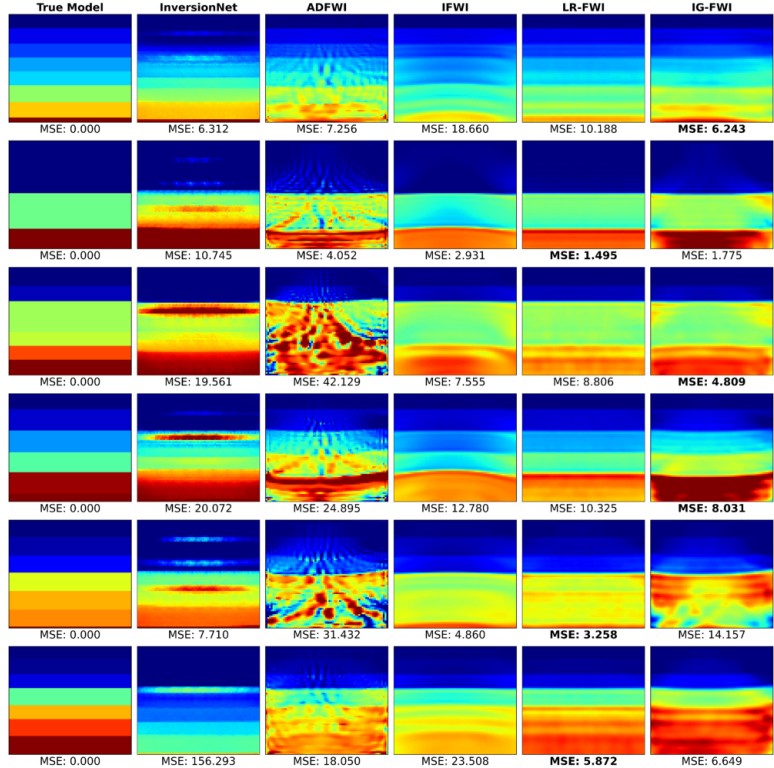

Figure 27: Inversion results on FlatVel dataset using different FWI methods.

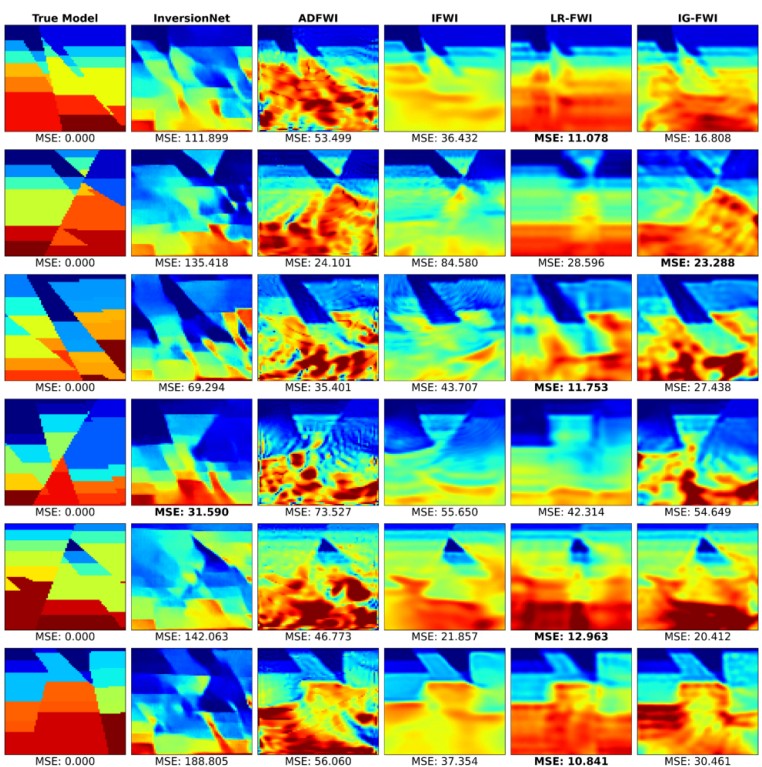

Figure 28: Inversion results on FlatFault dataset using different FWI methods.

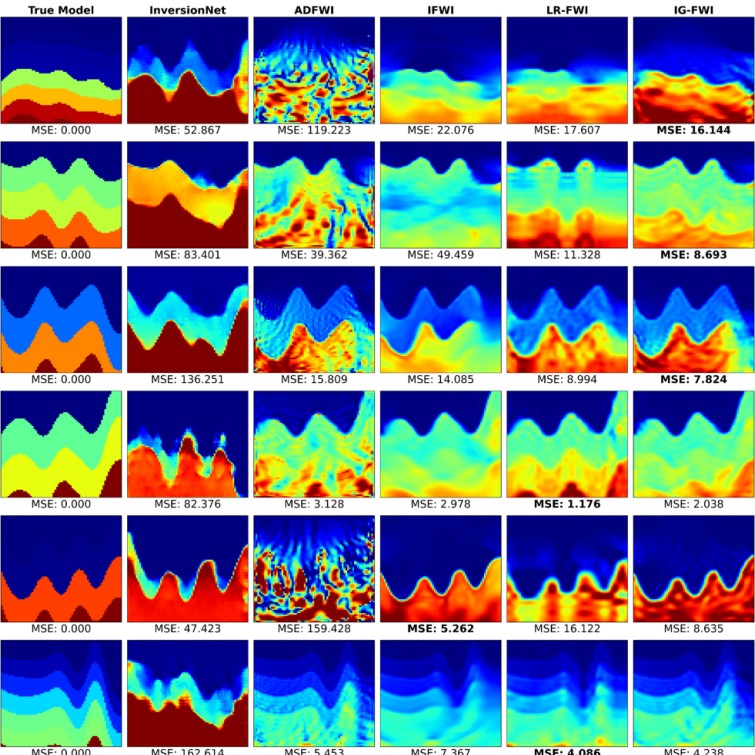

Figure 29: Inversion results on CurveVel dataset using different FWI methods.

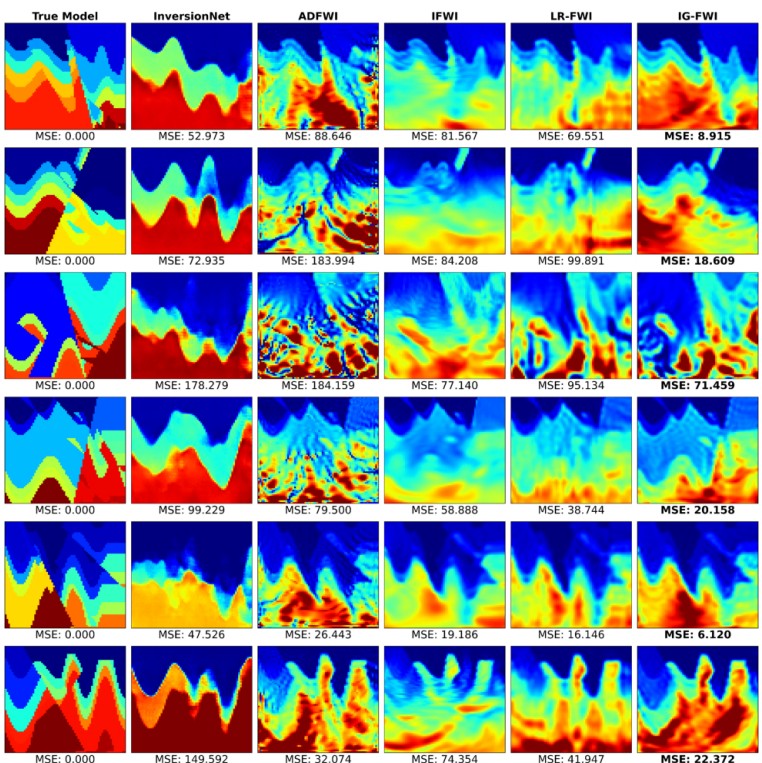

Figure 30: Inversion results on CurveFault dataset using different FWI methods.

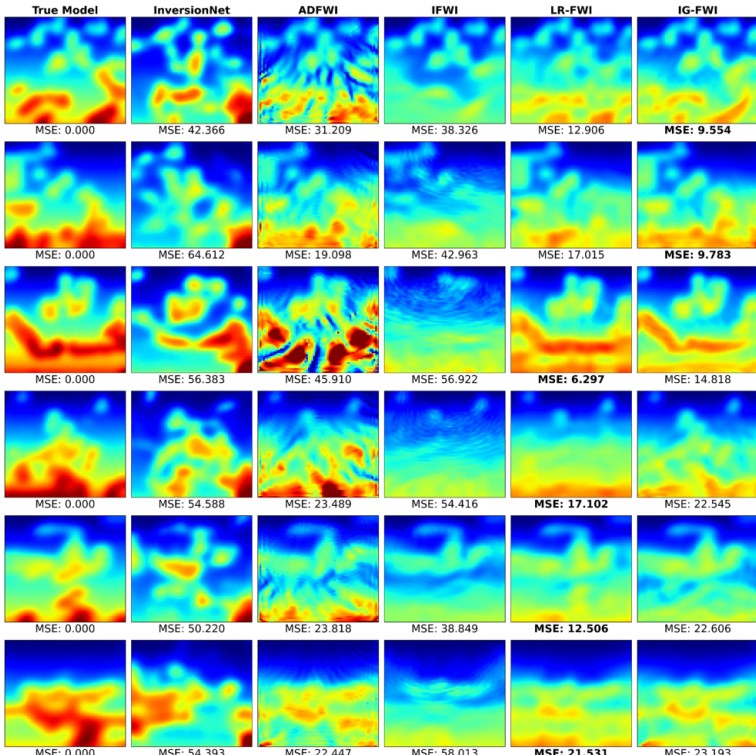

Figure 31: Inversion results on Style dataset using different FWI methods.

## C.4 TWO FAILURE FWI CASES FOR CR-FWI

Here, we present two failure cases using our proposed methods using the Canadian Foothills model with irregular topography (See Fig. 32) and the Overthrust model with a near-surface low-speed layer (See Fig. 33). The performance degradation in these challenging scenarios can be attributed to complex wavefield propagation patterns (Yang et al., 2025) that our current velocity parameterization struggles to accurately capture, which guide our subsequent algorithm design and integration with existing seismic techniques (e.g., illumination compensation techniques) in future research.

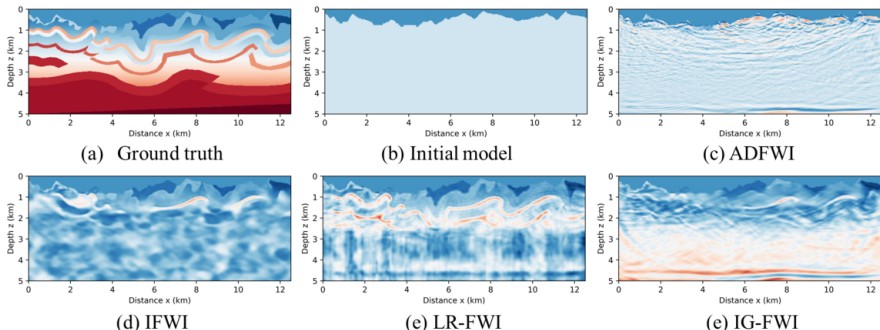

Figure 32:  Inversion results on the Canadian Foothills model with irregular topography

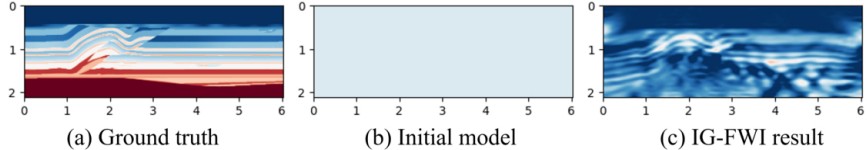

Figure 33:  Inversion results on the Overthrust model with a near-surface low-speed layer

## C.5 CONTRIBUTION TO SEISMIC COMMUNITY

Our theoretical analysis not only explain the underlying mechanism of CRFWI, but also provide a methodology of continuous representation structure design. For instance, we can develop more novel hybrid CR-FWI method by replacing the tiny INR in IG-FWI with other representation, e.g., a low-rank representation (termed LRG-FWI). Here, we provide experiments using the Marmousi, Salt and Overthrust models with a constant initial model using LRG-FWI (see Fig. 34 and Tab. 7). We see that this novel CR-FWI achieves high-precision inversion results and achieves performance comparable to that of IG-FWI .

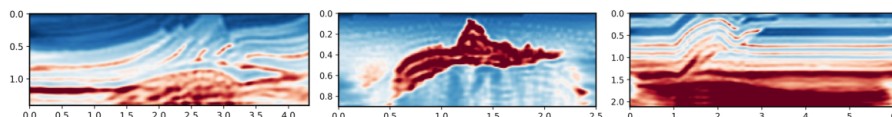

Figure 34:  Inversion results using LRG-FWI method.

Table 7: Comparisons with LRG-FWI and other CR-FWI methods.

| Method | MPE-FWI | LR-FWI | LRG-FWI | IG-FWI |
|---|---|---|---|---|
| Marmousi (MSE ↓) | 2.2266 | 0.2893 | 0.1995 | 0.2961 |
| Overthrust (MSE ↓) | 1.2887 | 0.1592 | 0.1354 | 0.5724 |
| Salt body (MSE ↓) | 1.1008 | 0.2368 | 0.2179 | 0.1883 |

## THEORETICAL PROOF

Next, we present the proofs of the main theorems introduced in this paper. The overall structure of the proofs is illustrated in Fig. 35. We begin with preliminary foundations, including the con-

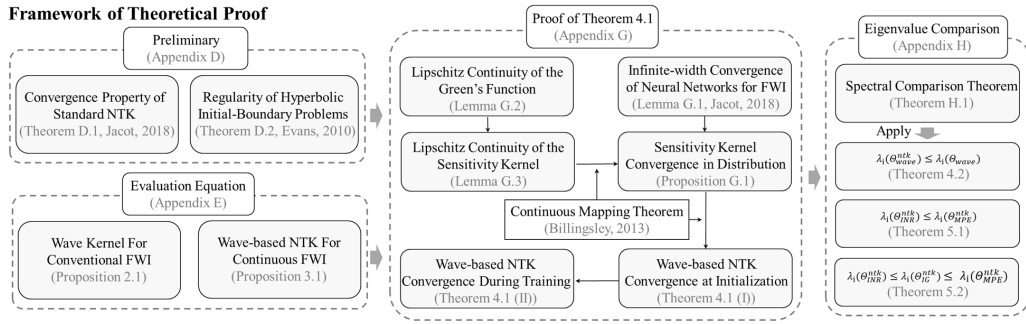

Figure 35: Framework of our theoretical proof.

vergence properties of the standard Neural Tangent Kernel, regularity conditions for hyperbolic initial-boundary problems, and the formulation of the evaluation equation. Subsequently, the proof of Theorem 4.1 is structured around key lemmas such as the Lipschitz continuity of the Green's function and the sensitivity kernel, supported by the continuous mapping theorem. This section also establishes the convergence of the wave-based NTK during training and at initialization, alongside distributional convergence of the sensitivity kernel. The final part focuses on eigenvalue comparisons, proposing and applying a spectral comparison theorem to demonstrate several inequalities regarding eigenvalues under different parameter settings, as stated in Theorems 4.2, 5.1, and 5.2.

## D  PRELIMINARY

### D.1  NOTATIONS

We denote scalars, vectors, and matrices by $x$, $\mathbf{x}$, and $\mathbf{X}$, respectively. The spatial domain is represented as an open set $U \subset \mathbb{R}^d$, with boundary $\partial U$, and the space-time domain as $U_T = U \times [0, T)$. Scalar-valued and vector-valued functions are denoted by $u : U \to \mathbb{R}$ and $\mathbf{u} : U \to \mathbb{R}^m$, respectively. The first and second derivatives of a function $\sigma(\cdot)$ are denoted by $\sigma'(\cdot)$ and $\sigma''(\cdot)$. The gradient of a function $f(\mathbf{w}, \mathbf{x})$ with respect to $\mathbf{w}$ is written as $\nabla_{\mathbf{w}} f(\mathbf{w}, \mathbf{x}) \in \mathbb{R}^d$. For function spaces, $C^k(U)$ refers to the space of $k$-times continuously differentiable functions, and $L^p(U)$ denotes the Lebesgue space of $p$-integrable functions, where $1 \leq p \leq +\infty$. Regarding norms, for vectors, $\|\cdot\|$ and $\|\cdot\|_{\infty}$ represent the Euclidean norm and the infinity norm, respectively; for matrices, $\|\cdot\|$ and $\|\cdot\|_F$ denotes the spectral norm and the Frobenius norm. For functions, the $L^p$-norm and the essential supremum norm are denoted by $\|\cdot\|_{L^p(U)}$ and $\|\cdot\|_{L^\infty(U)}$, respectively.

### D.2  STANDARD NTK THEORY

**Assumption D.1.** *The neural network $F_{\boldsymbol{\theta}}$ defined in equation 5 satisfies the following properties:*

*1. **Parameter boundedness**: There exists a constant $C > 0$, independent of the hidden layer width $n$, such that the network parameters remain uniformly bounded:*

$$\sup_{\tau} \|\boldsymbol{\theta}(\tau)\|_{\infty} \leq C.$$

*2. **Smoothness of activation function**: The activation function $\sigma$ belongs to $C^{k+1}(\mathbb{R})$ for some $k \geq 2$, and there exists a constant $C > 0$ such that for any scalar input $x \in \mathbb{R}$,*

$$\max_{0 \leq i \leq k+1} \sup_{x \in \mathbb{R}} \left| \sigma^{(i)}(x) \right| \leq C.$$

**Remark D.1.** *This is a common technical condition in related literature (Wang et al., 2022).*

Next, we summarize prior theoretical results concerning the standard NTK. Consider a neural network or, generally, a machine learning model $f_\tau(\mathbf{x}, \boldsymbol{\theta})$ with learnable parameters $\boldsymbol{\theta} \in \mathbb{R}^p$, like equation 5. The empirical NTK is defined as the inner product of the gradients of the network's output w.r.t. all of its parameters. Denoted by $K_\tau(\mathbf{x}, \mathbf{x}'; \boldsymbol{\theta}) : \mathbb{R}^d \times \mathbb{R}^d \to \mathbb{R}$, it is given by:

$$K_\tau(\mathbf{x}, \mathbf{x}'; \boldsymbol{\theta}) = \langle \nabla_{\boldsymbol{\theta}} f_\tau(\mathbf{x}, \boldsymbol{\theta}), \nabla_{\boldsymbol{\theta}} f_\tau(\mathbf{x}', \boldsymbol{\theta}) \rangle = \sum_{i=1}^{p} \frac{\partial f_\tau(\mathbf{x}, \boldsymbol{\theta})}{\partial \boldsymbol{\theta}_i} \cdot \frac{\partial f_\tau(\mathbf{x}', \boldsymbol{\theta})}{\partial \boldsymbol{\theta}_i}. \tag{21}$$

Under the infinite-width limit, the empirical NTK converges to a deterministic kernel $\widetilde{K}(\mathbf{x}, \mathbf{x}')$ that remains constant throughout training, a phenomenon referred to as the "lazy training" regime.

**Theorem D.1** (**Convergence Property of Standard NTK**). *Consider a neural network $f_\tau(\mathbf{x}, \boldsymbol{\theta})$ of width $n$ like equation 5, under Assumption D.1. Suppose the parameters $\boldsymbol{\theta}(0)$ are initialized as i.i.d. draws from $\mathcal{N}(0, 1)$. Then, as the width $n \to \infty$, the empirical NTK $K_\tau(\mathbf{x}, \mathbf{x}'; \boldsymbol{\theta})$ converges in probability to a deterministic kernel $\widetilde{K}(\mathbf{x}, \mathbf{x}')$, both at initialization and during training, i.e.,*

$$K_\tau(\mathbf{x}, \mathbf{x}'; \boldsymbol{\theta}) \xrightarrow{\mathbb{P}} \widetilde{K}(\mathbf{x}, \mathbf{x}').$$

*Moreover, this convergence is uniform over all input pairs $(\mathbf{x}, \mathbf{x}')$ and all time points $\tau \geq 0$. Specifically, denote $\boldsymbol{\theta}_0$ as initial parameters, under certain regularity conditions, with high probability,*

$$\sup_{\mathbf{x}, \mathbf{x}'} \sup_{\tau \geq 0} |K_\tau(\mathbf{x}, \mathbf{x}'; \boldsymbol{\theta}) - K(\mathbf{x}, \mathbf{x}', \boldsymbol{\theta}_0))| \xrightarrow{\mathbb{P}} 0,$$

*where $\xrightarrow{\mathbb{P}}$ denotes convergence in probability.*

**Remark D.2.** *The proof can be found in (Jacot et al., 2018). This theorem elucidates the fundamental concept of the lazy training regime. The convergence of the empirical NTK to a static, deterministic limit $\widetilde{K}$ implies that the function $f_\tau(\mathbf{x}, \boldsymbol{\theta})$ evolves in a manner asymptotically equivalent to a linear model in the feature space defined by the gradient at initialization. Consequently, the training dynamics of an infinitely wide neural network under gradient descent are governed by the deterministic kernel $\widetilde{K}$, simplifying the analysis of its convergence and generalization.*

### D.3 HYPERBOLIC PARTIAL DIFFERENTIAL EQUATION THEORY

Let $\Omega \subset \mathbb{R}^n$ be a bounded domain with $C^2$ boundary $\partial\Omega$. Consider the second-order linear partial differential operator depending on time $t$:

$$L(t) = -\sum_{i,j=1}^{n} a_{ij}(t, \mathbf{x}) \frac{\partial^2}{\partial x_i \partial x_j} + \sum_{i=1}^{n} b_i(t, \mathbf{x}) \frac{\partial}{\partial x_i} + c(t, \mathbf{x}), \tag{22}$$

And the hyperbolic initial-boundary value problem:

$$\begin{cases} \partial_t^2 u(t, \mathbf{x}) + L(t)u(t, \mathbf{x}) = f(t, \mathbf{x}), & \mathbf{x} \in \Omega, \, t > 0, \\ u(t, \mathbf{x}) = 0, & \mathbf{x} \in \partial\Omega, \, t > 0, \\ u(0, \mathbf{x}) = \varphi(\mathbf{x}), \quad \partial_t u(0, \mathbf{x}) = \psi(\mathbf{x}), & \mathbf{x} \in \Omega. \end{cases} \tag{23}$$

Next, we consider the regularity for hyperbolic initial-boundary value problems in equation 23.

**Theorem D.2** (**Regularity for Hyperbolic Initial-Boundary Value Problems**). *Assume the coefficients satisfy $a_{ij}, \partial_t a_{ij}, \partial_t^2 a_{ij} \in C^1(\overline{\Omega}_T)$ $(i, j = 1, 2, \ldots, n)$ and $b_i, \partial_t b_i, c, \partial_t c \in L^\infty(\Omega_T)$ $(i = 1, 2, \ldots, n)$. There exists $\theta > 0$ such that for all $(t, \mathbf{x}) \in [0, T] \times \Omega$ and $\xi \in \mathbb{R}^n$,*

$$\sum_{i,j} a_{ij}(t, \mathbf{x}) \xi_i \xi_j \geq \theta |\xi|^2,$$

*and that the source and boundary functions satisfy:*

$$f \in H^1([0, \infty), L^2(\Omega)), \quad \varphi \in H^2(\Omega) \cap H_0^1(\Omega), \quad \psi \in H_0^1(\Omega).$$

*Let $u$ be a weak solution of (4.4.20). Then:*

$$\begin{cases} u \in L^\infty([0, \infty), H^2(\Omega) \cap H_0^1(\Omega)), \\ \partial_t u \in L^\infty([0, \infty), H_0^1(\Omega)), \\ \partial_t^2 u \in L^\infty([0, \infty), L^2(\Omega)), \end{cases} \tag{24}$$

*and the following estimate holds for all $T > 0$:*

$$\|u\|_{L^\infty([0,T],H^2(\Omega))} + \|\partial_t u\|_{L^\infty([0,T],H^1(\Omega))} + \|\partial_t^2 u\|_{L^\infty([0,T],L^2(\Omega))}$$
$$\leq C(T) \left( \|f\|_{H^1((0,T),L^2(\Omega))} + \|\varphi\|_{H^2(\Omega)} + \|\psi\|_{H^1(\Omega)} \right),$$

*where $C(T) > 0$ is a constant depending on $T$ and the coefficients of $L(t)$.*

*Proof.* This proof can be found in (Cui, 2015) and (Evans, 2010). □

**Assumption D.2.** *Let $U \subset \mathbb{R}^d$ be a bounded domain with $C^2$ boundary, and let $T > 0$. Consider the acoustic wave equation equation 1. The velocity model $m$ belongs to the admissible set:*

$$\mathcal{A} = \{m \in C^1(\mathbb{R}^d) : 0 < m_1 \leq m(\mathbf{x}) \leq m_2 < \infty \text{ for a.e. } \mathbf{x} \in \mathbb{R}^d\},$$

*where $m_1$ and $m_2$ are positive constants. The source term $s_j$ is assumed to be in $H^1((0,T); L^2(U))$. The domain $D$ (where the receivers are located) is disjoint from the source domain $U$, i.e., there exists a constant $\delta > 0$ such that $\mathrm{dist}(D, U) \geq \delta$.*

**Remark D.3.** *This assumption can be easily satisfied using an implicit neural representation (Sitzmann et al., 2020) using an activation function with $C^1$ continuity, such as IFWI (Sun et al., 2023a) and WinFWI (Yang & Ma, 2025).*

**Proposition D.1** (Regularity for the Acoustic Wave Equation). *Under Assumption D.2, consider the acoustic wave equation equation 1 with zero initial conditions:*

$$u(0, \mathbf{x}) = 0, \quad \partial_t u(0, \mathbf{x}) = 0 \quad \text{for } \mathbf{x} \in U.$$

*Then, for any $T > 0$, there exists a unique weak solution $u_j$ satisfying the regularity properties:*

$$\begin{cases} u_j \in L^\infty(0, T; H^2(U) \cap H_0^1(U)), \\ \partial_t u_j \in L^\infty(0, T; H_0^1(U)), \\ \partial_t^2 u_j \in L^\infty(0, T; L^2(U)), \end{cases}$$

*and the following energy estimate holds:*

$$\|u_j\|_{L^\infty(0,T;H^2(U))} + \|\partial_t u_j\|_{L^\infty(0,T;H^1(U))} + \|\partial_t^2 u_j\|_{L^\infty(0,T;L^2(U))}$$
$$\leq C(T, U, m_1, m_2) \|s_j\|_{H^1((0,T);L^2(U))}.$$

*Proof.* We rewrite the acoustic wave equation equation 1 in the form:

$$\partial_t^2 u - Lu = s_j(t, \mathbf{x}), \tag{25}$$

where the spatial operator is $Lu = -m^2(\mathbf{x})\Delta u$. This fits the framework of Theorem D.2 with coefficients $a_{ij}(\mathbf{x}) = m^2(\mathbf{x})\delta_{ij} \in C^1(\overline{U_T})$, and $b_i = c = 0$. The uniform ellipticity condition is satisfied since $m(\mathbf{x}) \geq m_1 > 0$ implies:

$$\sum_{i,j=1}^d a_{ij}(\mathbf{x})\xi_i\xi_j = m^2(\mathbf{x})|\boldsymbol{\xi}|^2 \geq m_1^2|\boldsymbol{\xi}|^2 \quad \text{for all } \boldsymbol{\xi} \in \mathbb{R}^d. \tag{26}$$

The zero initial conditions satisfy $\varphi = \psi = 0$, and the source term $s_j$ has the required regularity. Applying Theorem D.2 yields the desired regularity and the estimate, where the constant $C$ depends on $T$, the domain $U$, and the bounds $m_1, m_2$. □

**Remark D.4.** *This proposition implies that the wavefield $u_j$ and its second time derivative $\partial_t^2 u_j$ are bounded in the corresponding norms, which is important for subsequent estimates.*

## E  THEORETICAL PROOF

### E.1  PROOF OF PROPOSITION 2.1

*Proof.* For a fixed data point $(\mathbf{x}, t)$, the derivative of the synthetic data $u_{\text{syn}}^D$ with respect to the flow parameter $\tau$ is given by the chain rule:

$$\frac{\partial u_{\text{syn}}^D(\mathbf{x}, t)}{\partial \tau} = \frac{\partial \mathcal{G}[m(\tau)](\mathbf{x}, t)}{\partial \tau} = \int_U \frac{\delta \mathcal{G}(\mathbf{x}, t)}{\delta m(\mathbf{y})}[m] \cdot \frac{\partial m(\mathbf{y})}{\partial \tau} d\mathbf{y}. \tag{27}$$

Substituting the gradient flow dynamics from equation 3 into equation 27 yields:

$$
\begin{aligned}
\frac{\partial u_{\text{syn}}^D(\mathbf{x}, t)}{\partial \tau} &= -\int_U \frac{\delta \mathcal{G}(\mathbf{x}, t)}{\delta m(\mathbf{y})}[m] \cdot \frac{\delta \mathcal{J}}{\delta m(\mathbf{y})}[m] dy \\
&= -\int_U \frac{\delta \mathcal{G}(\mathbf{x}, t)}{\delta m(\mathbf{y})}[m] \cdot \left[ \int_D \int_T \frac{\delta \mathcal{G}(\mathbf{x}', t')}{\delta m(\mathbf{y})}[m] \cdot (u_{\text{syn}}^D - u_{\text{obs}}^D) dt' d\mathbf{x}' \right] dy \\
&= -\int_D \int_T \left[ \int_U \frac{\delta \mathcal{G}(\mathbf{x}, t)}{\delta m(\mathbf{y})}[m] \frac{\delta \mathcal{G}(\mathbf{x}', t')}{\delta m(\mathbf{y})}[m] dy \right] \cdot (u_{\text{syn}}^D - u_{\text{obs}}^D) dt' d\mathbf{x}',
\end{aligned}
\tag{28}
$$

Here, the last step of the equation is due to Fubini's theorem, where we assume the integrand is absolutely integrable over $U \times D \times T$. We now define the *wave kernel* $\Theta$ as the inner product of the Fréchet derivatives:

$$
\Theta\left((\mathbf{x}, t), (\mathbf{x}', t'); m\right) \triangleq \int_U \frac{\delta \mathcal{G}(\mathbf{x}, t)}{\delta m(\mathbf{y})}[m] \cdot \frac{\delta \mathcal{G}(\mathbf{x}', t')}{\delta m(\mathbf{y})}[m] d\mathbf{y}.
\tag{29}
$$

This kernel is symmetric and characterizes the interaction between wavefield perturbations at different data points $(\mathbf{x}, t)$ and $(\mathbf{x}', t')$ due to model changes. $\qquad\square$

### E.2 PROOF OF PROPOSITION 3.1

*Proof.* For a fixed data point $(\mathbf{x}, t)$, the derivative of the synthetic data $u_{\text{syn}}^D(\mathbf{x}, t)$ with respect to the flow parameter $\tau$ is given by the chain rule:

$$
\frac{\partial u_{\text{syn}}^D(\mathbf{x}, t)}{\partial \tau} = \sum_{i=1}^{p} \frac{\partial \mathcal{G}[m_{\boldsymbol{\theta}}](\mathbf{x}, t)}{\partial \boldsymbol{\theta}_i} \cdot \frac{\partial \boldsymbol{\theta}_i}{\partial \tau}.
\tag{30}
$$

Substituting the gradient flow dynamics $\frac{\partial \boldsymbol{\theta}_i}{\partial \tau} = -\frac{\partial \mathcal{J}}{\partial \boldsymbol{\theta}_i}[\boldsymbol{\theta}]$ yields:

$$
\frac{\partial u_{\text{syn}}^D(\mathbf{x}, t)}{\partial \tau} = -\sum_{i=1}^{p} \frac{\partial \mathcal{G}[m_{\boldsymbol{\theta}}](\mathbf{x}, t)}{\partial \boldsymbol{\theta}_i} \cdot \frac{\partial \mathcal{J}}{\partial \boldsymbol{\theta}_i}[\boldsymbol{\theta}].
\tag{31}
$$

The derivative of the cost functional $\mathcal{J}$ is given by:

$$
\frac{\partial \mathcal{J}}{\partial \boldsymbol{\theta}_i}[\boldsymbol{\theta}] = \int_D \int_T \frac{\partial \mathcal{G}[m_{\boldsymbol{\theta}}](\mathbf{x}', t')}{\partial \boldsymbol{\theta}_i} \cdot \left( u_{\text{syn}}^D(\mathbf{x}', t') - u_{\text{obs}}^D(\mathbf{x}', t') \right) dt' d\mathbf{x}'.
\tag{32}
$$

Substituting this back, we obtain:

$$
\begin{aligned}
\frac{\partial d_{\text{syn}}^D(\mathbf{x}, t)}{\partial \tau} &= -\sum_{i=1}^{p} \frac{\partial \mathcal{G}[m_{\boldsymbol{\theta}}](\mathbf{x}, t)}{\partial \boldsymbol{\theta}_i} \cdot \left[ \int_D \int_T \frac{\partial \mathcal{G}[m_{\boldsymbol{\theta}}](\mathbf{x}', t')}{\partial \boldsymbol{\theta}_i} \cdot (u_{\text{syn}}^D - u_{\text{obs}}^D) dt' d\mathbf{x}' \right] \\
&= -\int_D \int_T \left[ \sum_{i=1}^{p} \frac{\partial \mathcal{G}[m_{\boldsymbol{\theta}}](\mathbf{x}, t)}{\partial \boldsymbol{\theta}_i} \cdot \frac{\partial \mathcal{G}[m_{\boldsymbol{\theta}}](\mathbf{x}', t')}{\partial \boldsymbol{\theta}_i} \right] \cdot (u_{\text{syn}}^D - u_{\text{obs}}^D) dt' d\mathbf{x}'.
\end{aligned}
\tag{33}
$$

The interchange of the sum and integrals is justified provided the sum converges absolutely. We now define the *wave-based NTK* as the inner product of the parameter gradients:

$$
\Theta_{\text{wave}}^{\text{ntk}}\left((\mathbf{x}, t), (\mathbf{x}', t'); \boldsymbol{\theta}\right) \triangleq \sum_{i=1}^{p} \frac{\partial \mathcal{G}[m_{\boldsymbol{\theta}}](\mathbf{x}, t)}{\partial \boldsymbol{\theta}_i} \cdot \frac{\partial \mathcal{G}[m_{\boldsymbol{\theta}}](\mathbf{x}', t')}{\partial \boldsymbol{\theta}_i}.
\tag{34}
$$

This kernel can be further expanded by applying the chain rule. The derivative with respect to a network parameter is related to the Fréchet derivative with respect to the model $m$:

$$
\frac{\partial \mathcal{G}(\mathbf{x}, t)}{\partial \boldsymbol{\theta}_i}[\boldsymbol{\theta}] = \int_U \frac{\delta \mathcal{G}(\mathbf{x}, t)}{\delta m(\mathbf{y})}[m_{\boldsymbol{\theta}}] \cdot \frac{\partial m_{\boldsymbol{\theta}}(\mathbf{y})}{\partial \boldsymbol{\theta}_i} d\mathbf{y}.
\tag{35}
$$

Substituting this form into the definition of $\Theta_{\text{wave}}^{\text{ntk}}$ gives:

$$
\begin{aligned}
\Theta_{\text{wave}}^{\text{ntk}} &= \sum_{i=1}^{p} \left( \int_U \frac{\delta \mathcal{G}(\mathbf{x}, t)}{\delta m(\mathbf{y})}[m_{\boldsymbol{\theta}}] \frac{\partial m_{\boldsymbol{\theta}}(\mathbf{y})}{\partial \boldsymbol{\theta}_i} d\mathbf{y} \right) \cdot \left( \int_U \frac{\delta \mathcal{G}(\mathbf{x}', t')}{\delta m(\mathbf{z})}[m_{\boldsymbol{\theta}}] \frac{\partial m_{\boldsymbol{\theta}}(\mathbf{z})}{\partial \boldsymbol{\theta}_i} d\mathbf{z} \right) \\
&= \int_U \int_U \frac{\delta \mathcal{G}(\mathbf{x}, t)}{\delta m(\mathbf{y})}[m_{\boldsymbol{\theta}}] \frac{\delta \mathcal{G}(\mathbf{x}', t')}{\delta m(\mathbf{z})}[m_{\boldsymbol{\theta}}] \cdot \left( \sum_{i=1}^{p} \frac{\partial m_{\boldsymbol{\theta}}(\mathbf{y})}{\partial \boldsymbol{\theta}_i} \frac{\partial m_{\boldsymbol{\theta}}(\mathbf{z})}{\partial \boldsymbol{\theta}_i} \right) d\mathbf{y} d\mathbf{z}.
\end{aligned}
\tag{36}
$$

The term in parentheses is recognized as the standard NTK for the network output $m_{\boldsymbol{\theta}}$:

$$K_{\tau}(\mathbf{y}, \mathbf{z}; \boldsymbol{\theta}) = \sum_{i=1}^{p} \frac{\partial m_{\boldsymbol{\theta}}(\mathbf{y})}{\partial \boldsymbol{\theta}_i} \frac{\partial m_{\boldsymbol{\theta}}(\mathbf{z})}{\partial \boldsymbol{\theta}_i}. \tag{37}$$

Therefore, the wave-based NTK is expressed as a double integral of the physical wave kernels coupled with the neural NTK:

$$\Theta_{\text{wave}}^{\text{ntk}}((\mathbf{x}, t), (\mathbf{x}', t'); \boldsymbol{\theta}) = \int_U \int_U \frac{\delta \mathcal{G}(\mathbf{x}, t)}{\delta m(\mathbf{y})}[m_{\boldsymbol{\theta}}] \frac{\delta \mathcal{G}(\mathbf{x}', t')}{\delta m(\mathbf{z})}[m_{\boldsymbol{\theta}}] \cdot K_{\tau}(\mathbf{y}, \mathbf{z}; \boldsymbol{\theta}) d\mathbf{y} d\mathbf{z}. \tag{38}$$

Substituting this back into the evolution equation concludes the proof. $\qquad\square$

### E.3 PROOF OF THEOREM 4.1

To establish the stochastic properties of the wave-based NTK both at initialization and during training, we first prove the convergence of the continuous representations under the infinite-width limit. We then demonstrate that the wave-based NTK is Lipschitz continuous, and employ the Continuous Mapping Theorem (Oksendal, 2013) to prove the convergence in distribution of the wave-based NTK. Accordingly, we begin by analyzing the initial distribution of the velocity model using a continuous representation under the infinite-width assumption.

**Lemma E.1 (Infinite-width Convergence of Neural Networks).** *Consider a fully-connected deep neural network $F_{\boldsymbol{\theta}}(\mathbf{x})$ as defined in equation 5, under Assumption D.1. Suppose the parameters $\boldsymbol{\theta}$ are initialized according to a proper scaling. Then, as the widths of all hidden layers sequentially or jointly, for any finite collection of inputs $\{\mathbf{x}_1, \mathbf{x}_2, \ldots, \mathbf{x}_k\} \subset U$, the output function $F_{\boldsymbol{\theta}}(\mathbf{x})$ converges in distribution to a zero-mean Gaussian process:*

$$(F_{\boldsymbol{\theta}}(\mathbf{x}_1), \ldots, F_{\boldsymbol{\theta}}(\mathbf{x}_k)) \xrightarrow{\mathcal{D}} \mathcal{N}(\mathbf{0}, \boldsymbol{\Sigma}),$$

*where the covariance matrix $\boldsymbol{\Sigma}$ has entries $\boldsymbol{\Sigma}_{ij} = \Sigma(\mathbf{x}_i, \mathbf{x}_j)$ given by a deterministic kernel function $\Sigma : U \times U \to \mathbb{R}$ that can be computed recursively through the network's layers. **Consequently**, the reparameterized velocity model $m_{\boldsymbol{\theta}}(\mathbf{x}) = m_0(\mathbf{x}) + F_{\boldsymbol{\theta}}(\mathbf{x})$ using a continuous representation converges in distribution to a Gaussian process $\mathcal{GP}(m_0(\mathbf{x}), \Sigma(\mathbf{x}, \mathbf{x}'))$.*

*Proof.* The proof relies on the application of the Central Limit Theorem (CLT) and induction over the network's layers. For a detailed proof, see (Bonfanti et al., 2024b) or (Jacot et al., 2018). Since $m_0(\mathbf{x})$ is a deterministic function, adding it to the zero-mean GP $F_{\boldsymbol{\theta}}$ results in a GP with mean function $m_0$ and the same covariance kernel $\Sigma$. $\qquad\square$

Next, we prove the continuity of the Green function and the sensitive kernel.

**Lemma E.2 (Lipschitz Continuity of the Green Function).** *Let $G_m$ denote the fundamental solution (Green function) associated with the wave operator $\mathcal{L}_m = \nabla^2 - m(\mathbf{x})\partial_t^2$. Under Assumption D.2, in particular given the lower bound $m_1 > 0$ and the separation $\delta > 0$ between the source region $U$ and the receiver region $D$, the mapping $m \mapsto G_m|_{D \times T}$ is Lipschitz continuous from $L^{\infty}(U)$ to $L^2(D \times T)$, i.e., there exists a constant $L_g = L_g(m_1, m_2, \delta, D, T) > 0$ such that:*

$$\|G_{m_1}(\cdot, \cdot; \mathbf{x}_0, t_0) - G_{m_2}(\cdot, \cdot; \mathbf{x}_0, t_0)\|_{L^2(D \times T)} \leq L_g \|m_1 - m_2\|_{L^{\infty}(U)},$$

*for any $m_1, m_2 \in \mathcal{A}$.*

*Proof.* Let $m_1, m_2 \in \mathcal{A}$ be two velocity models and define their difference as $\Delta m = m_1 - m_2$. Consider the difference of Green functions $w(\mathbf{x}, t; \mathbf{x}_0, t_0) = G_{m_1}(\mathbf{x}, t; \mathbf{x}_0, t_0) - G_{m_2}(\mathbf{x}, t; \mathbf{x}_0, t_0)$, which satisfies the following wave equation:

$$\mathcal{L}_{m_1} w(\mathbf{x}, t) = \Delta m(\mathbf{x}) \partial_t^2 G_{m_2}(\mathbf{x}, t; \mathbf{x}_0, t_0). \tag{39}$$

We apply energy estimates to the wave equation satisfied by $w$. Define the energy functional:

$$E(t) = \frac{1}{2} \int_{\mathbb{R}^d} \left[ m_1(\mathbf{x}) |\partial_t w(\mathbf{x}, t)|^2 + |\nabla w(\mathbf{x}, t)|^2 \right] d\mathbf{x}. \tag{40}$$

Differentiating with respect to time and using the wave equation:

$$\begin{aligned}
\frac{dE}{dt} &= \int_{\mathbb{R}^d} \left[ m_1(\mathbf{x})\partial_t w \partial_t^2 w + \nabla w \cdot \nabla \partial_t w \right] d\mathbf{x} \\
&= \int_{\mathbb{R}^d} \left[ \partial_t w (m_1 \partial_t^2 w - \nabla^2 w) + \nabla \cdot (\partial_t w \nabla w) \right] d\mathbf{x} \\
&= \int_{\mathbb{R}^d} \left[ -\partial_t w \cdot \mathcal{L}_{m_1} w + \nabla \cdot (\partial_t w \nabla w) \right] d\mathbf{x} \\
&= \int_{\mathbb{R}^d} \left[ -\partial_t w \cdot (\Delta m \partial_t^2 G_{m_2}) + \nabla \cdot (\partial_t w \nabla w) \right] d\mathbf{x}.
\end{aligned} \tag{41}$$

Integrating over space $\mathbb{R}^d$ and applying the divergence theorem (the boundary term vanishes due to finite propagation speed), we have:

$$\begin{aligned}
\frac{dE}{dt} &= \int_{\mathbb{R}^d} \left[ -\partial_t w \cdot (\Delta m \partial_t^2 G_{m_2}) + \nabla \cdot (\partial_t w \nabla w) \right] d\mathbf{x} \\
&= -\int_{\mathbb{R}^d} \partial_t w \cdot (\Delta m \partial_t^2 G_{m_2}) d\mathbf{x} + \oint_{\partial \mathbb{R}^d} (\partial_t w \nabla w) \cdot \mathbf{n} d\mathbf{s} \\
&= -\int_{\mathbb{R}^d} \partial_t w \cdot (\Delta m \partial_t^2 G_{m_2}) d\mathbf{x}.
\end{aligned} \tag{42}$$

Using the boundedness of $\Delta m$ and the Cauchy-Schwarz inequality:

$$\begin{aligned}
\left| \frac{dE}{dt} \right| &\leq \|\Delta m\|_{L^\infty(U)} \int_{\mathbb{R}^d} |\partial_t w||\partial_t^2 G_{m_2}| d\mathbf{x} \\
&\leq \|\Delta m\|_{L^\infty(U)} \left( \int_{\mathbb{R}^d} |\partial_t w|^2 d\mathbf{x} \right)^{1/2} \left( \int_{\mathbb{R}^d} |\partial_t^2 G_{m_2}|^2 d\mathbf{x} \right)^{1/2} \\
&\leq \|\Delta m\|_{L^\infty(U)} \sqrt{\frac{2}{m_1}} \sqrt{E(t)} \|\partial_t^2 G_{m_2}(\cdot, t; \mathbf{x}_0, t_0)\|_{L^2(\mathbb{R}^d)},
\end{aligned} \tag{43}$$

where the last inequality holds due to the energy definition and the lower bound $m_1 > 0$, i.e.,

$$\int_{\mathbb{R}^d} |\partial_t w|^2 d\mathbf{x} \leq \frac{2}{m_1} E(t). \tag{44}$$

This gives us the differential inequality:

$$\frac{d}{dt} \sqrt{E(t)} \leq \frac{1}{\sqrt{2m_1}} \|\Delta m\|_{L^\infty(U)} \|\partial_t^2 G_{m_2}(\cdot, t; \mathbf{x}_0, t_0)\|_{L^2(\mathbb{R}^d)}. \tag{45}$$

Integrating from $t_0$ to $t$:

$$\sqrt{E(t)} \leq \frac{1}{\sqrt{2m_1}} \|\Delta m\|_{L^\infty(U)} \int_{t_0}^t \|\partial_t^2 G_{m_2}(\cdot, s; \mathbf{x}_0, t_0)\|_{L^2(\mathbb{R}^d)} ds. \tag{46}$$

From the theory of hyperbolic equations and under Assumption D.2, we have the regularity results for the Green function $G_{m_2}$. For a fixed source point $(\mathbf{x}_0, t_0)$ and away from the singularity, $G_{m_2}$ enjoys higher regularity. Specifically, for $\mathbf{x} \in D$ where $\text{dist}(D, U) \geq \delta > 0$, we have:

$$\|\partial_t^2 G_{m_2}(\cdot, t; \mathbf{x}_0, t_0)\|_{L^2(D)} \leq C_1(\delta, m_1, m_2, T_0). \tag{47}$$

The energy of the wavefield decays sufficiently fast away from the source due to finite propagation speed and geometric optics approximations. This ensures that:

$$\int_{t_0}^t \|\partial_t^2 G_{m_2}(\cdot, s; \mathbf{x}_0, t_0)\|_{L^2(\mathbb{R}^d)} ds \leq C_2(m_1, m_2, T_0). \tag{48}$$

Combining the previous estimates and using Poincaré-type inequalities, we obtain:

$$
\begin{aligned}
\|w(\cdot,t;\mathbf{x}_0,t_0)\|_{L^2(D)} &= \left\| \int_{t_0}^t \partial_\tau w(\mathbf{x},\tau)d\tau \right\|_{L^2(D)} \\
&\leq \int_{t_0}^t \|\partial_\tau w(\cdot,\tau)\|_{L^2(D)}d\tau \\
&\leq \int_{t_0}^t \sqrt{\frac{2}{m_1}} \cdot \sqrt{E(\tau)}d\tau \triangleq C_3(D,T_0) \cdot \sqrt{E(t)} \\
&\leq C_3(D,T) \cdot \frac{1}{\sqrt{2m_1}}\|\Delta m\|_{L^\infty(D)}C_2(m_1,m_2,T_0).
\end{aligned}
\tag{49}
$$

Integrating over time $t \in [0, T_0]$:

$$
\begin{aligned}
\|w\|_{L^2(D\times T)} &\leq T_0^{1/2} \sup_{t\in T} \|w(\cdot,t;\mathbf{x}_0,t_0)\|_{L^2(D)} \\
&\leq T_0^{1/2}C_3(D,T_0)\frac{1}{\sqrt{2m_1}}C_2(m_1,m_2,T_0)\|\Delta m\|_{L^\infty(U)} \\
&\triangleq L_g\|m_1 - m_2\|_{L^\infty(U)},
\end{aligned}
\tag{50}
$$

where $L_g = \frac{T_0^{1/2}C_3C_2}{\sqrt{2m_1}}$ depends on $m_1, m_2, \delta, D, T_0$. This completes the proof. $\square$

**Lemma E.3** (**Lipschitz Continuity of the Sensitivity Kernel**). *Consider the acoustic wave equation equation 1 under Assumption D.2. The Fréchet derivative $\frac{\delta\mathcal{G}(\mathbf{x},t)}{\delta m(\mathbf{y})}[m]$ is Lipschitz continuous with respect to the velocity model $m$. That is, there exists a constant $L = L(m_1, m_2, \delta, D, T) > 0$ such that for any $m_1, m_2 \in \mathcal{A}$:*

$$
\left\| \frac{\delta\mathcal{G}(\mathbf{x},t)}{\delta m(\mathbf{y})}[m_1] - \frac{\delta\mathcal{G}(\mathbf{x},t)}{\delta m(\mathbf{y})}[m_2] \right\|_{L^2(D\times T)} \leq L\|m_1 - m_2\|_{L^\infty(U)}.
$$

*Proof.* The Fréchet derivative of the wavefield $u_j$ w.r.t. the model parameter $m$ is given by:

$$
\frac{\delta u_j(\mathbf{x},t)}{\delta m(\mathbf{y})}[m] = -\int_0^t G_m(\mathbf{x},t;\mathbf{y},t')\partial_{t'}^2 u_j(\mathbf{y},t')dt',
\tag{51}
$$

where $G_m$ is the Green function for the wave operator. For $m_1, m_2 \in \mathcal{A}$, consider the difference:

$$
\begin{aligned}
&\frac{\delta u_j}{\delta m}[m_1] - \frac{\delta u_j}{\delta m}[m_2] \\
&= -\int_0^t \left[ G_{m_1}(\mathbf{x},t;\mathbf{y},t')\partial_{t'}^2 u_j(m_1)(\mathbf{y},t') - G_{m_2}(\mathbf{x},t;\mathbf{y},t')\partial_{t'}^2 u_j(m_2)(\mathbf{y},t') \right]dt' \\
&= -\int_0^t (G_{m_1} - G_{m_2})\partial_{t'}^2 u_j(m_1)dt' - \int_0^t G_{m_2}\left(\partial_{t'}^2 u_j(m_1) - \partial_{t'}^2 u_j(m_2)\right)dt' \\
&\triangleq A_j + B_j.
\end{aligned}
\tag{52}
$$

Using the Cauchy-Schwarz inequality and the properties of the wave equation:

$$
\begin{aligned}
\|A_j\|_{L^2(D\times T)} &\leq \int_0^T \left\|(G_{m_1} - G_{m_2})\partial_{t'}^2 u_j(m_1)\right\|_{L^2(D)}dt' \\
&\leq \int_0^T \|G_{m_1} - G_{m_2}\|_{L^2(D)}\|\partial_{t'}^2 u_j(m_1)\|_{L^\infty(U)}dt'.
\end{aligned}
\tag{53}
$$

From Lemma E.2, we have:

$$
\|G_{m_1} - G_{m_2}\|_{L^2(D)} \leq L_g\|m_1 - m_2\|_{L^\infty(U)}.
\tag{54}
$$

From Proposition D.1, we have:

$$
\|\partial_{t'}^2 u_j(m_1)\|_{L^\infty(U)} \leq C_1\|s_j\|_{H^1((0,T),L^2(U))}.
\tag{55}
$$

Therefore:

$$\|A_j\|_{L^2(D\times T)} \le TL_gC_1\|s_j\|_{H^1((0,T),L^2(U))}\|m_1 - m_2\|_{L^\infty(U)}. \tag{56}$$

**Similarly:**

$$\|B_j\|_{L^2(D\times T)} \le \int_0^T \|G_{m_2}\|_{L^2(D)}\|\partial_{t'}^2 u_j(m_1) - \partial_{t'}^2 u_j(m_2)\|_{L^\infty(U)}dt' \tag{57}$$

$$\le \int_0^T C_2\|\partial_{t'}^2 u_j(m_1) - \partial_{t'}^2 u_j(m_2)\|_{L^\infty(U)}dt',$$

where $C_2$ is a bound on $\|G_{m_2}\|_{L^2(D)}$ which exists due to the regularity of the Green function away from the source. To estimate $\|\partial_{t'}^2 u_j(m_1) - \partial_{t'}^2 u_j(m_2)\|_{L^\infty(U)}$, we consider the difference $v_j = u_j(m_1) - u_j(m_2)$, which satisfies:

$$\mathcal{L}_{m_1} v_j = (m_1 - m_2)\partial_t^2 u_j(m_2). \tag{58}$$

Using energy estimates for hyperbolic equations and the regularity from Proposition D.1, we obtain:

$$\|\partial_{t'}^2 v_j\|_{L^\infty(U)} \le C_3\|m_1 - m_2\|_{L^\infty(U)}\|\partial_t^2 u_j(m_2)\|_{L^\infty(U)}. \tag{59}$$

Therefore:

$$\|B_j\|_{L^2(D\times T)} \le TC_2C_3C_4\|s_j\|_{H^1((0,T),L^2(U))}\|m_1 - m_2\|_{L^\infty(U)}, \tag{60}$$

where $C_4$ is a bound for $\|\partial_t^2 u_j(m_2)\|_{L^\infty(U)}$. Combining the estimates for $A_j$ and $B_j$, we obtain:

$$\left\|\frac{\delta u_j}{\delta m}[m_1] - \frac{\delta u_j}{\delta m}[m_2]\right\|_{L^2(D\times T)} \le L\|m_1 - m_2\|_{L^\infty(U)}, \tag{61}$$

where $L = T(L_gC_1 + C_2C_3C_4)\|s_j\|_{H^1((0,T),L^2(U))}$. This completes the proof. $\square$

Then, we focus on the convergence of the initial sensitive kernel as the width tends to infinity.

**Proposition E.1** (**Sensitivity Kernel Convergence in Distribution**). *Consider a fully-connected neural network $F_{\boldsymbol{\theta}}$ as defined in equation 5, under Assumptions D.1 and D.2. Suppose the parameters $\boldsymbol{\theta}$ are initialized with appropriate scaling. Then, as the width $n \to \infty$, the finite-dimensional distributions of the sensitivity kernel converge:*

$$\left(\frac{\delta\mathcal{G}(\mathbf{x}_1,t_1)}{\delta m(\mathbf{y}_1)}[m_{\boldsymbol{\theta}}],\ldots,\frac{\delta\mathcal{G}(\mathbf{x}_k,t_k)}{\delta m(\mathbf{y}_k)}[m_{\boldsymbol{\theta}}]\right) \xrightarrow{\mathcal{D}} \left(\frac{\delta\mathcal{G}(\mathbf{x}_1,t_1)}{\delta m(\mathbf{y}_1)}[\mathcal{GP}],\ldots,\frac{\delta\mathcal{G}(\mathbf{x}_k,t_k)}{\delta m(\mathbf{y}_k)}[\mathcal{GP}]\right)$$

*For any finite collection of points $(\mathbf{x}_i,t_i,\mathbf{y}_i) \in D \times T \times U$, $i = 1,\ldots,k$, where $\mathcal{GP}$ denotes the Gaussian process $\mathcal{GP}(m_0,\Sigma)$ from Lemma E.1.*

*Proof.* By Lemma E.1, as the width $n \to \infty$, the finite-dimensional distributions of $m_{\boldsymbol{\theta}}$ converge to those of $\mathcal{GP}(m_0,\Sigma)$. That is, for any finite set of points $\{\mathbf{x}_1,\ldots,\mathbf{x}_k\} \subset U$,

$$(m_{\boldsymbol{\theta}}(\mathbf{x}_1),\ldots,m_{\boldsymbol{\theta}}(\mathbf{x}_k)) \xrightarrow{\mathcal{D}} \mathcal{N}(\boldsymbol{\mu},\boldsymbol{\Sigma}), \tag{62}$$

where $\boldsymbol{\mu} = (m_0(\mathbf{x}_1),\ldots,m_0(\mathbf{x}_k))$ and $\boldsymbol{\Sigma}_{ij} = \Sigma(\mathbf{x}_i,\mathbf{x}_j)$. Furthermore, by Lemma E.3, the Fréchet derivative operator

$$\Phi : m \mapsto \frac{\delta\mathcal{G}(\mathbf{x},t)}{\delta m(\mathbf{y})}[m] \tag{63}$$

Is Lipschitz continuous from $(L^\infty(U), \|\cdot\|_{L^\infty})$ to $(L^2(D\times T), \|\cdot\|_{L^2})$. In particular, for any fixed finite collection of points $(\mathbf{x}_i,t_i,\mathbf{y}_i)$, the mapping

$$m \mapsto \left(\frac{\delta\mathcal{G}(\mathbf{x}_1,t_1)}{\delta m(\mathbf{y}_1)}[m],\ldots,\frac{\delta\mathcal{G}(\mathbf{x}_k,t_k)}{\delta m(\mathbf{y}_k)}[m]\right) \tag{64}$$

is continuous from $(L^\infty(U), \|\cdot\|_{L^\infty})$ to $\mathbb{R}^k$. Since $m_{\boldsymbol{\theta}}$ converges to $\mathcal{GP}(m_0,\Sigma)$ in finite-dimensional distributions, and the mapping $\Phi$ is continuous, by the continuous mapping theorem for finite-dimensional distributions (Billingsley, 2013), we have:

$$\left(\frac{\delta\mathcal{G}(\mathbf{x}_1,t_1)}{\delta m(\mathbf{y}_1)}[m_{\boldsymbol{\theta}}],\ldots,\frac{\delta\mathcal{G}(\mathbf{x}_k,t_k)}{\delta m(\mathbf{y}_k)}[m_{\boldsymbol{\theta}}]\right) \xrightarrow{\mathcal{D}} \left(\frac{\delta\mathcal{G}(\mathbf{x}_1,t_1)}{\delta m(\mathbf{y}_1)}[\mathcal{GP}],\ldots,\frac{\delta\mathcal{G}(\mathbf{x}_k,t_k)}{\delta m(\mathbf{y}_k)}[\mathcal{GP}]\right). \tag{65}$$

This completes the proof. $\square$

Next, we begin to prove the conclusions (I) and (II) in Theorem 4.1.

### E.3.1 PROOF OF THEOREM 4.1 (I)

*Proof.* Recall the definition of the wave-based neural tangent kernel from Proposition E.2:

$$\Theta_{\text{wave}}^{\text{ntk}}\big((\mathbf{x},t),(\mathbf{x}',t');\boldsymbol{\theta}\big) = \int_U \int_U \frac{\delta\mathcal{G}(\mathbf{x},t)}{\delta m(\mathbf{y})}[m] \cdot \frac{\delta\mathcal{G}(\mathbf{x}',t')}{\delta m(\mathbf{z})}[m] \cdot K_\tau(\mathbf{y},\mathbf{z};\boldsymbol{\theta})dyd\mathbf{z}. \quad (66)$$

By Lemma 4.1, the standard NTK $K_\tau(\mathbf{y},\mathbf{z};\boldsymbol{\theta})$ converges in probability to a deterministic limiting kernel, i.e., $K_\tau(\mathbf{y},\mathbf{z};\boldsymbol{\theta}) \xrightarrow{\mathbb{P}} \widetilde{K}(\mathbf{y},\mathbf{z})$ as $n \to \infty$. Furthermore, by Proposition E.1, the following sensitivity kernels converge in distribution to the following random process as $n \to \infty$, i.e.,

$$\frac{\delta\mathcal{G}(\mathbf{x},t)}{\delta m(\mathbf{y})}[m_{\boldsymbol{\theta}}] \xrightarrow{\mathcal{D}} \frac{\delta\mathcal{G}(\mathbf{x},t)}{\delta m(\mathbf{y})}[\mathcal{GP}(m_0,\Sigma)] \quad \text{and} \quad \frac{\delta\mathcal{G}(\mathbf{x}',t')}{\delta m(\mathbf{z})}[m_{\boldsymbol{\theta}}] \xrightarrow{\mathcal{D}} \frac{\delta\mathcal{G}(\mathbf{x}',t')}{\delta m(\mathbf{y})}[\mathcal{GP}(m_0,\Sigma)] \quad (67)$$

Applying Slutsky's theorem, we conclude that for each fixed $(\mathbf{y},\mathbf{z})$, the following product:

$$\frac{\delta\mathcal{G}(\mathbf{x},t)}{\delta m(\mathbf{y})}[m_{\boldsymbol{\theta}}] \cdot \frac{\delta\mathcal{G}(\mathbf{x}',t')}{\delta m(\mathbf{z})}[m_{\boldsymbol{\theta}}] \cdot K_\tau(\mathbf{y},\mathbf{z};\boldsymbol{\theta}) \quad (68)$$

converges in distribution to the following random product under the infinite-width condition:

$$\frac{\delta\mathcal{G}(\mathbf{x},t)}{\delta m(\mathbf{y})}[\mathcal{GP}(m_0,\Sigma)] \cdot \frac{\delta\mathcal{G}(\mathbf{x}',t')}{\delta m(\mathbf{z})}[\mathcal{GP}(m_0,\Sigma)] \cdot \widetilde{K}(\mathbf{y},\mathbf{z}). \quad (69)$$

Since this convergence holds point-wise in $(\mathbf{y},\mathbf{z})$, and the integral is a continuous mapping, we apply the continuous mapping theorem to conclude:

$$\Theta_{\text{wave}}^{\text{ntk}}\big((\mathbf{x},t),(\mathbf{x}',t');\boldsymbol{\theta}\big) \xrightarrow{\mathcal{D}} \int_U \int_U \frac{\delta\mathcal{G}(\mathbf{x},t)}{\delta m(\mathbf{y})}[\mathcal{GP}] \cdot \frac{\delta\mathcal{G}(\mathbf{x}',t')}{\delta m(\mathbf{z})}[\mathcal{GP}] \cdot \widetilde{K}(\mathbf{y},\mathbf{z})dyd\mathbf{z}. \quad (70)$$

The limiting expression is a random variable for each fixed input pair $(\mathbf{x},t,\mathbf{x}',t')$, which we denote by $\widehat{\Theta_{\text{wave}}^{\text{ntk}}}\big((\mathbf{x},t),(\mathbf{x}',t')\big)$. This kernel is non-deterministic due to the randomness in the Gaussian process realizations of the sensitivity kernels. Therefore, we have established the point-wise convergence in distribution, i.e.,

$$\Theta_{\text{wave}}^{\text{ntk}}\big|_{\tau=0}^n \xrightarrow{\mathcal{D}} \widehat{\Theta_{\text{wave}}^{\text{ntk}}} \quad \text{as } n \to \infty, \quad (71)$$

where $\widehat{\Theta_{\text{wave}}^{\text{ntk}}}$ is a random kernel. □

### E.3.2 PROOF OF THEOREM 4.1 (II)

*Proof.* By Assumption D.1 and the properties of gradient flow, we assume that there exists a sequence $\tau_n \to \infty$ such that $m_{\boldsymbol{\theta}(\tau_n)} \to m^*$ almost surely as $n \to \infty$, where $m^*$ is a minimizer of the loss function. Define the sensitivity kernel operator for fixed $(\mathbf{x},t,\mathbf{y})$ as:

$$\Lambda(m) = \frac{\delta\mathcal{G}(\mathbf{x},t)}{\delta m(\mathbf{y})}[m]. \quad (72)$$

By Proposition E.1, $\Lambda$ is Lipschitz continuous w.r.t the $L^\infty$ norm. Thus, since $m_{\boldsymbol{\theta}(\tau_n)} \to m^*$ almost surely, we have $\Lambda(m_{\boldsymbol{\theta}(\tau_n)}) \to \Lambda(m^*)$ almost surely for each fixed $(\mathbf{x},t,\mathbf{y})$. The wave-based NTK at time $\tau$ and width $n$ is defined as:

$$\Theta_{\text{wave}}^{\text{ntk}}\big|_\tau^n = \int_U \int_U \Lambda(m_{\boldsymbol{\theta}(\tau)})(\mathbf{x},t,\mathbf{y}) \cdot K_\tau(\mathbf{y},\mathbf{z}) \cdot \Lambda(m_{\boldsymbol{\theta}(\tau)})(\mathbf{x}',t',\mathbf{z}) \, d\mathbf{y}\, d\mathbf{z}, \quad (73)$$

where $K_\tau$ is the standard NTK for the model output. By Theorem 4.1, at initialization, the standard NTK $K_0$ converges uniformly to a deterministic kernel $\widetilde{K}$ as the width tends to infinity. Moreover, in the infinite-width limit, the standard NTK remains constant during training, so $K_\tau \to \widetilde{K}$ for all $\tau \geq 0$. Hence, the infinite-width limit of the wave-based NTK at initialization is:

$$\Theta_{\text{wave}}^{\text{ntk}}\big|_0^\infty = \int_U \int_U \Lambda(m_{\boldsymbol{\theta}(0)})(\mathbf{x},t,\mathbf{y}) \cdot \widetilde{K}(\mathbf{y},\mathbf{z}) \cdot \Lambda(m_{\boldsymbol{\theta}(0)})(\mathbf{x}',t',\mathbf{z}) \, d\mathbf{y}\, d\mathbf{z}. \quad (74)$$

Now, consider the difference in the sequence $\tau_n$, and apply the triangle inequality, we have:

$$\left| \Theta_{\text{wave}}^{\text{ntk}} \big|_{\tau_n}^n - \Theta_{\text{wave}}^{\text{ntk}} \big|_0^\infty \right| \leq I_1 + I_2, \tag{75}$$

where

$$I_1 = \left| \iint_U \int_U \Lambda(m_{\boldsymbol{\theta}(\tau_n)}) \cdot K_{\tau_n} \cdot \Lambda(m_{\boldsymbol{\theta}(\tau_n)})' - \Lambda(m_{\boldsymbol{\theta}(\tau_n)}) \cdot \widetilde{K} \cdot \Lambda(m_{\boldsymbol{\theta}(\tau_n)})' \, d\mathbf{y} \, d\mathbf{z} \right|, \tag{76}$$

$$I_2 = \left| \iint_U \int_U \Lambda(m_{\boldsymbol{\theta}(\tau_n)}) \cdot \widetilde{K} \cdot \Lambda(m_{\boldsymbol{\theta}(\tau_n)})' - \Lambda(m_{\boldsymbol{\theta}(0)}) \cdot \widetilde{K} \cdot \Lambda(m_{\boldsymbol{\theta}(0)})' \, d\mathbf{y} \, d\mathbf{z} \right|, \tag{77}$$

and we use shorthand $\Lambda(m_{\boldsymbol{\theta}(\tau_n)})'$ for $\Lambda(m_{\boldsymbol{\theta}(\tau_n)})(\mathbf{x}', t', \mathbf{z})$. For $I_1$, since $K_{\tau_n} \to \widetilde{K}$ (by Theorem 4.1), and $\Lambda(m_{\boldsymbol{\theta}(\tau_n)})$ is uniformly bounded (by Lemma E.3 and the boundedness of $\mathcal{A}$), we have:

$$I_1 \leq \|\Lambda(m_{\boldsymbol{\theta}(\tau_n)})\|_\infty^2 \cdot \|K_{\tau_n}^n - \widetilde{K}\|_\infty \cdot |U|^2 \to 0 \quad \text{almost surely as } n \to \infty. \tag{78}$$

For $I_2$, define the function $f : \mathcal{A} \to \mathbb{R}$ for fixed $(\mathbf{x}, t, \mathbf{x}', t')$ by:

$$f(m) = \int_U \int_U \Lambda(m)(\mathbf{x}, t, \mathbf{y}) \cdot \widetilde{K}(\mathbf{y}, \mathbf{z}) \cdot \Lambda(m)(\mathbf{x}', t', \mathbf{z}) \, d\mathbf{y} \, d\mathbf{z}. \tag{79}$$

By the continuity of $\Lambda$ and the boundedness of $U$, $f$ is continuous on $\mathcal{A}$. Then:

$$I_2 = |f(m_{\boldsymbol{\theta}(\tau_n)}) - f(m_{\boldsymbol{\theta}(0)})|. \tag{80}$$

Since $m_{\boldsymbol{\theta}(\tau_n)} \to m^*$ almost surely, we have $f(m_{\boldsymbol{\theta}(\tau_n)}) \to f(m^*)$ almost surely. At initialization, $m_{\boldsymbol{\theta}(0)}$ is random with a non-degenerate Gaussian distribution (by the initialization assumption and the wide network theory), and $f$ is not constant on $\mathcal{A}$ (because the sensitivity kernel depends non-trivially on $m$). Therefore, the set $\{m \in \mathcal{A} : f(m) = f(m^*)\}$ has measure zero. Hence, almost surely, $f(m_{\boldsymbol{\theta}(0)}) \neq f(m^*)$, and thus for large $n$, $|f(m_{\boldsymbol{\theta}(\tau_n)}) - f(m_{\boldsymbol{\theta}(0)})| > 0$. Then, combining the estimates for $I_1$ and $I_2$, we conclude that almost surely:

$$\liminf_{n \to \infty} \left| \Theta_{\text{wave}}^{\text{ntk}} \big|_{\tau_n}^n - \Theta_{\text{wave}}^{\text{ntk}} \big|_0^\infty \right| > 0. \tag{81}$$

This implies that for large $n$, the supremum over $\tau$ satisfies:

$$\sup_{\tau \geq 0} \left| \Theta_{\text{wave}}^{\text{ntk}} \big|_\tau^n - \Theta_{\text{wave}}^{\text{ntk}} \big|_0^\infty \right| > 0, \tag{82}$$

since the value at $\tau_n$ is positive. Therefore, almost surely:

$$\liminf_{n \to \infty} \sup_{\tau \geq 0} \left| \Theta_{\text{wave}}^{\text{ntk}} \big|_\tau^n - \Theta_{\text{wave}}^{\text{ntk}} \big|_0^\infty \right| > 0. \tag{83}$$

This completes the proof. $\qquad\square$

### E.4 PROOF OF THE EIGENVALUE DECAY COMPARISON THEOREM

The spectral properties of the wave kernel and wave-based NTK are fundamental to understanding the convergence behavior and robustness in FWI and CR-FWI. To analyze the eigenvalue of the wave kernel and the wave-based NTK, we define the Fréchet derivative integral operator $\mathcal{F}_m : L^2(U) \to L^2(D \times T)$ and its adjoint $\mathcal{F}_m^* : L^2(U) \to L^2(D \times T)$ as follows:

$$\mathcal{F}_m[h](\mathbf{x}, t) = \int_U \frac{\delta \mathcal{G}(x, t)}{\delta m(\mathbf{y})}[m]h(\mathbf{y})d\mathbf{y}, \quad \mathcal{F}_m^*[g](\mathbf{y}) = \int_D \int_T \frac{\delta \mathcal{G}(\mathbf{x}, t)}{\delta m(\mathbf{y})}[m]g(\mathbf{x}, t)dtd\mathbf{x}. \tag{84}$$

Moreover, we define the kernel integral operator $K : L^2(U) \to L^2(U)$ as

$$K[h](\mathbf{y}) = \int_U K(\mathbf{y}, \mathbf{z})h(\mathbf{z})d\mathbf{z}. \tag{85}$$

Hence, the wavefield evolution equation of CR-FWI can be expressed as follows:

$$\frac{\partial d_{\text{syn}}^D(\mathbf{x}, t)}{\partial \tau} = \mathcal{F}_m \circ K \circ \mathcal{F}_m^*[d_{\text{obs}}^D - d_{\text{syn}}^D] \triangleq \mathcal{T}(K)[d_{\text{obs}}^D - d_{\text{syn}}^D]. \tag{86}$$

where $\mathcal{T}(K) \triangleq \mathcal{F}_m \circ K \circ \mathcal{F}_m^*$ denotes the integral operator of wave-based NTK.

Next, we compare the eigenvalue for two operators of wave-based NTK, i.e., $\mathcal{T}(K_1)$ and $\mathcal{T}(K_2)$.

**Theorem E.1** (**Spectral Comparison Theorem of CR-FWI**). *For two self-adjoint and positive semi-definite operators $K_1, K_2 : L^2(U) \to L^2(U)$, define the wave-based NTK operators as:*

$$\mathcal{T}_i \triangleq \mathcal{F}_m K_i \mathcal{F}_m^*, \quad i = 1, 2. \tag{87}$$

*Let $\lambda_j(\mathcal{T}_i)_{j=1}^{\infty}$ denote the eigenvalues of $\mathcal{T}_i$ arranged in non-increasing order. If the operators satisfy the dominance condition $K_1 \succeq K_2$ (i.e., $K_1 - K_2$ is a positive semi-definite operator), then the eigenvalues of the corresponding wave-based NTK operators are similarly ordered:*

$$\lambda_j(\mathcal{T}_1) \geq \lambda_j(\mathcal{T}_2) \quad \text{for all } j = 1, 2, \ldots. \tag{88}$$

*Proof.* The proof follows directly from the min-max principle and the properties of the operators. Since $K_1 \succeq K_2$, we have that $K_1 - K_2$ is positive semi-definite. This means that for any $g \in L^2(U)$:

$$\langle K_1 g, g \rangle_{L^2(U)} \geq \langle K_2 g, g \rangle_{L^2(U)}. \tag{89}$$

Now, for any $f \in L^2(D \times T)$, let $g = \mathcal{F}_m^* f \in L^2(U)$. Then the quadratic forms of the wave-based NTK operators satisfy:

$$\begin{aligned}
\langle \mathcal{T}_1 f, f \rangle_{L^2(D \times T)} &= \langle \mathcal{F}_m K_1 \mathcal{F}_m^* f, f \rangle_{L^2(D \times T)} \\
&= \langle K_1 \mathcal{F}_m^* f, \mathcal{F}_m^* f \rangle_{L^2(U)} \\
&\geq \langle K_2 \mathcal{F}_m^* f, \mathcal{F}_m^* f \rangle_{L^2(U)} \\
&= \langle \mathcal{F}_m K_2 \mathcal{F}_m^* f, f \rangle_{L^2(D \times T)} \\
&= \langle \mathcal{T}_2 f, f \rangle_{L^2(D \times T)}.
\end{aligned} \tag{90}$$

Thus, we have established that:

$$\langle \mathcal{T}1 f, f \rangle \geq \langle \mathcal{T}2 f, f \rangle \quad \text{for all } f \in L^2(D \times T). \tag{91}$$

By the Courant-Fischer (min-max) theorem, for each $j = 1, 2, \ldots$:

$$\begin{aligned}
\lambda_j(\mathcal{T}1) &= \min_{\substack{V \subset L^2(D \times T) \\ \dim(V) = j}} \max_{\substack{f \in V \\ |f| = 1}} \langle \mathcal{T}_1 f, f \rangle \\
&\geq \min_{\substack{V \subset L^2(D \times T) \\ \dim(V) = j}} \max_{\substack{f \in V \\ |f| = 1}} \langle \mathcal{T}_2 f, f \rangle \\
&= \lambda_j(\mathcal{T}_2).
\end{aligned} \tag{92}$$

The inequality follows because for any fixed $j$-dimensional subspace $V$, we have:

$$\max_{f \in V \ |f| = 1} \langle \mathcal{T}_1 f, f \rangle \geq \max_{f \in V \ |f| = 1} \langle \mathcal{T}_2 f, f \rangle, \tag{93}$$

and taking the minimum over all such $V$ preserves the inequality. $\square$

**Remark E.1.** *According to the Spectral Comparison Theorem (Theorem E.1), if two self-adjoint positive semi-definite kernel operators satisfy $K_1 \succeq K_2$, then the eigenvalues of the corresponding wave-based neural tangent kernel operators $\mathcal{T}_1$ and $\mathcal{T}_2$ satisfy $\lambda_j(\mathcal{T}_1) \geq \lambda_j(\mathcal{T}_2)$. In particular, both the wave kernel $\Theta_{wave}$ defined in Proposition 2.1 and the wave neural tangent kernel $\Theta_{wave}^{ntk}$ defined in Proposition 3.1 can be expressed in the form $\mathcal{T}(K)$ as in the theorem, where $K$ corresponds to the identity operator $K_\delta$ (see the following Proposition E.2) and the neural network's NTK kernel $K_\tau$, respectively. Therefore, if the NTK kernels of two neural networks satisfy $K_{\tau,1} \succeq K_{\tau,2}$, then the eigenvalues of their wave neural tangent kernels will also satisfy the same ordering relation.*

To apply Theorem E.1, we need to rewrite the formulation of the wave kernel and prove the self-adjoint and positive semi-definite of wave-based NTK.

**Proposition E.2.** *The wave-based NTK $\Theta_{wave}^{ntk}$ defined in Proposition 3.1 is symmetric and semi-positive definite, under the assumption that NTK $K_\tau$ is a symmetric and semi-positive definite kernel.*

*Proof.* We aim to prove that the wave-based neural NTK defined by

$$\Theta_{\text{wave}}^{\text{ntk}}\big((\mathbf{x}, t), (\mathbf{x}', t'); \boldsymbol{\theta}\big) = \int_U \int_U \frac{\delta \mathcal{G}(\mathbf{x}, t)}{\delta m(\mathbf{y})}[m] \cdot \frac{\delta \mathcal{G}(\mathbf{x}', t')}{\delta m(\mathbf{z})}[m] \cdot K_\tau(\mathbf{y}, \mathbf{z}; \boldsymbol{\theta}) dy d\mathbf{z} \tag{94}$$

**Symmetry.** To show symmetry, we observe that the expression for $\Theta_{\text{wave}}^{\text{ntk}}$ is symmetric in $(\mathbf{x}, t)$ and $(\mathbf{x}', t')$. Explicitly, interchanging $(\mathbf{x}, t)$ and $(\mathbf{x}', t')$ yields:

$$\begin{aligned}
\Theta_{\text{wave}}^{\text{ntk}}\big((\mathbf{x}', t'), (\mathbf{x}, t); \boldsymbol{\theta}\big) &= \int_U \int_U \frac{\delta\mathcal{G}(\mathbf{x}', t')}{\delta m(\mathbf{y})}[m] \cdot \frac{\delta\mathcal{G}(\mathbf{x}, t)}{\delta m(\mathbf{z})}[m] \cdot K_\tau(\mathbf{y}, \mathbf{z}; \boldsymbol{\theta}) dy d\mathbf{z} \\
&= \int_U \int_U \frac{\delta\mathcal{G}(\mathbf{x}, t)}{\delta m(\mathbf{y})}[m] \cdot \frac{\delta\mathcal{G}(\mathbf{x}', t')}{\delta m(\mathbf{z})}[m] \cdot K_\tau(\mathbf{z}, \mathbf{y}; \boldsymbol{\theta}) d\mathbf{z} dy \\
&= \Theta_{\text{wave}}^{\text{ntk}}\big((\mathbf{x}, t), (\mathbf{x}', t'); \boldsymbol{\theta}\big)
\end{aligned} \tag{95}$$

where $K_\tau(\mathbf{y}, \mathbf{z}; \boldsymbol{\theta}) = K_\tau(\mathbf{y}, \mathbf{z}; \boldsymbol{\theta})$ by the symmetry of $K_n$.

**Semi-Positive Definiteness.** To show semi-positive definiteness, we must verify that for any square-integrable function $f(\mathbf{x}, t)$, the double integral

$$\begin{aligned}
I &= \iint f(\mathbf{x}, t) \, \Theta_{\text{wave}}^{\text{ntk}}\big((\mathbf{x}, t), (\mathbf{x}', t'); \boldsymbol{\theta}\big) f(\mathbf{x}', t') d\mathbf{x} dt dx' dt' \\
&= \iint f(\mathbf{x}, t) \Big[ \int_U \int_U \frac{\delta\mathcal{G}(\mathbf{x}, t)}{\delta m(\mathbf{y})}[m] \frac{\delta\mathcal{G}(\mathbf{x}', t')}{\delta m(\mathbf{z})}[m] \\
&\qquad \cdot K_\tau(\mathbf{y}, \mathbf{z}; \boldsymbol{\theta}) dy dz \Big] f(\mathbf{x}', t') d\mathbf{x} dt dx' dt' \\
&= \int_U \int_U K_\tau(\mathbf{y}, \mathbf{z}; \boldsymbol{\theta}) \left[ \iint_{U \times T} f(\mathbf{x}, t) \frac{\delta\mathcal{G}(\mathbf{x}, t)}{\delta m(\mathbf{y})}[m] \, d\mathbf{x} \, dt \right] \\
&\qquad \cdot \left[ \iint_{U \times T} f(\mathbf{x}', t') \frac{\delta\mathcal{G}(\mathbf{x}', t')}{\delta m(\mathbf{z})}[m] \, d\mathbf{x}' \, dt' \right] dy \, d\mathbf{z} \\
&= \int_U \int_U K_\tau(\mathbf{y}, \mathbf{z}; \boldsymbol{\theta}) \, g(\mathbf{y}) \, g(\mathbf{z}) \, dy \, d\mathbf{z} \geq 0
\end{aligned} \tag{96}$$

where $g(\mathbf{y}) \triangleq \iint_{U \times T} f(\mathbf{x}, t) \frac{\delta\mathcal{G}(\mathbf{x}, t)}{\delta m(\mathbf{y})}[m] \, d\mathbf{x} \, dt$ and $K_\tau$ is semi-positive definite. Thus, $I \geq 0$, proving that $\Theta_{\text{wave}}^{\text{ntk}}$ is semi-positive definite. $\qquad\square$

**Proposition E.3.** *The wave-based NTK $\Theta_{wave}^{ntk}$ defined in Proposition 3.1 degrades into the wave kernel $\Theta_{wave}$ defined in Proposition 2.1 without neural representation.*

*Proof.* The proof consists of showing that under the given conditions, the parameter-space inner product that defines the wave-based NTK reduces to the model-space inner product that represents the wave kernel. The Fréchet derivative of the model $m_{\boldsymbol{\theta}}$ w.r.t. the parameters $\boldsymbol{\theta}$ at a point $\mathbf{z}$ is:

$$\frac{dm_{\boldsymbol{\theta}}(\mathbf{y})}{d\boldsymbol{\theta}(\mathbf{z})} = \frac{d\boldsymbol{\theta}(\mathbf{y})}{d\boldsymbol{\theta}(\mathbf{z})} = \delta(\mathbf{y} - \mathbf{z}). \tag{97}$$

This follows because a change in the parameter $\boldsymbol{\theta}$ (i.e., the velocity model $m$) at the point $\mathbf{z}$ directly and solely affects the model $m_{\boldsymbol{\theta}}$ at the point $\mathbf{y} = \mathbf{z}$. This wave-based NTK in the *model space*, $K_\tau$, which measures the inner product of the functional gradients of the model output, is then given by:

$$\begin{aligned}
K_\tau(\mathbf{y}, \mathbf{z}; \boldsymbol{\theta}) &= \int_U \frac{dm_{\boldsymbol{\theta}}(\mathbf{y})}{d\boldsymbol{\theta}(w)} \cdot \frac{dm_{\boldsymbol{\theta}}(\mathbf{z})}{d\boldsymbol{\theta}(w)} dw \\
&= \int_U \delta(\mathbf{y} - w)\delta(\mathbf{z} - w) dw = \delta(\mathbf{y} - \mathbf{z}).
\end{aligned} \tag{98}$$

The inner product is taken over the parameter space, resulting in the Dirac delta function due to the orthogonality of the functional gradients; a perturbation at $w$ affects the output only at $w$. The general definition of the wave-based NTK is an inner product over the parameter space $\boldsymbol{\theta}$:

$$\begin{aligned}
\Theta_{\text{wave}}^{\text{ntk}}\big((\mathbf{x}, t), (\mathbf{x}', t'); \boldsymbol{\theta}\big) &= \int_U \int_U \frac{\delta\mathcal{G}(\mathbf{x}, t)}{\delta m(\mathbf{y})}[m] \frac{\delta\mathcal{G}(\mathbf{x}', t')}{\delta m(\mathbf{z})}[m] \cdot K_m(\mathbf{y}, \mathbf{z}; \boldsymbol{\theta}) dy d\mathbf{z} \\
&= \int_U \int_U \frac{\delta\mathcal{G}(\mathbf{x}, t)}{\delta m(\mathbf{y})}[m] \frac{\delta\mathcal{G}(\mathbf{x}', t')}{\delta m(\mathbf{z})}[m] \cdot \delta(\mathbf{y} - \mathbf{z}) dy d\mathbf{z} \\
&= \int_U \frac{\delta\mathcal{G}(\mathbf{x}, t)}{\delta m(\mathbf{y})}[m] \frac{\delta\mathcal{G}(\mathbf{x}', t')}{\delta m(\mathbf{y})}[m] dy \\
&= \Theta_{\text{wave}}\big((\mathbf{x}, t), (\mathbf{x}', t'); m\big).
\end{aligned} \tag{99}$$

This completes the proof that the wave-based neural tangent kernel reduces to the standard wave kernel when the neural representation is the identity function. $\square$

**Proposition E.4.** *Consider the wavefield evolution equation governed by the wave kernel:*

$$\frac{\partial(u_{syn}^D(\tau) - u_{obs}^D)}{\partial\tau} = -\mathcal{K}(\Theta)[u_{syn}^D(\tau) - u_{obs}^D]$$

*where $\mathcal{K}(\Theta)$ is an integral operator defined as $\mathcal{K}(\Theta)[u] \triangleq \int_D \int_T \Theta\big((\mathbf{x}, t), (\mathbf{x}', t')\big) \cdot u(\mathbf{x}', t') dt' d\mathbf{x}'$. Then, $\mathcal{K}(\Theta)$ has spectral decomposition $\mathcal{K}(\Theta) = \sum_{k=1}^{\infty} \lambda_k \phi_k \otimes \phi_k$, where $\{\lambda_k\}_{k=1}^{\infty}$ is a sequence of non-negative eigenvalues and $\{\phi_k\}_{k=1}^{\infty}$ forms a complete orthonormal system of eigenfunctions. Define the spectral projection operator $\mathbf{Q}$ as $\mathbf{Q}f \triangleq \langle f, \phi_k \rangle_{k=1}^{\infty}$ and the diagonal matrix $\boldsymbol{\Lambda} = diag(\lambda_1, \lambda_2, \ldots)$. Then the evolution of the data residual in the spectral domain satisfies:*

$$|\mathbf{Q}(u_{syn}^D(\tau) - u_{obs}^D)| = e^{-\boldsymbol{\Lambda}\tau}|\mathbf{Q}(u_{obs}^D - u_{syn}^D(0))|$$

*where the absolute value is understood component-wise.*

*Proof.* Let the data residual function $e(\tau) = u_{\text{syn}}^D(\tau) - u_{\text{obs}}^D$, and the evolution equation becomes:

$$\frac{\partial e(\tau)}{\partial\tau} = -\mathcal{K}(\Theta)[e(\tau)] \tag{100}$$

Since $\mathcal{K}(\Theta)$ is self-adjoint, compact, and non-negative definite (see Proposition E.2), by the spectral theorem for compact self-adjoint operators, there exists a complete orthonormal system $\{\phi_k\}_{k=1}^{\infty}$ and non-negative eigenvalues $\{\lambda_k\}_{k=1}^{\infty}$ such that $\mathcal{K}(\Theta) = \sum_{k=1}^{\infty} \lambda_k \phi_k \otimes \phi_k$. Hence, we can expand the residual function in this eigen-basis:

$$e(\tau) = \sum_{k=1}^{\infty} a_k(\tau)\phi_k$$

where the expansion coefficients are given by $a_k(\tau) = \langle e(\tau), \phi_k \rangle$. The spectral projection operator is defined as $\mathbf{Q}e(\tau) = \{a_k(\tau)\}_{k=1}^{\infty}$. Substituting the expansion into equation 100:

$$\frac{\partial}{\partial\tau}\sum_{k=1}^{\infty} a_k(\tau)\phi_k = -\mathcal{K}(\Theta)\left[\sum_{k=1}^{\infty} a_k(\tau)\phi_k\right] = -\sum_{k=1}^{\infty} a_k(\tau)\mathcal{K}(\Theta)[\phi_k] = -\sum_{k=1}^{\infty} a_k(\tau)\lambda_k\phi_k$$

Due to the orthonormality of $\{\phi_k\}_{k=1}^{\infty}$, we obtain the system of ordinary differential equations:

$$\frac{da_k(\tau)}{d\tau} = -\lambda_k a_k(\tau) \quad \text{for all } k = 1, 2, \ldots \tag{101}$$

The general solution to equation equation 101 is $a_k(\tau) = a_k(0)e^{-\lambda_k\tau}$. Therefore, the solution becomes $a_k(\tau) = \langle u_{\text{syn}}^D(0) - u_{\text{obs}}^D, \phi_k \rangle e^{-\lambda_k\tau}$. In the spectral domain, we have:

$$\mathbf{Q}e(\tau) = \{a_k(\tau)\}_{k=1}^{\infty} = \{\langle u_{\text{syn}}^D(0) - u_{\text{obs}}^D, \phi_k \rangle e^{-\lambda_k\tau}\}_{k=1}^{\infty}$$
$$= -e^{-\boldsymbol{\Lambda}\tau}\mathbf{Q}(u_{\text{syn}}^D(0) - u_{\text{obs}}^D).$$

Taking absolute values component-wise:

$$|\mathbf{Q}e(\tau)| = e^{-\boldsymbol{\Lambda}\tau}|\mathbf{Q}(u_{\text{syn}}^D(0) - u_{\text{obs}}^D)|, \tag{102}$$

where $e^{-\boldsymbol{\Lambda}\tau}$ is a diagonal matrix with non-negative entries $e^{-\lambda_k\tau} \geq 0$. This completes the proof. $\square$

Next, we apply Theorem E.1 to prove Theorems 4.2, 5.1, and 5.2.

### E.4.1 Proof of Theorem 4.2

***Proof of Theorem*** *4.2.* Recall the definitions of the operators:

$$\mathcal{T}_{\text{wave}} = \mathcal{F}_m K_\delta \mathcal{F}_m^*, \quad \text{(wave kernel operator)}$$
$$\mathcal{T}_{\text{wave}}^{\text{ntk}} = \mathcal{F}_m K_\tau \mathcal{F}_m^*, \quad \text{(wave-based NTK operator)} \tag{103}$$

where $\mathcal{F}_m : L^2(U) \to L^2(D \times T)$ is the Fréchet derivative operator defined in the main text. The operator $K_\delta : L^2(U) \to L^2(U)$ denotes the identity kernel operator, which is equivalent to the identity operator $I$ on $L^2(U)$. Under proper neural network initialization, the NTK operator $K\tau : L^2(U) \to L^2(U)$ satisfies $|K_\tau|_{\text{op}} \leq 1$. This implies the operator inequality $K_\delta \succeq K_\tau$, i.e., $K_\delta - K_\tau$ is a positive semi-definite operator. Applying the Spectral Comparison Theorem E.1 with $K_1 = K_\delta$ and $K_2 = K_\tau$, we immediately obtain the eigenvalue comparison:

$$\lambda_j(\mathcal{T}_{\text{wave}}) = \lambda_j(\mathcal{F}_m K_\delta \mathcal{F}_m^*) \geq \lambda_j(\mathcal{F}_m K_\tau \mathcal{F}_m^*) = \lambda_j(\mathcal{T}_{\text{wave}}^{\text{ntk}}) \tag{104}$$

for all $j = 1, 2, \dots$. This establishes the desired eigenvalue decay relationship between the standard wave kernel and the wave-based NTK. $\square$

### E.4.2 Proof of Theorem 5.1

***Proof of Theorem*** *5.1.* Recall the operator definitions for the two learning paradigms:

$$\mathcal{T}_{\text{INR}} = \mathcal{F}m K_{\text{INR}} \mathcal{F}_m^*, \quad \text{(operator of INR-FWI)}$$
$$\mathcal{T}_{\text{MPE}} = \mathcal{F}_m K_{\text{MPE}} \mathcal{F}_m^*, \quad \text{(operator of MPE-FWI)} \tag{105}$$

where $K_{\text{INR}}$ and $K_{\text{MPE}}$ are the NTK operators for the INR and MPE models, respectively. From Audia et al. (2025), we have the operator decomposition:

$$K_{\text{MPE}} = K_{\text{INR}} + K^+ \tag{106}$$

where $K^+$ is a positive semi-definite operator. This decomposition implies the operator inequality $K_{\text{MPE}} \succeq K_{\text{INR}}$, as $K_{\text{MPE}} - K_{\text{INR}} = K^+ \succeq 0$. Applying Theorem E.1 with $K_1 = K_{\text{MPE}}$ and $K_2 = K_{\text{INR}}$, we directly obtain the spectral enhancement property:

$$\lambda_j(\mathcal{T}_{\text{MPE}}) = \lambda_j(\mathcal{F}_m K_{\text{MPE}} \mathcal{F}_m^*) \geq \lambda_j(\mathcal{F}_m K_{\text{INR}} \mathcal{F}_m^*) = \lambda_j(\mathcal{T}_{\text{INR}}) \tag{107}$$

for all $j = 1, 2, \dots$. This completes the proof that the MPE approach exhibits enhanced spectral properties compared to the standard INR approach. $\square$

### E.4.3 Proof of Theorem 5.2

*Proof.* We consider three CR-FWI methods: INR, MPE, and IG (INR-Grid). Let their respective model representations be:

$$m_{\text{INR}}(\mathbf{x}) = \text{MLP}_{\boldsymbol{\theta}}(\phi_{\boldsymbol{\theta}1}^{\text{INR}}(\mathbf{x})),$$
$$m_{\text{MPE}}(\mathbf{x}) = \text{MLP}_{\boldsymbol{\theta}}(\phi_{\boldsymbol{\theta}2}^{\text{MPE}}(\mathbf{x})), \tag{108}$$
$$m_{\text{IG}}(\mathbf{x}) = \text{MLP}_{\boldsymbol{\theta}}(\phi_{\boldsymbol{\theta}3}^{\text{IG}}(\mathbf{x})),$$

where $\phi_{\boldsymbol{\theta}_2}^{\text{IG}}(\mathbf{x}) = \sqrt{\alpha} \cdot \phi_{\boldsymbol{\theta}_2}^{\text{INR}}(\mathbf{x}) \bigoplus \sqrt{(1-\alpha)} \cdot \phi_{\boldsymbol{\theta}_2}^{\text{MPE}}(\mathbf{x})$ denotes the concatenation of feature vectors with a weighting factor $\alpha \in [0, 1]$. A key observation is that the NTK for each method admits a natural decomposition into two components: one from the MLP parameters and one from the feature encoding parameters. For any method, we have:

$$K(\mathbf{y}, \mathbf{z}) = \sum_{i=1}^{p} \frac{\partial m_{\boldsymbol{\theta}}(\mathbf{y})}{\partial \boldsymbol{\theta}i} \frac{\partial m\boldsymbol{\theta}(\mathbf{z})}{\partial \boldsymbol{\theta}i}$$
$$= \underbrace{\sum_{i=1}^{p_1-1} \frac{\partial m_{\boldsymbol{\theta}}(\mathbf{y})}{\partial \boldsymbol{\theta}i} \frac{\partial m\boldsymbol{\theta}(\mathbf{z})}{\partial \boldsymbol{\theta}i}}_{K_{\text{MLP}}} + \underbrace{\sum_{i=p_1}^{p} \frac{\partial m_{\boldsymbol{\theta}}(\mathbf{y})}{\partial \boldsymbol{\theta}i} \frac{\partial m\boldsymbol{\theta}(\mathbf{z})}{\partial \boldsymbol{\theta}i}}_{K_{\text{Encode}}} \tag{109}$$

Crucially, we assume that all three methods share the same MLP architecture, and thus have identical $K_{\text{MLP}}$ components in the infinite-width limit. The differences arise only in the encoding components:

$$
\begin{aligned}
K_{\text{INR}} &= K_{\text{MLP}} + K_{\text{Encode}}^{\text{INR}}, \\
K_{\text{MPE}} &= K_{\text{MLP}} + K_{\text{Encode}}^{\text{MPE}}, \\
K_{\text{IG}} &= K_{\text{MLP}} + K_{\text{Encode}}^{\text{IG}}
\end{aligned}
\tag{110}
$$

From Audia et al. (2025), we have the operator inequality $K_{\text{Encode}}^{\text{INR}} \preceq K_{\text{Encode}}^{\text{MPE}}$. For the IG method, we assume the INR and MPE feature gradients are orthogonal, and we obtain:

$$
\begin{aligned}
K_{\text{Encode}}^{\text{IG}}(\mathbf{x}, \mathbf{x}') &= \langle \sqrt{\alpha} \cdot \phi_{\boldsymbol{\theta}_1}^{\text{INR}}(\mathbf{x}) \oplus \sqrt{1-\alpha} \cdot \phi_{\boldsymbol{\theta}_2}^{\text{MPE}}(\mathbf{x}), \sqrt{\alpha} \cdot \phi_{\boldsymbol{\theta}_1}^{\text{INR}}(\mathbf{x}') \oplus (1-\alpha) \cdot \phi_{\boldsymbol{\theta}_2}^{\text{MPE}}(\mathbf{x}') \rangle \\
&= \alpha \cdot \langle \phi_{\boldsymbol{\theta}_1}^{\text{INR}}(\mathbf{x}), \phi_{\boldsymbol{\theta}_1}^{\text{INR}}(\mathbf{x}') \rangle + (1-\alpha) \cdot \langle \phi_{\boldsymbol{\theta}_2}^{\text{MPE}}(\mathbf{x}), \phi_{\boldsymbol{\theta}_2}^{\text{MPE}}(\mathbf{x}') \rangle \\
&= \alpha \cdot K_{\text{Encode}}^{\text{INR}}(\mathbf{x}, \mathbf{x}') + (1-\alpha) \cdot K_{\text{Encode}}^{\text{MPE}}(\mathbf{x}, \mathbf{x}')
\end{aligned}
\tag{111}
$$

This linear combination preserves the ordering:

$$
K_{\text{Encode}}^{\text{INR}} \preceq K_{\text{Encode}}^{\text{IG}} \preceq K_{\text{Encode}}^{\text{MPE}}
\tag{112}
$$

Since all methods share the same $K_{\text{MLP}}$ component, this ordering extends to the full NTK:

$$
K_{\text{INR}} \preceq K_{\text{IG}} \preceq K_{\text{MPE}}
\tag{113}
$$

Defining the following wave-based NTK operators:

$$
\begin{aligned}
\mathcal{T}_{\text{INR}} &= \mathcal{F}_m K_{\text{INR}} \mathcal{F}_m^*, \\
\mathcal{T}_{\text{IG}} &= \mathcal{F}_m K_{\text{IG}} \mathcal{F}_m^*, \\
\mathcal{T}_{\text{MPE}} &= \mathcal{F}_m K_{\text{MPE}} \mathcal{F}_m^*,
\end{aligned}
\tag{114}
$$

and applying Theorem E.1 twice yields the desired spectral ordering:

$$
\lambda_i(\mathcal{T}_{\text{INR}}) \leq \lambda_i(\mathcal{T}_{\text{IG}}) \leq \lambda_i(\mathcal{T}_{\text{MPE}}) \quad \text{for all } i = 1, 2, \ldots
\tag{115}
$$

This completes the proof. $\qquad\square$

