# OpenReview forum: "Unveiling the Mechanism of Continuous Representation Full-Waveform Inversion: A Wave Based Neural Tangent Kernel Framework"
_ICLR.cc/2026/Conference — ICLR 2026 Poster_

### Official Review · Reviewer_Lw1m · 2025-10-28

**Soundness:** 3
**Presentation:** 3
**Contribution:** 3
**Rating:** 6
**Confidence:** 3

**Summary:**

In this paper, the authors propose a unified theoretical analysis framework based on the wave kernel and
wave-based NTK to investigate the convergence and robustness of both conventional and continuous representation FWI. Their analysis reveals the distinct eigenvalue decay patterns, which explain why traditional FWI has fast convergence but lacks robustness, while INR-based CRFWI has improved robustness but with slower convergence. Motivated by theoretical analysis, they introduce a hybrid
representation that integrates INR with a multigrid strategy for FWI, aiming to balance robustness
and convergence. Experiments demonstrate the superior performance of the proposed methods.

**Strengths:**

1. The paper proposes a unified wave-based Neural Tangent Kernel (NTK) framework that connects conventional FWI and CR-FWI. It provides a solid theoretical explanation for observed differences in convergence and robustness across FWI variants.

2. Across multiple benchmarks, the proposed methods show strong performance and robustness under different conditions (e.g., noisy data, poor initial models, missing low-frequency content, etc).

3. For the most part, the manuscript is generally well written, clearly organized, and easy to follow.

**Weaknesses:**

1. Missing discussion of weighting factor $\alpha$. The weighting factor α in IG-FWI is introduced but never discussed or ablated. Since it determines the balance between Robustness and Convergence, its selection strategy and sensitivity analysis should be provided.

2. Lack of convergence evidence. Although the convergence rate is repeatedly discussed in the paper, the paper does not show any related results, e.g., convergence curves, generated velocity visualizations corresponding to different iterations, or the number of iterations required to converge.

3. The claim that “IG-FWI achieves an optimal trade-off between robustness and convergence rate” is overclaimed. The paper provides no formal definition or proof of optimality.

4. The multi-grid parametric encoding module is largely adapted from the existing INR paper. While this integration is effective, it somewhat limits the originality of the methodological contribution.

5. The Figures 10-15 in the Appendix are in very low resolution. Replacing them with high-resolution or vector graphics would improve readability.

**Questions:**

1. Missing experiments about $\alpha$ and convergence. Please refer to Weakness for more details.

2. The introduction of LR-FWI seems insufficiently motivated and conceptually detached from the main theoretical analysis. Its role within the paper is unclear. Could you please clarify its motivation and the connection of the LR-FWI with other parts of the paper?

---

> ### Author Response · Authors · 2025-11-19
> **Response to Reviewer Lw1m**
>
> We sincerely thank the reviewer for the valuable and constructive comments! We have carefully conducted further discussions and supplemented additional experiments based on your comments. Please see the details in the following and the revised manuscript.
>
> >**W1: Missing discussion of weighting factor $a$. The weighting factor $a$ in IG-FWI is introduced but never discussed or ablated. Since it determines the balance between Robustness and Convergence, its selection strategy and sensitivity analysis should be provided.**
>
> Thanks! In IG-FWI, the weighting factor $a$ was set to 0.5 across all experiments to balance robustness and high-frequency convergence. Following your suggestion, we have conducted a comprehensive ablation study to investigate the impact of the weighting factor $a$, as detailed in the revised manuscript (See Fig. 9). We also paste tables here for your convenience. These results show that our proposed IG-FWI method is robust across a range of weighting factor $a$ from 0.3 to 0.7.
>
> **Table. Ablation experiment about the weighting factor $a$ in IG-FWI.**
>
> | Weighting factor | 0.1 | 0.3 | 0.5 | 0.7 | 0.9 |
> |:-----------------|:---:|:---:|:---:|:---:|:---:|
> | Marmousi (MSE) | 0.6343 | 0.2754 | 0.2973 | 0.3228 | 0.3840 |
> | Salt body (MSE) | 0.2480 | 0.1674 | 0.1895 | 0.3286 | 0.2713 |
>
> Although an appropriate weighting factor must be selected, our proposed IG-FWI method exhibits relative insensitivity to intermediate values of the weighting factor. In practice, when data quality and the initial model are reliable, a larger $a$ can be adopted to accelerate convergence; conversely, a smaller $a$ is advisable when data or initial model quality is poor.
>
>
> >**W2: Lack of convergence evidence. Although the convergence rate is repeatedly discussed in the paper, the paper does not show any related results, e.g., convergence curves, generated velocity visualizations corresponding to different iterations, or the number of iterations required to converge.**
>
> Thanks. We have additionally provided convergence curves of velocity model error, generated velocity visualizations corresponding to different iterations, and the number of iterations required to converge, as detailed in Appendix B.6.7 of the revised manuscript (See Fig. 7 and Figs. 18-21). We also paste tables here for your convenience. The results indicate that conventional FWI and MPE-FWI methods achieve rapid convergence of high-frequency details at the cost of increased sensitivity, while IFWI and WinFWI enhance robustness at the expense of high-frequency details of inversion results. Our proposed IG-FWI and LR-FWI methods achieve a more balanced trade-off, maintaining robustness without sacrificing the convergence rate for high-frequency information.
>
> **Table. Epochs and Time Required to achieve the MSE equals 1 on the Overthrust model.**
>
> |Methods | IFWI | WinFWI | LR-FWI (Ours) | IG-FWI (Ours) |
> |----------------|---------------|--------|---------------|----------------|
> | Epoch    |  824  | 498  |  294    |  367    |
> | Time (min)   | 25.8  | 15.8  |   9.36    |  11.7     |
>
>
> >**W3: The claim that “IG-FWI achieves an optimal trade-off between robustness and convergence rate” is overclaimed. The paper provides no formal definition or proof of optimality.**
>
> Thanks! We have revised the manuscript to replace the imprecise phrase "optimal trade-off" with the more accurate description "a more balanced trade-off".

---

> > ### Author Response · Authors · 2025-11-19
> > **Response to Reviewer Lw1m**
> >
> > >**W4: The multi-grid parametric encoding module is largely adapted from the existing INR paper. While this integration is effective, it somewhat limits the originality of the methodological contribution.**
> >
> > Thanks. We agree that the multi-grid parametric encoding module is largely adapted from the existing INR paper, but it may not be universally applicable to all FWI scenarios (See Fig. 13-17 in the revised manuscript). In our study, the unified wave-based NTK framework reveals the critical link between the eigenvalue decay rate of wave-based NTK and convergence behavior. Based on wave-based NTK theory, we integrate INR and MPE to achieve both robust and high-precision inversion results, which are suitable and tailored for FWI. Hence, our primary methodological novelty lies in its theory-guided algorithm design in the community. For instance, we can replace the tiny INR in IG-FWI with a low-rank tensor function representation (termed LRG-FWI). We have presented additional experiments using the Marmousi, SEG/EAGE Salt and Overthrust models with a constant initial model, as detailed in Appendix C.5 of the revised manuscript (See Fig. 34). We also paste the table here for your convenience. We see that this novel CR-FWI achieves high-precision inversion results and achieves performance comparable to that of IG-FWI .
> >
> > **Table. Comparisons with LRG-FWI and other CR-FWI methods.**
> >
> > | Methods | MPE-FWI | LR-FWI | LRG-FWI | IG-FWI |
> > |:---------------|:----------------:|:-------------:|:--------------:|-------:|
> > | Marmousi (MSE) | 2.2266 | 0.2893 | 0.1995 | 0.2961 |
> > | Salt body (MSE) | 1.1008 | 0.2368 | 0.2179 | 0.1883 |
> >
> >
> > >**W5: The Figures 10-15 in the Appendix are in very low resolution. Replacing them with high-resolution or vector graphics would improve readability.**
> >
> > Thanks. We have replaced Figs. 10-15 in the Appendix with high-resolution vector graphics to improve clarity. The updated figures are now included in the revised manuscript (See Figs. 13-17).
> >
> > >**Q1: Missing experiments about $a$ and convergence. Please refer to Weakness for more details.**
> >
> > Thanks. We have provided additional experiments about the weighing factor $a$ and convergence evidence in the previous response.
> >
> > >**Q2: The introduction of LR-FWI seems insufficiently motivated and conceptually detached from the main theoretical analysis. Its role within the paper is unclear. Could you please clarify its motivation and the connection of the LR-FWI with other parts of the paper?**
> >
> > Thanks! We acknowledge that this discussion is key to understanding its promising performance of the LR-FWI method. In seismic exploration, the parameter models of the wave equation (P-wave velocity, S-wave velocity, and density) typically exhibit low-rank characteristics due to the inherent regularity of geological sedimentation [R1]. The technique of low-rank tensor function representation (LRTFR) [R2] offers a low-rank continuous representation that incorporates both smoothness and low-rank regularization, which has demonstrated its effectiveness in diverse applications such as seismic denoising, hyperspectral image processing, and traffic flow prediction. In this study, we incorporate this low-rank representation into FWI, introducing structural low-rank regularization to improve inversion robustness. Moreover, the proposed LR-FWI method enhances the convergence rate for high-frequency components due to an appropriate eigenvalue decay rate, which we have empirically verified through numerical experiments (See Fig. 5 (c) in the revised manuscript). However, providing a rigorous mathematical proof of the eigenvalue decay rate remains challenging due to the complex structure of the tensor product, which will be a key objective of our future research based on tensor theory [R3] and matrix analysis [R4].
> >
> > [R1] Jiahang Li, Hitoshi Mikada, and Junichi Takekawa. Inexact augmented Lagrangian method-based full-waveform inversion with randomized singular value decomposition. Journal of Geophysics and Engineering, 21(2):572–597, 2024a.
> >
> > [R2] Yisi Luo, Xile Zhao, Zhemin Li, Michael K Ng, and Deyu Meng. Low-rank tensor function representation for multi-dimensional data recovery. IEEE transactions on pattern analysis and machine intelligence, 46(5):3351–3369, 2023.
> >
> > [R3] Tamara G. Kolda and Brett W. Bader. Tensor decompositions and applications. SIAM review, 51(3):455–500, 2009.
> >
> > [R4] Roger A. Horn and Charles R. Johnson. Matrix analysis. Cambridge University Press, 2012.
> >
> >
> > We will include the above discussions and additional experiments in the revised manuscript, and all revisions made to the paper will be highlighted in blue for your convenience. Should you need further information, please let us know. We look forward to hearing your feedback!

---

> ### Comment · Reviewer_Lw1m · 2025-11-26
>
> Thanks to the authors’ detailed response, my questions were answered. I will increase my score to 8.

---

> > ### Author Response · Authors · 2025-11-26
> >
> > We thank the reviewer again for the valuable and constructive comments, which have been invaluable in improving the quality of our paper!

---

### Official Review · Reviewer_GUpL · 2025-10-31

**Soundness:** 3
**Presentation:** 3
**Contribution:** 3
**Rating:** 8
**Confidence:** 4

**Summary:**

The authors study full waveform inversion (FWI), in particular the case where the material field is parameterized with a neural network (the "continuous representation FWI"). Previous work has identified challenges such as slow convergence for high frequencies, but good performance and fast convergence for smooth fields. Two main questions are adressed: how to explain the differences in robustness and convergence between conventional and network-based FWI, and how to find a continuous representation (e.g., neural network) that can achieve a good trade-off between accuracy in the high frequency domain and convergence speed.
The authors use the neural tangent kernel perspective to study the frequency domain properties, and introduce a combination of representations tailored to FWI problems, including multigrid approaches, to resolve the frequency issues.
The theoretical results are supported by several computational experiments in seismic imaging, where the approach works well.

**Strengths:**

1) The authors employ NTK theory to study the spectral bias problem when using neural networks in an FWI setting. This is a very well defined problem, and has not been addressed in a lot of previous studies; even though the idea of using neural netowkrs for the material field is used in several works. This makes the study particularly relevant, not just for seismic imaging, but also nondestructive testing.
2) The authors not only analyze the issues of FWI in this setting, but also propose a new method using multigrid, and demonstrate its performance in 2D examples. This makes this work a significant advancement in the field.
3) The appendix includes a thorough explanation of the proof, as well as details on the experiments.

**Weaknesses:**

* The manuscript contains the term "nonlinear partial differential equation" multiple times (l011, l046, l104, l107, l232), but equation (1) is a linear PDE (the classical wave equation). The linearity of the wave equation does not matter for the inverse problem being nonlinear, but it is strange that the authors refer to the nonlinearity so often.
 * More literature on FWI with neural networks material fields should be cited.

 - Rasht‐Behesht, Majid, Christian Huber, Khemraj Shukla, and George Em Karniadakis. 2022. “Physics‐Informed Neural Networks (PINNs) for Wave Propagation and Full Waveform Inversions.” Journal of Geophysical Research: Solid Earth 127 (5). https://doi.org/10.1029/2021JB023120.
 - Herrmann, Leon, Tim Bürchner, Felix Dietrich, and Stefan Kollmannsberger. 2023. “On the Use of Neural Networks for Full Waveform Inversion.” Computer Methods in Applied Mechanics and Engineering 415 (October): 116278. https://doi.org/10.1016/j.cma.2023.116278.

 * Only 2D examples are considered, not 3D. There is a significant computational challenge in FWI going from 2D to 3D, and it is not clear if the proposed method can still be used there.

**Questions:**

1) Why do the authors refer to the wave equation being nonlinear, and then state it as a linear PDE? There are nonlinear wave equations, but eq. 1 is not.
2) What is the time and memory complexity of the method introduced in the manuscript? FWI is used beyond seismic imaging (e.g. in nondestructive testing), in 3D domains, where the computational challenge is much larger than in 2d domains.

---

> ### Author Response · Authors · 2025-11-19
> **Response to Reviewer GUpL**
>
> We sincerely thank the reviewer for the valuable and constructive comments! We have carefully conducted further discussions and supplemented additional experiments based on your comments. Please see the details in the following and the revised manuscript.
>
> >**W1: The manuscript contains the term "nonlinear partial differential equation" multiple times (l011, l046, l104, l107, l232), but equation (1) is a linear PDE (the classical wave equation). The linearity of the wave equation does not matter for the inverse problem being nonlinear, but it is strange that the authors refer to the nonlinearity so often.**
>
> Thanks for your careful review. We agree and have corrected in the revised manuscript to clarify that the wave equation itself is linear for the wavefield data, and the wave equation-based inverse problem is nonlinear due to the parameter-to-wavefield relationship, i.e., $G(m_1 + m_2) \neq G(m_1) + G(m_2)$.
>
>
> >**W2: More literature on FWI with neural networks material fields should be cited.**
>
> **[1] Rasht‐Behesht, Majid, Christian Huber, Khemraj Shukla, and George Em Karniadakis. 2022. “Physics‐Informed Neural Networks (PINNs) for Wave Propagation and Full Waveform Inversions.” Journal of Geophysical Research: Solid Earth 127 (5). https://doi.org/10.1029/2021JB023120.**
>
> **[2] Herrmann, Leon, Tim Bürchner, Felix Dietrich, and Stefan Kollmannsberger. 2023. “On the Use of Neural Networks for Full Waveform Inversion.” Computer Methods in Applied Mechanics and Engineering 415 (October): 116278. https://doi.org/10.1016/j.cma.2023.116278.**
>
> Thanks. We have cited related literature on FWI with neural networks material fields in the related work part of the revised manuscript, i.e.,
>
> **"Herrmannetal.(2023) and Rasht-Behesht
>  etal.(2022) introduced a physics-informed neural network (PINN)-based FWI methods, which use two INRs to represent the velocity model and the seismic wavefield, respectively. Physical constraints are incorporated via a penalty term derived from the wave equation within the loss function."**
>
>
> >**W3: Only 2D examples are considered, not 3D. There is a significant computational challenge in FWI going from 2D to 3D, and it is not clear if the proposed method can still be used there.**
>
> Thanks. Following your suggestion, we have provided additional 3D FWI experiments using the SEG/EAGE Overthrust model with a smooth initial model, as detailed in Appendix C.2 of the revised manuscript (See Fig. 25 and Tab. 6). We also paste tables here for your convenience. These results indicate that our proposed CR-FWI methods can be used in 3D inversion scenarios by incorporating computational techniques such as checkpointing strategies [R1], random mini-batch algorithms [R2], and gradient accumulation [R3].
>
> **Table. Experiments on 3D SEG/EAGE Overthrust model using CR-FWI methods.**
>
> | Methods | ADFWI | IFWI | LR-FWI | IG-FWI |
> | :--- | :--- | :--- | :--- | :--- |
> | Normalized MSE | 0.1469 | 0.1302 | 0.0968 | 0.1151 |
>
> [R1] John E. Anderson, Lijian Tan, and Don Wang. Time-reversal checkpointing methods for RTM and FWI. Geophysics, 77(4):S93–S103, 2012.
>
> [R2] Matthieu Simeoni, Sepand Kashani, Paul Hurley, and Martin Vetterli. Deepwave: A recurrent neural
> network for real-time acoustic imaging. Advances in Neural Information Processing Systems, 32, 2019.
>
> [R3] Dirk Philip van Herwaarden, Christian Boehm, Michael Afanasiev, Solvi Thrastarson, Lion Krischer, Jeannot Trampert, and Andreas Fichtner. Accelerated full-waveform inversion using dynamic mini-batches. Geophysical Journal International, 221(2):1427–1438, 2020.

---

> > ### Author Response · Authors · 2025-11-19
> > **Response to Reviewer GUpL**
> >
> > >**Q1: Why do the authors refer to the wave equation being nonlinear, and then state it as a linear PDE? There are nonlinear wave equations, but eq. 1 is not.**
> >
> > Thanks. We agree and have corrected in the revised manuscript to clarify that the wave equation itself is linear for the wavefield data $u$, i.e., $G(m)[u_1+u_2] = G(m)[u_1]+G(m)[u_2]$, and the wave equation-based inverse problem is nonlinear due to the parameter-to-wavefield relationship, i.e., $G(m_1 + m_2) \neq G(m_1) + G(m_2)$.
> >
> >
> > >**Q2: What is the time and memory complexity of the method introduced in the manuscript? FWI is used beyond seismic imaging (e.g., in nondestructive testing), in 3D domains, where the computational challenge is much larger than in 2d domains.**
> >
> > Thanks. The 3D FWI experiments provided in our previous response demonstrate the capability of our method in handling large-scale data. Furthermore, we provide computational analysis and memory requirements for our proposed CR-FWI methods in 2D and 3D FWI, which incorporate techniques such as checkpointing strategies, random mini-batch algorithms, and gradient accumulation, as detailed in Appendix C.3 of the revised manuscript (See Tab. 6). We also paste tables here for your convenience. These results show that our proposed CR-FWI methods do not impose additional computational burden compared with conventional FWI methods.
> >
> > **Table. Computational and memory requirements for CR-FWI in 3D FWI.**
> >
> > | Methods   | Conventional FWI | IFWI | LR-FWI (Ours) | IG-FWI (Ours) |
> > |---------------|----------------------|-----------|----------------|---------------|
> > | Time (s/iter)   |   13.82   |   15.08   |    14.61    |  14.09  |
> > | Memory (GB)  |  15.89   | 30.61  | 22.84  |  16.01  |
> >
> >
> > We will include the above discussions and additional experiments in the revised manuscript, and all revisions made to the paper will be highlighted in blue for your convenience. Should you need further information, please let us know. We look forward to hearing your feedback!

---

> > > ### Comment · Reviewer_GUpL · 2025-11-19
> > >
> > > I am happy with the provided answers and congratulate the authors to their nice work.
> > > The two most important points (linearity of wave equation, and 3D performance) are addressed properly now.

---

> > > > ### Author Response · Authors · 2025-11-19
> > > >
> > > > We thank the reviewer again for the valuable and constructive comments, which have been invaluable in improving the quality of our paper!

---

### Official Review · Reviewer_4aqd · 2025-11-01

**Soundness:** 3
**Presentation:** 3
**Contribution:** 3
**Rating:** 8
**Confidence:** 3

**Summary:**

This paper focuses on the problem of full-waveform Inversion (FWI) and investigates the mechanisms that cause the differences between conventional FWI and continuous representation FWI (CR-FWI) in robustness and convergence. Through a series of theoretical analyses from the perspective of the neural tangent kernel (NTK), the authors identified that these differences arise from different eigenvalue decay behaviors. Based on these observations, the authors further proposed LR-FWI, MPE-FWI, and IG-FWI, which aim to achieve a trade-off between robustness and convergence. Experimental results on several datasets and scenarios demonstrate the superior performance of the proposed methods.

**Strengths:**

1. I think the insights regarding eigenvalue decay behaviors are valuable to the community for future algorithm development.
2. The authors provided solid, step-by-step derivations of the theories proposed in the paper.
3. The proposed methods (MPE-FWI and IG-FWI) have clear theories to explain their performance in terms of accuracy, robustness and convergence.
4. The proposed methods achieved superior performance on the public benchmark datasets, which is convincing.

**Weaknesses:**

1. Among the proposed methods, LR-FWI lacks an analysis of its eigenvalue decay behavior and is discussed much less than the other two methods (MPE-FWI and IG-FWI). I think it is important as LR-FWI yields promising performance, and it even outperforms IG-FWI in some cases.
2. Although data-driven and physics-informed FWI methods are covered in the related works section, they are missing in the experiments. The experiments can be more solid by including a few of them as the additional baselines.

**Questions:**

1. The authors proposed to integrate a tiny INR and MPE in IG-FWI. Can the tiny INR be replaced with other INR-based methods such as LR-FWI?
2. In IG-FWI, the features from the tiny INR and MPE are concatenated with a weighting factor $\alpha$. What are the values of $\alpha$ in different experiments? Is the algorithm sensitive to the value of $\alpha$?
3. In Table 1, given smooth initial models, MPE-FWI performs worse than CR-FWI baselines except for Marmousi, but it should have slower eigenvalue decay. Could you please provide an explanation or analysis of these results?
4. [minor] In Equation 4 and Equation 7, there is $(\mathbf{x}’, t’)$ before $dt’d\mathbf{x}’$. What’s the meaning of this notation? From my understanding, they are redundant.

---

> ### Author Response · Authors · 2025-11-19
> **Response to Reviewer 4aqd**
>
> We sincerely thank the reviewer for the valuable and constructive comments! We have carefully conducted further discussions and supplemented additional experiments based on your comments. Please see the details in the following and the revised manuscript.
>
> >**W1: Among the proposed methods, LR-FWI lacks an analysis of its eigenvalue decay behavior and is discussed much less than the other two methods (MPE-FWI and IG-FWI). I think it is important as LR-FWI yields promising performance, and it even outperforms IG-FWI in some cases.**
>
> Thanks! We acknowledge that this discussion is key to understanding its promising performance of the LR-FWI method. In seismic exploration, the parameter models of the wave equation (P-wave velocity, S-wave velocity, and density) typically exhibit low-rank characteristics due to the inherent regularity of geological sedimentation [R1]. The technique of low-rank tensor function representation (LRTFR) [R2] offers a low-rank continuous representation that incorporates both smoothness and low-rank regularization, which has demonstrated its effectiveness in diverse applications such as seismic denoising, hyperspectral image processing, and traffic flow prediction. In this study, we incorporate this low-rank representation into FWI, introducing structural low-rank regularization to improve inversion robustness. Moreover, the proposed LR-FWI method enhances the convergence rate for high-frequency components due to an appropriate eigenvalue decay rate, which we have empirically verified through numerical experiments (See Fig. 5 (c) in the revised manuscript). However, providing a rigorous mathematical proof of the eigenvalue decay rate remains challenging due to the complex structure of the tensor product, which will be a key objective of our future research based on tensor theory [R3] and matrix analysis [R4].
>
> [R1] Jiahang Li, Hitoshi Mikada, and Junichi Takekawa. Inexact augmented Lagrangian method-based full-waveform inversion with randomized singular value decomposition. Journal of Geophysics and Engineering, 21(2):572–597, 2024a.
>
> [R2] Yisi Luo, Xile Zhao, Zhemin Li, Michael K Ng, and Deyu Meng. Low-rank tensor function representation for multi-dimensional data recovery. IEEE transactions on pattern analysis and machine intelligence, 46(5):3351–3369, 2023.
>
> [R3] Tamara G. Kolda and Brett W. Bader. Tensor decompositions and applications. SIAM review, 51(3):455–500, 2009.
>
> [R4] Roger A. Horn and Charles R. Johnson. Matrix analysis. Cambridge University Press, 2012.
>
>
>
>
> >**W2: Although data-driven and physics-informed FWI methods are covered in the related works section, they are missing in the experiments. The experiments can be more solid by including a few of them as the additional baselines.**
>
>
> Thanks for your valuable suggestion. We have presented additional comparisons with a data-driven FWI (e.g., supervised CNN-based method, InversionNet [R5]) method on OpenFWI datasets, as detailed in Appendix C.3 of the revised manuscript (See Figs. 26-32 and Tab. 2). We also paste the table here for your convenience. The results show that the LR-FWI and IG-FWI methods may yield better performance than some supervised CNN-based methods. Moreover, there is a lack of open-source code for PINN-FWI, which we are attempting to reproduce.
>
> **Table: Comparisons with data-driven FWI and CR-FWI methods.**
>
> | Methods | InversionNet | ADFWI | IFWI | LR-FWI (Ours) | IG-FWI (Ours) |
> |:---------------|:--------------:|:----------:|:--------------:|:-------------:|--------:|
> | FaltVel (MSE) | 36.78 | 21.30 | 11.72 | 6.65 | 6.94 |
> | CurveVel (MSE) | 94.16 | 57.07 | 16.87 | 9.89 | 7.93 |
> | FaltFault (MSE) | 113.18 | 48.23 | 46.59 | 19.59 | 28.84 |
> | CurveFault (MSE) | 100.09 | 99.14 | 65.89 | 60.23 | 24.61 |
> | Style (MSE) | 53.76 | 27.66 | 48.25 | 14.56 | 17.08 |
>
> While supervised frameworks suffer from generalization issues caused by the distribution shift between synthetic training data (e.g., from OpenFWI) and complex field models (e.g., Marmousi, Overthrust), their computational speed remains highly appealing. A promising pathway is to embed such methods within our proposed CR-FWI framework to rapidly generate initial models, which we plan to explore in future work.
>
> [R5] Yue Wu and Youzuo Lin. Inversionnet: An efficient and accurate data-driven full waveform inversion. IEEE Transactions on Computational Imaging,6:419–433,2019b.

---

> > ### Author Response · Authors · 2025-11-19
> > **Response to Reviewer 4aqd**
> >
> > >**Q1: The authors proposed to integrate a tiny INR and MPE in IG-FWI. Can the tiny INR be replaced with other INR-based methods, such as LR-FWI?**
> >
> > Thanks for your insightful question! Following your suggestion, we have developed a new hybrid CR-FWI method by replacing the tiny INR in IG-FWI with a low-rank representation (termed LRG-FWI). We have presented additional experiments using the Marmousi and SEG/EAGE Salt models with a constant initial model, as detailed in Appendix C.5 of the revised manuscript (See Fig. 34). We also paste the table here for your convenience. We see that this novel CR-FWI achieves high-precision inversion results and achieves performance comparable to that of IG-FWI .
> >
> > **Table. Comparisons with LRG-FWI and other CR-FWI methods.**
> >
> > | Methods | MPE-FWI | LR-FWI | LRG-FWI | IG-FWI |
> > |:---------------|:----------------:|:-------------:|:--------------:|-------:|
> > | Marmousi (MSE) | 2.2266 | 0.2893 | 0.1995 | 0.2961 |
> > | Salt body (MSE) | 1.1008 | 0.2368 | 0.2179 | 0.1883 |
> >
> >
> >
> > >**Q2: In IG-FWI, the features from the tiny INR and MPE are concatenated with a weighting factor a. What are the values of a in different experiments? Is the algorithm sensitive to the value of a ?**
> >
> > Thanks! In IG-FWI, the weighting factor $a$ was set to 0.5 across all experiments to balance robustness and high-frequency convergence. Following your suggestion, we have conducted a comprehensive ablation study to investigate the impact of the weighting factor $a$, as detailed in the revised manuscript (See Fig. 9). We also paste tables here for your convenience. These results show that our proposed IG-FWI method is robust across a range of weighting factor $a$ from 0.3 to 0.7.
> >
> > **Table. Ablation experiment about the weighting factor $a$ in IG-FWI.**
> >
> > | Weighting factor | 0.1 | 0.3 | 0.5 | 0.7 | 0.9 |
> > |:-----------------|:---:|:---:|:---:|:---:|:---:|
> > | Marmousi (MSE) | 0.6343 | 0.2754 | 0.2973 | 0.3228 | 0.3840 |
> > | Salt body (MSE) | 0.2480 | 0.1674 | 0.1895 | 0.3286 | 0.2713 |
> >
> > Although an appropriate weighting factor must be selected, our proposed IG-FWI method exhibits relative insensitivity to intermediate values of the weighting factor. In practice, when data quality and the initial model are reliable, a larger $a$ can be adopted to accelerate convergence; conversely, a smaller $a$ is advisable when data or initial model quality is poor.
> >
> >
> >
> > >**Q3: In Table 1, given smooth initial models, MPE-FWI performs worse than CR-FWI baselines except for Marmousi, but it should have slower eigenvalue decay. Could you please provide an explanation or analysis of these results?**
> >
> > Thanks for your careful review. On the complex salt model, both conventional FWI and MPE-FWI yield worse results than the CR-FWI baselines, even when starting from a smooth initial velocity model. This is due to numerical scattering from the finite-difference discretization, combined with strong wavefield complexities (e.g., multiples and converted waves) generated by the intricate top-salt geometry. Such challenges make this model significantly more difficult for conventional FWI than the Marmousi case, which is a well-known issue in the seismic community. These inherent limitations motivates the development of our robust IG-FWI approach, which successfully acquire satisfactory inversion results.
> >
> >
> >
> > >**Q4: [minor] In Equation 4 and Equation 7, there is $({\bf x}^{\prime},t^{\prime})$ before dt^{\prime}d{\bf x}^{\prime}. What’s the meaning of this notation? From my understanding, they are redundant.**
> >
> > Thank you for your comment. Our initial intention was to highlight the variables of integration. We have followed this suggestion and removed it from Equations 4 and 7 in the revised manuscript.
> >
> >
> >
> > We will include the above discussions and additional experiments in the revised manuscript, and all revisions made to the paper will be highlighted in blue for your convenience. Should you need further information, please let us know. We look forward to hearing your feedback!

---

### Official Review · Reviewer_sJvk · 2025-11-01

**Soundness:** 3
**Presentation:** 3
**Contribution:** 2
**Rating:** 4
**Confidence:** 3

**Summary:**

The paper develops a wave-based Neural Tangent Kernel (NTK) analysis for continuous-representation FWI (CR-FWI), explains why INRs help (and why they converge slowly at high frequency), and proposes practical CR-FWI variants (including a hybrid INR + multiresolution grid, IG-FWI). The submission claims strong empirical results on standard FWI benchmarks (Marmousi, SEG/EAGE Salt & Overthrust, 2004 BP, 2014 Chevron).

**Strengths:**

- Interdisciplinary theoretical contribution. The wave-based NTK framework is a thoughtful specialization of NTK tools to FWI that helps explain observed phenomena (robustness to initialization, spectral bias / slow high-freq convergence). This new perspective can guide architecture and sampling choices.

- Use of standard, realistic benchmarks. The authors report results on well-known FWI models (Marmousi, SEG/EAGE Salt & Overthrust, BP, Chevron). Using these makes the claims more credible to applied communities and shows practical intent.

- Actionable design proposal (IG-FWI). The hybrid INR + multi-resolution grid is a practical idea that directly follows from the kernel eigenvalue arguments and aims to trade off robustness vs convergence speed.

**Weaknesses:**

- Ablation & failure modes. The paper claims IG-FWI is an “optimal trade-off” — I’d expect ablations that vary eigenvalue decay (e.g., different INR frequency bases, different grid scales) and show how performance changes. Please also show failure cases.

- The paper does not compare CR-FWI or IG-FWI with data-driven inversion methods (e.g., supervised CNN-based methods) on synthetic datasets such as OpenFWI. These baselines are now standard in the ML-driven FWI literature. The omission limits the ability to judge the practical competitiveness of the proposed framework relative to current deep-learning-based inversion systems.

**Questions:**

see weekness.

---

> ### Author Response · Authors · 2025-11-19
> **Response to Reviewer sJvk**
>
> We sincerely thank the reviewer for the valuable and constructive comments! We have carefully conducted further discussions and supplemented additional experiments based on your comments. Please see the details in the following and the revised manuscript.
>
> >**W1: Ablation & failure modes. The paper claims IG-FWI is an “optimal trade-off” — I’d expect ablations that vary eigenvalue decay (e.g., different INR frequency bases, different grid scales) and show how performance changes. Please also show failure cases.**
>
> Thanks! We have revised the manuscript to replace the imprecise phrase "optimal trade-off" with the more accurate description "a more balanced trade-off". Additionally, we have conducted a comprehensive ablation study to investigate the impact of key parameters (e.g., different INR frequency bases and different grid scales), as detailed in the revised manuscript (See Fig. 9). We also paste tables here for your convenience. The results show that both excessively high and low grid resolutions in the MPE, as well as frequency settings in the tiny-INR, may reduce the quality of inversion results. Moreover, the inversion results of IG-FWI are relatively robust to the hyperparameters within the appropriate range.
>
> **Table. Ablation experiment about frequency bases in IG-FWI.**
>
> | Frequency (Hz) | 20 | 25 | 30 | 35 | 40 |
> |:---------------|:--------------:|:----------:|:--------------:|:-------------:|--------:|
> |Marmousi (MSE) | 0.9605 | 0.3921 | 0.2973 | 0.3046 | 0.4219 |
> |Salt body (MSE) | 0.6396 | 0.2810 | 0.1895 | 0.1475 | 0.1429 |
>
> **Table. Ablation experiment about grid resolution in IG-FWI.**
>
> | Resolution | 30 | 40 | 50 | 60 | 70|
> |:---------------|:--------------:|:----------:|:--------------:|:-------------:|--------:|
> |Marmousi (MSE) | 0.2232 | 0.2184 | 0.2973 | 0.3178 | 0.2522 |
> |Salt body (MSE) | 0.2455 | 0.2282 | 0.1895 | 0.1859 | 0.2360 |
>
> Furthermore, we present two failure cases using our proposed methods in Appendix C.4 of the revised manuscript, using the Canadian Foothills model with irregular topography (See Fig. 32) and the Overthrust model with a near-surface low-speed layer (See Fig. 33). The performance degradation in these challenging scenarios can be attributed to complex wavefield propagation patterns [R1] that our current velocity parameterization struggles to accurately capture, which guide our subsequent algorithm design and integration with existing seismic techniques (e.g., illumination compensation techniques) in future research.
>
>
> [R1] Dinghui Yang, Xingpeng Dong, Jiandong Huang, Zhilong Fang, Xueyuan Huang, Shaolin Liu, Mengxue Liu, and Weijuan Meng. High-resolution full-waveform seismic imaging: Progresses, challenges, and prospects. Science China Earth Sciences, pp.1–28,2025.
>
>
> >**W2: The paper does not compare CR-FWI or IG-FWI with data-driven inversion methods (e.g., supervised CNN-based methods) on synthetic datasets such as OpenFWI. These baselines are now standard in the ML-driven FWI literature. The omission limits the ability to judge the practical competitiveness of the proposed framework relative to current deep-learning-based inversion systems.**
>
> Thanks for your valuable suggestion. We have presented additional comparisons with a data-driven FWI (e.g., supervised CNN-based method, InversionNet [R2]) method on OpenFWI datasets, as detailed in Appendix C.3 of the revised manuscript (See Figs. 26-31 and Tab. 2). We also paste the table here for your convenience. The results show that the LR-FWI and IG-FWI methods may yield better performance than some supervised CNN-based methods.
>
> **Table: Comparisons with data-driven FWI and CR-FWI methods.**
>
> | Methods | InversionNet | ADFWI | IFWI | LR-FWI (Ours) | IG-FWI (Ours) |
> |:---------------|:--------------:|:----------:|:--------------:|:-------------:|--------:|
> | FaltVel (MSE) | 36.78 | 21.30 | 11.72 | 6.65 | 6.94 |
> | CurveVel (MSE) | 94.16 | 57.07 | 16.87 | 9.89 | 7.93 |
> | FaltFault (MSE) | 113.18 | 48.23 | 46.59 | 19.59 | 28.84 |
> | CurveFault (MSE) | 100.09 | 99.14 | 65.89 | 60.23 | 24.61 |
> | Style (MSE) | 53.76 | 27.66 | 48.25 | 14.56 | 17.08 |
>
>
> While supervised frameworks suffer from generalization issues caused by the distribution shift between synthetic training data (e.g., from OpenFWI) and complex field models (e.g., Marmousi, Overthrust), their computational speed remains highly appealing. A promising pathway is to embed such methods within our proposed CR-FWI framework to rapidly generate initial models, which we plan to explore in future work.
>
>
> [R2] Yue Wu and Youzuo Lin. Inversionnet: An efficient and accurate data-driven full waveform inversion. IEEE Transactions on Computational Imaging,6:419–433,2019b.
>
> We will include the above discussions and additional experiments in the revised manuscript, and all revisions made to the paper will be highlighted in blue for your convenience. Should you need further information, please let us know. We look forward to hearing your feedback!

---

> > ### Author Response · Authors · 2025-11-27
> >
> > Dear Reviewer sJvk
> >
> > I hope this message finds you well. If there are any additional feedback or points for us to consider, please let us know. Your insights are invaluable to us, and we are committed to addressing any remaining concerns to further improve our work.
> >
> > Thank you for your time and effort in reviewing our paper.

---

### Comment · Area_Chair_bUkS · 2025-11-27
**Post rebuttal discussion**

Dear reviewers who haven't replied so far: the end of the rebuttal phase is less than a week away. Please check and acknowledge the author's rebuttal posts (and of course feel free to ask follow up questions). Thanks in advance!

---

### Author Response · Authors · 2025-11-29
**Brief summary of the discussion phase**

Dear AC and Reviewers,

We sincerely appreciate the time and effort you have dedicated to reviewing our manuscript. We fully understand the additional workload and challenges brought about by the current situation, and we truly appreciate your continued efforts.

In the hope of facilitating your decision-making process, we would like to provide a summary of the discussion phase:

**Reviewer Lw1m** raised questions regarding the ablation experiment for the weighting factor, convergence evidence, and the originality of the methodological contribution. Following our clarifications, the reviewer explicitly confirmed that the questions were answered and raised the score from 6 to **8** before the reverting process.

**Reviewer GUpL** raised questions regarding the linearity of the wave equation and 3D performance. After our replies, the reviewer explicitly confirmed that the two important points are addressed properly and maintained a positive score of **8** before the reverting process.

**Reviewer 4aqd** raised questions regarding the ablation experiment for the weighting factor, comparison with data-driven FWI, and analysis of eigenvalue decay behavior for LR-FWI. We conducted a comprehensive ablation study, additional comparisons with a data-driven FWI (InversionNet) method on five OpenFWI datasets, and added a discussion on the motivations for the proposed LR-FWI method and its observed eigenvalue decay behavior. After our replies, this reviewer maintained a positive score of **8** before the reverting process.

**Reviewer sJvk** gave an initial score of 4 and raised questions regarding the ablation experiments for hyperparameters and comparison experiments with the data-driven FWI method. We conducted a comprehensive ablation study for all hyperparameters mentioned by the reviewer on multiple datasets and present additional comparisons with a data-driven FWI (InversionNet) method on five OpenFWI datasets.

As a result, the final scores before the reverting process were **8 / 8 / 8 / 4**. All revisions and additional clarifications provided during the discussion have been carefully incorporated into the updated manuscript, with changes highlighted in blue for your convenience. We hope this summary could assist your re-evaluation process.

---

### Meta-Review · Area_Chair_VqBb · 2026-01-07

**Summary:**

The paper investigates Continuous Representation Full-Waveform Inversion (CR-FWI), focusing on why Implicit Neural Representations (INR) improve robustness to initial models but suffer from slow high-frequency convergence. The authors introduce a "Wave-based Neural Tangent Kernel (NTK)" framework to analyze these phenomena. They derive that the eigenvalue decay rate of the wave-based NTK dictates the trade-off between robustness and convergence speed . Based on this theory, they propose two new methods: Low-Rank FWI (LR-FWI) and Integrated Grid FWI (IG-FWI), which combine INR with multi-resolution grids to achieve a balanced eigenvalue decay .

Strengths

- Theoretical Novelty: The adaptation of NTK theory to the specific physics of FWI (Wave-based NTK) provides a valuable theoretical grounding for understanding the "spectral bias" observed in deep learning-based inversion.

- Methodological Contribution: The proposed IG-FWI and LR-FWI methods are theoretically motivated by the spectral analysis and demonstrate clear empirical improvements in balancing convergence speed with robustness.


- Comprehensive Evaluation: The paper evaluates performance on standard, challenging geophysical benchmarks including Marmousi, SEG/EAGE Salt & Overthrust, 2004 BP, and the realistic 2014 Chevron blind test .

**Reviewer Concerns:**

The reviewers initially raised concerns regarding the lack of 3D experiments, the absence of comparisons with data-driven baselines, and the need for ablation studies on hyperparameters.

The authors provided a thorough rebuttal that addressed these points:

- 3D Validation: They added experiments on the 3D SEG/EAGE Overthrust model, demonstrating the method's scalability.

- Baselines: They included comparisons with InversionNet (a supervised CNN method) on the OpenFWI dataset, showing favorable performance.

- Ablations & Failure Modes: They provided sensitivity analyses for the weighting factor $\alpha$ and grid resolution, as well as an analysis of failure cases in complex topographies.

**Reviewer Scores:**

The paper makes a solid contribution by bridging the gap between deep learning theory (NTK) and geophysical inversion (FWI). The theoretical insights regarding eigenvalue decay offer a plausible explanation for the behavior of INRs in inverse problems. The authors effectively addressed the reviewers' concerns regarding scalability and baselines during the rebuttal phase.

---

### Decision · Program_Chairs · 2026-01-26

Accept (Poster)